# An improved global land cover mapping in 2015 with 30 m resolution (GLC-2015) based on a multi-source product fusion approach

Bingjie Li [1], Xiaocong Xu [1], Xiaoping Liu [1 2], Qian Shi [1], Haoming Zhuang [1], Yaotong Cai [1] and Da He [1]

[1]School of Geography and Planning, Sun Yat-Sen University, Guangzhou, 510275, China

[2]Southern Marine Science and Engineering Guangdong Laboratory (Zhuhai), Zhuhai, 519080, China

*Correspondence to*: Xiaoping Liu (liuxp3@mail.sysu.edu.cn)

**Abstract.** Global land cover (GLC) information with fine spatial resolution is a fundamental data input for studies on biogeochemical cycles of the Earth system and global climate change. Although there are several public GLC products with 30 m resolution, considerable inconsistencies were found among them especially in fragmented regions and transition zones, which brings great uncertainties to various application tasks. In this paper, we developed an improved global land cover map in 2015 with 30 m resolution (GLC-2015) by fusing multiple existing land cover (LC) products based on the Dempster-Shafer theory of evidence (DSET). Firstly, we used more than 160,000 global point-based samples to locally evaluated the reliability of the input products for each land cover class within each 4°×4° geographical grid for the establishment of the basic probability assignment (BPA) function. Then, the Dempster's rule of combination was used for each 30 m pixel to derive the combined probability mass of each possible land cover class from all the candidate maps. Finally, each pixel was determined with a land cover class based on a decision rule. Through this fusing process, each pixel is expected to be assigned with the land cover class that contributes to achieve a higher accuracy. We assessed our product separately with 34,711 global point-based samples and 201 global patch-based samples. Results show that, the GLC-2015 map achieved the highest mapping performance globally, continentally, and eco-regionally compared with the existing 30 m GLC maps, with an overall accuracy of 79.5% (83.6%) and a kappa coefficient of 0.757 (0.566) against the point-based (patch-based) validation samples. Additionally, we found that the GLC-2015 map showed substantial outperformance in the areas of

inconsistency, with an accuracy improvement of 19.3%-28.0% in areas of moderate inconsistency, and

27.5%-29.7% in areas of high inconsistency. Hopefully, this improved GLC-2015 product can be applied

to reduce uncertainties in the research on global environmental changes, ecosystem service assessments,

and hazard damage evaluations, etc. The GLC-2015 map developed in this study is available at

https://doi.org/10.6084/m9.figshare.22358143.v2 (Li et al., 2022).

## 1. Introduction

Land cover (LC), influenced by both nature and human activities (Running, 2008; Gong et al., 2013;

Song et al., 2018; Liu et al., 2021a), is a significant component of the Earth system (Yang and Huang,

2021). Global land cover (GLC) products can serve as fundamental data for various studies, such as

climate and environmental changes (Bounoua et al., 2002; Foley et al., 2005; Grimm et al., 2008; Yang

et al., 2013; Schewe et al., 2019), food security (Verburg et al., 2013; Ban et al., 2015), carbon cycling

(Moody and Woodcock, 1994; Defries et al., 2002; Gómez et al., 2016), biodiversity conservation

(Chapin et al., 2000; Giri et al., 2005) and land management (Mayaux et al., 2004; Verburg et al., 2011).

Therefore, there is a pressing need for detailed, accurate, and high-quality GLC product to support global

change research and sustainable development.

In the preliminary stage, LC mapping mainly relied on visual interpretation, which is time-

consuming, labor-intensive and difficult to be applied at the global scale (Gong, 2012). In recent decades,

satellite remote sensing data, which can provide information of large area coverage and long-term

monitoring, has been adopted to generate GLC products. With coarse resolution satellite data such as

Advanced Very High Resolution Radiometer (AVHRR), Moderate Resolution Imaging

Spectroradiometer (MODIS), Medium Resolution Imaging Spectrometer (MERIS), and Global Land

Surface Satellite (GLASS), a variety of GLC products have been developed at 5 km to 300 m

resolution(Loveland et al., 2000; Hansen et al., 2000; Bartholomé and Belward, 2005; Friedl et al., 2010;

Defourny et al., 2018; Liu et al., 2020a). Although these GLC products have been widely applied to many

applications, it has been proved that the differences between sensors, classification systems, and

considerably low accuracies in areas prevent harmonization of these products (Herold et al., 2008;

Verburg et al., 2011; Grekousis et al., 2015). Also, these products are far from providing enough fine

spatial details of LC due to their relatively coarse spatial resolution, which does not meet the demand of

many studies (Giri et al., 2013; Yang et al., 2017). To allow researches which can capture most human activity, finer-resolution (e.g., 30 m) GLC products are demanded (Giri et al., 2013).

With the free accessibility of high-resolution satellite remote sensing data, GLC mapping at fine resolution has been successfully conducted. Using Landsat imagery, there has been a milestone achievement that the two GLC products are generated with fine resolution of 30 m, namely Finer Resolution Observation and Monitoring of Global Land Cover product (FROM_GLC)(Gong et al., 2013) and Globeland30 (Chen et al., 2015). After that, a 30 m-resolution GLC mapping in 2017 was achieved using the first all-season sample set (Li et al., 2017). More recently, Zhang et al. (2021) used both Landsat time series imagery and high-quality training data from the Global Spatial Temporal Spectra Library (GSPECLib) to produce a 30 m GLC map in 2015 (GLC_FCS30) with a two-level classification scheme. Several attempts have been made to improve accuracy of 30 m GLC products which are prevail in the generation of GLC mapping task over the last few years. FROM_GLC was created by employing four classification algorithms to classify the Landsat images and choosing time series of MODIS EVI data for training and test. Globeland30 was created by proposing a pixel-object-knowledge-based (POK) method to assure consistency and accuracy. GLC_FCS30 was generated by adopting local adaptive random forest models with high-quality training samples derived from GSPECLib. The Globeland30, FROM_GLC, and GLC_FCS30 are excellent and indispensable GLC products which have contributed much to various research, such as biodiversity conservation (Wu et al., 2020; Meng et al., 2023), climate change (Kim et al., 2016; Xue et al., 2021; Zheng et al., 2022), and land management (Shafizadeh-Moghadam et al., 2019). In addition to these multiple-class GLC products, GLC products for individual LC classes, such as cropland (Yu et al., 2013; Lu et al., 2020), forest (Hansen et al., 2013; Shimada et al., 2014; Zhang et al., 2020), wetland (Hu et al., 2017; Zhang et al., 2023), water (Liao et al., 2014; Pekel et al., 2016; Pickens et al., 2020), and impervious surfaces (Gong et al., 2020; Huang et al., 2021; Huang et al., 2022; Liu et al., 2020b), have been successfully generated.

Despite the great efforts in producing more accurate products, the existing 30 m GLC products still show unstable performance in certain LC classes and some specific areas (Sun et al., 2016; Kang et al., 2020). Furthermore, the existing 30 m products showed great agreement in overall spatial distribution patterns but significant spatial inconsistency in some specific areas (heterogeneous areas and transition zones) and spectrally similar classes (forest and shrubland, cropland and grassland) (Gao et al., 2020;

Liu et al., 2021b). The spatial inconsistency between the existing 30m GLC products are resulted from
differences in their classification systems, classification techniques employed, source data, and spatial
distribution and size of training samples (Yang et al., 2017; Gao et al., 2020). Due to the aforesaid
limitations, users of GLC products still have difficulties in an appropriate selection of data for their
specific application. Ultimately, this situation leads to uncertainties in outcomes of related researches
when different 30 m GLC products are used. For GLC mapping with fine spatial resolution, more efforts
should be focused on improving the mapping in heterogenous and fragmented landscape (Herold et al.,
2008; Liu et al., 2021b). Therefore, it is pressing to generate a more accurate and reliable GLC product
with high classification accuracy, especially for spatially inconsistent regions and low-accuracy LC
classes.
According to Gong et al. (2016), inconsistencies between LC products indicate available
complementary information and more robust and reliable data can be generated by integrating the input
maps with the data fusion method. Given that different maps have disagreement and provide accurate
information in different locations, we can make a best choice for the class label assigned to each pixel
by weighting the credibility of all the available information and combining them through a decision rule
(Clinton et al., 2015). In this way, the output map of integration on input maps can reduce the overall risk
of assigning a wrong class label to a pixel and at least achieve the average performance of input maps.
Several attempts have been made to produce an accurate and consistent LC map using various methods,
such as majority voting (MV), fuzzy agreement and Bayesian theory. Iwao et al. (2011) created a GLC
map based on a simple majority voting method. Jung et al. (2006) generated a 1km GLC map by
combination of MODIS, GLC2000 and GLCC data based on fuzzy agreement scoring. Subsequently,
Fritz et al. (2011) extended the synergy method of Jung et al. (2006) by ranking LC maps and mapped
the cropland extent in Sub-Saharan Africa. See et al. (2015) generated two GLC products by integrating
medium resolution LC products with geographically weighted regression (GWR). Gengler and Bogaert
(2018) proposed a Bayesian data fusion method and applied it to the LC mapping for a specific region in
Belgium. All these researches have demonstrated that the fusion method can create an integrated LC
product where the mapping accuracy is greatly improved by combing the best of candidate maps.
However, the MV method is sensitive to the quality of the candidate maps and has significant
uncertainties when the input products exhibit great disagreement (Chen and Venkataramanan, 2005). The
fuzzy agreement is highly subjective since it depends on expert assessment, while the Bayesian theory
requires a prior knowledge or conditional probabilities and fails to handle the states of ignorance (Liu
and Xu, 2021).

The Dempster-Shafer theory of evidence (DSET) is an evidence-based approach to reason with

uncertainties. Unlike the majority voting, the DSET method can discount evidence from inaccurate
information with a probability mass that reflects the degree of belief rather than a binary decision (Razi
et al., 2019). In contrast to the Bayesian theory, the DSET can integrate evidence from a variety of sources
without the requirement of prior knowledge (Chen and Venkataramanan, 2005). Moreover, the reliability
of the final fused results based on the DSET method is measured with a total degree of belief. Although
previous literatures focused on the application of the DSET method in multisource data aggregation, very
little research has been conducted globally due to the lack of accurate and sufficient samples and the
demand for adequate computing resources.

In this research, we propose a multi-source product fusion approach on the Google Earth Engine

(GEE) platform to produce an improved GLC product in 2015 (GLC-2015) with 30 m resolution. The
fusion approach we proposed aims to deal with the inconsistency between previous 30 m GLC products
and generate a map which has better mapping performance than any of the candidate maps by evaluating
the mapping accuracy of these existing products at the local scale and choosing the most credible LC
class. To fulfill the purpose, we first performed reliability evaluation, where the accuracy of each product
for each LC class in each $4° \times 4°$ geographical grid is regarded as the evidential probability to create the
basic probability assignment (BPA) function. Then, the BPA values of all the LC classes from different
products were fused according to the Dempster's rule of combination. Finally, the GLC-2015 map was
integrated after a final accepted LC class with the maximum combined probability mass was assigned to
each 30 m pixel. The GLC-2015 map was separately validated with two different validation sets, namely
global point-based samples and global patch-based samples, and compared with the existing products.
Moreover, we provided an analysis for the mapping improvement of the GLC-2015 compared to other
GLC products in areas of high mapping inconsistency. The GLC-2015 map is proved to be accurate and
credible and can significantly improve the mapping accuracy in areas of high inconsistency.
**2. Datasets**
**2.1 Multiple-class GLC products**
Three existing 30 m GLC products with multiple classes, including GlobeLand30, FROM_GLC and
GLC_FCS30, were employed as input maps in the fusion based on DSET. A summary of their detailed
information is shown in Table 1.
GlobeLand30, a widely-used global geo-information product, was produced by the POK-based
method using Landsat and HJ-1 satellite images. Globeland30 products are freely accessible online at
the website (http://www.globalland30.org) for 2000 and 2010. From the accuracy assessment, the
Globeland30 for the year 2010 had an overall accuracy excessed 80.0% using large samples (Chen et al.,
2015). Although the data time of GlobeLand30 is 2010, which has a five-year gap with other products,
it was used because the changed areas of LC caused by the time interval are tiny compared to the global
land area. In addition, there is relatively less uncertainty due to LC changes than due to inaccurate
classification (Xu et al., 2014). Most spatial disagreements between the existing maps are about
classification errors rather than LC changes over the time interval (Mccallum et al., 2006; See et al.,

2015).

FROM_GLC was first generated using numerous Landsat images, which has a fine classification
system with a two-level structure. It achieved an OA of 64.5% through validation with the complete test
samples and 71.5% with a subset of test samples in homogeneous areas (Gong et al., 2013).
GLC_FCS30 was developed using Landsat time series data and large training samples from the
GSPECLib. It has a two-level classification scheme that contains 16 global LCCS LC classes and 14
detailed regional LC classes. The overall accuracy of the GLC_FCS30 according to LCCS level-1
validation scheme reached 71.4% (Zhang et al., 2021).
**Table 1. Detailed information of GLC products and national-scale LC products used in this paper.**

| Product name | Satellite sensors | Year of reference | Access | Literature |
|---|---|---|---|---|
| Globeland30 | Landsat TM/ETM+<br><br>HJ-1 A/B | 2010 | http://www.globallandcover.com/ | (Chen et al., 2015) |
| FROM_GLC | Landsat TM/ETM+/OLI | 2015 | http://data.ess.tsinghua.edu.cn/ | (Gong et al., 2013) |
| GLC_FCS30 | Landsat OLI | 2015 | https://doi.org/10.5281/zenodo.3986872 | (Zhang et al., 2021) |

| | | | | | |
|---|---|---|---|---|---|
| GAUD | Landsat TM/ETM+/OLI | 2015 | https://doi.org/10.6084/m9.figshare.115131 78.v1 | (Liu et al., 2020c) |
| GFC | Landsat TM/ETM+ | 2015 | http://earthenginepartners.appspot.com/scie nce-2013-global-forest | (Hansen et al., 2013) |
| JRC GSW | Landsat TM/ETM+/OLI | 2015 | http://global-surface-water.appspot.com/ | (Pekel et al., 2016) |
| GMW | ALOS PALSAR Landsat TM/ETM+ | 2015 | https://data.unep-wcmc.org/datasets/45 | (Bunting et al., 2018) |
| NLCD 2016 | Landsat TM /OLI | 2016 | https://www.mrlc.gov/data/nlcd-2016-land-cover-conus | (Yang et al., 2018) |
| CLUD | Landsat TM HJ-1 CBERS-1 | 2015 | / | (Liu et al., 2014) |
| CLCD | Landsat TM/ETM+/OLI | 2015 | https://doi.org/10.5281/zenodo.4417810 | (Yang and Huang, 2021) |

**2.2 Single-class GLC products**

To improve the quality of the fusing result, a set of highly qualified GLC products with single class at 30 m fine resolution were also used. Compared to the multiple-class GLC products, these single-class GLC products are more likely to provide accurate information since they usually focus on promoting the mapping performance of a specific LC class. These products include Global Forest Change (GFC) (Hansen et al., 2013), Global Annual Urban Dynamics (GAUD) (Liu et al., 2020b), Joint Research Centre's Global Surface Water (JRC GSW) (Pekel et al., 2016), and Global Mangrove Watch (GMW) (Bunting et al., 2018). While these single-class products are either annual or multi-epoch, we only selected these products in the target year of 2015. The background information of these single-class products was considered as another land cover class (e.g., non-water) participating in the fusion. The accuracy of the background information was defaulted to 0 since it did not provide information about any of the other nine categories in our classification system. Table 1 also describes the information of these selected single-class GLC products.

GFC was resulted from a time-series analysis of growing season Landsat scenes, aiming to provide information about global tree cover extent, gain, and loss at a 30 m spatial resolution. The accuracy

assessment was performed at global and climate domain scales and the forest gain reached an overall
accuracy of 99.6% and forest loss reached 99.7% across the globe (Hansen et al., 2013). Up to now, it
has a temporary coverage from 2000 to 2020.

GAUD, which provides 30m annual urban extent for the time period of 1985 to 2015, was generated

using numerous Landsat images with both data fusion approach and temporal segmentation approach on
the GEE platform. Validation was conducted across different urban ecoregions and the globe by the
product developer. The accuracy of mapping urbanized year was 76.0% for the period of 1985 to 2000
and 82.0% for the period of 2000 to 2015 at humid regions worldwide (Liu et al., 2020c).

JRC GSW dataset provides a monthly presentation of global surface water changes from 1984 to

2015 at a fine 30 m resolution. Expert systems, visual analytics and evidential reasoning were exploited
to detect water extent and changes. Based on 40,124 validation points over the globe and across the 32
years, commission accuracies were determined with overall accuracies of 99.45% (TM), 99.35% (ETM+)
and 99.54% (OLI) and omission accuracies were reflected in overall accuracies of 97.01% (TM), 95.79%
(ETM+) and 96.25%(OLI) (Pekel et al., 2016). We used the GSW Yearly Water Classification History
v1.1 in the GEE catalog. A single 'waterClass' band is present in each image that provides the water's
seasonality throughout the year with four types: no data, no water, seasonal water, and permanent water.
Since the seasonal water in GSW data is not as reliable as the permanent water (Meyer et al., 2020), we
selected permanent water bodies and excluded seasonal water bodies.

GMW dataset was produced as a result of the GMW initiative, which aims to provide consistent

information of mangrove extent. The global mangrove map in 2010 was generated as a baseline map
employing the Extremely Randomized Trees classifier to classify ALOS PALSAR and Landsat imagery.
Assessed by a total of 53,878 sample points globally, the overall accuracy of the baseline map reached
95.3% and the producer's accuracy achieved 94.0% (Bunting et al., 2018). Based on the baseline in 2010,
mangrove extent maps for six epochs between 1996 and 2016 have been established and annual change
monitoring from 2018 and onwards are undertaken.
**2.3 National-scale LC products**

Land cover products which focus on a national scale are more likely to possess higher accuracy

because they were produced by experts who have good knowledge of land cover classes nationally. Thus,
the National Land Cover Database 2016 (NLCD 2016) for the year 2016 over the conterminous United
States (CONUS) (Yang et al., 2018), China's land-use/cover dataset (CLUD) (Liu et al., 2014) for 2015,
and the annual China land cover dataset (CLCD) (Yang and Huang, 2021) for 2015 were also included
in the fusion. The detailed information of these national-scale products was listed in Table 1.
NLCD 2016 database, which provides continuous and accurate information about land cover and
change from 2001 to 2016 at an interval of 2 or 3 years, was produced based on a pixel- and object-based
approach and an effective post-classification process (Yang et al., 2018). The level-1 and level-2 overall
accuracy of NLCD 2016 database for 2016 was 90.6% and 86.4% for CONUS, respectively (Wickham
et al., 2021). CLUD, developed by the digital interpretation method using Landsat images, provide land
cover information over China from 1980s to 2015. The overall accuracy of CLUD reached 94.3% and
91.2% for level-1 and level-2 land cover classes, respectively (Liu et al., 2014). CLCD was generated
with stable training samples derived from CLUD and Landsat time series. Assessed with 5,463 validation
samples, CLCD obtained an overall accuracy of 79.31% (Yang and Huang, 2021).
**2.4 Global point-based and patch-based samples**
In this study, we collected two sets of global samples, namely the global point-based samples and the
global patch-based samples. To collect representative and sufficient samples efficiently, we divided the
world's terrestrial area into 4° × 4° geographical grids. A total of 1,507 grids are distributed evenly across
the globe, shown as Fig. 1.

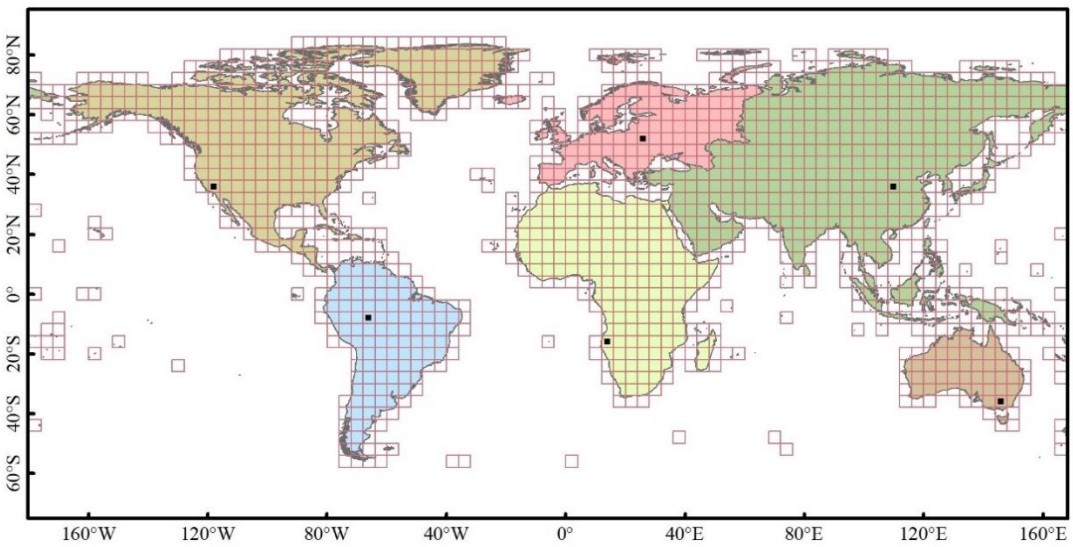

**Figure 1. Spatial distribution of the 4° × 4° geographical grids over the world. Six black rectangle tiles with**
**size of 0.25° were used for visual comparation between our product and other three products.**

To derive the global point-based samples, we adopted stratified random sampling in each grid. The

stratified random sampling depends on area ratio of classes from a land cover product. We used the
FROM_GLC as prior knowledge rather than the Globeland30 and GLC_FCS30 with two considerations:
(1) the FROM_GLC has the same data time as our target map (GLC-2015) while the Globeland30 has a
5-year interval from our samples, which affects the size of samples for each LC class; (2)the 10 level-1
land cover classes of the FROM_GLC is similar to that in the classification system of the GLC-2015,
while the GLC_FCS30 has differences with the GLC-2015 in the classification scheme and definition of
land cover classes. First, the FROM_GLC product was used to calculate the area ratio of each LC class.
Then, points were randomly extracted from the FROM_GLC according to the area ratio and spatial
location of each class. Finally, more than 200,000 global samples were collected. Through the sampling
method mentioned above, the global point-based samples were even across the globe and sufficient for
each class in each grid. Therefore, more than 50 points could be easily derived for classes with a small
area ratio in the $4° \times 4°$ grid. The FROM_GLC shows low accuracy for some LC classes, especially for
cropland and forest (Gao et al., 2020; Liu et al., 2021b; Zhang et al., 2021; Zhang et al., 2022). If the
global samples were extracted with LC class label from the FROM_GLC, there would be inevitable
errors. Therefore, the FROM_GLC was only used to determine the size and location of samples for each
class. Instead, all the points were manually labeled according to Google Earth high-resolution images.
The whole sample set was randomly split into two subsets: 80% of the global samples were used to assess
the accuracy of each GLC product for various LC classes at the global scale and in each grid. The
remaining 20% were used for the validation of the GLC-2015 map and data inter-comparison between
different products. Figure 2 presents the distribution of the whole global point-based samples and the
subset for accuracy assessment and data inter-comparison.

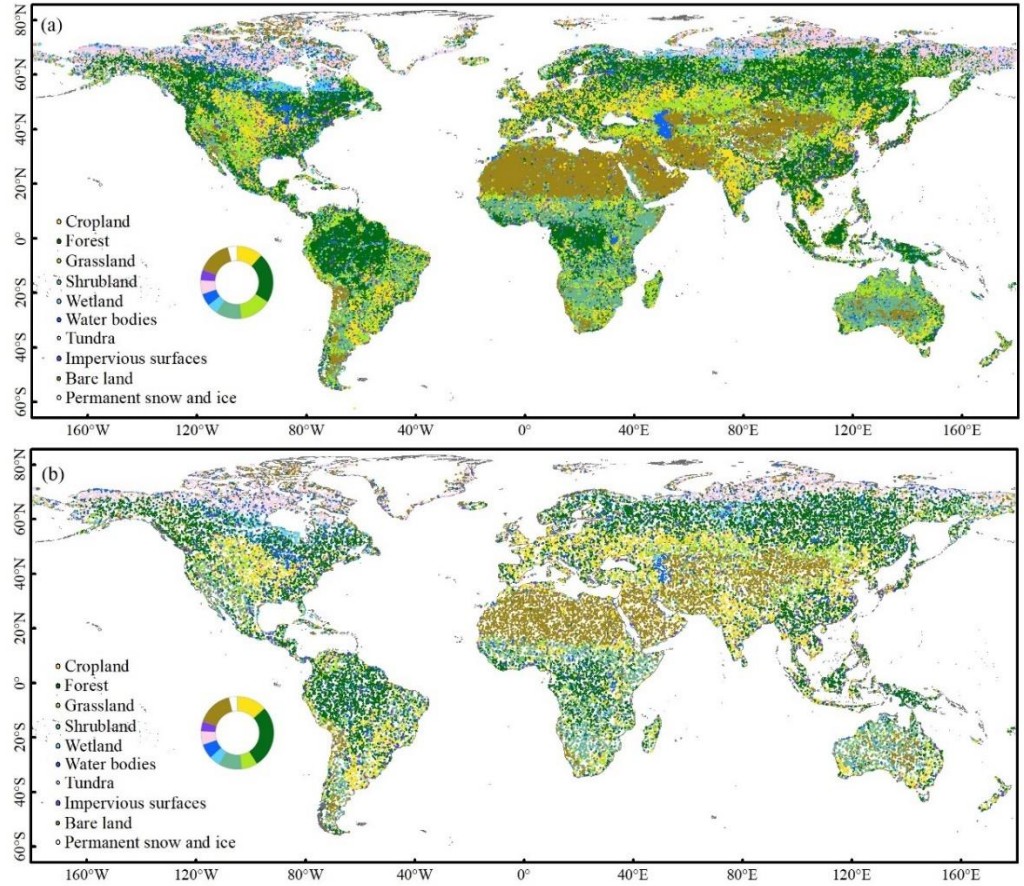


**Figure 2. Spatial distribution of (a) the global point-based samples, (b) the subset of the global point-based**
**samples for accuracy assessment and data inter-comparison, the proportions of each LC class are shown in**
**the pie chart.**
To verify the consistency between the GLC-2015 and the actual pattern of the landscape at the local
scale, we also established the global patch-based samples. Simple random sampling was used to derive
5 km × 5 km blocks over the world's terrestrial area and across different ecoregions because it is easy to
perform and capable to augment the sample size from target areas (Pengra et al., 2020). Since
inconsistency between current GLC maps tends to appear in the heterogeneous areas, such as fragmented
regions and transition zones, we slightly increased the sample size for areas with the heterogeneous
landscape to better evaluate our mapping results. In total, there were 201 blocks selected as the global
patch-based samples, as displayed in Fig. 3a. Then, for each block in the patch-based samples, we used
ArcGIS 10.5 software to derive polygons (patches) of various sizes which captured the real landscape on
the high-resolution images. Meanwhile, each polygon was manually labeled with a LC class. Four
examples of producing patch-based samples are shown in Fig. 3b and c.

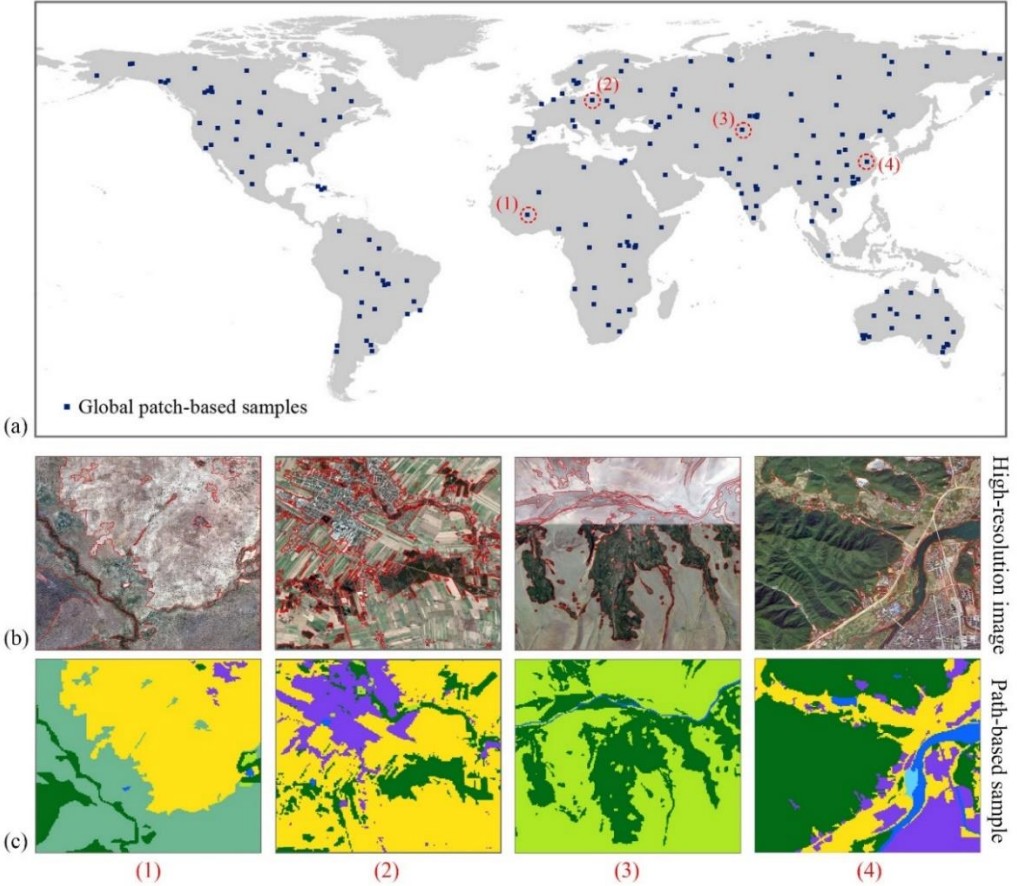


**Figure 3. Spatial distribution and selected examples of the global patch-based samples. The location of 5 km**
**× 5 km patch-based samples are shown as panel (a), the locations of four selected samples are remarked by**
**red dash circles. Panels (b) and (c) illustrate the production of global patch-based samples on manual**
**interpretation. The red lines in high-resolution images circa 2015 are results after vectorization using ArcGIS**
**10.5 software. Four corresponding patch-based samples are shown as (c).**
**3. Methods**
In this study, we proposed a multi-source product fusion method to produce the GLC-2015 map. The
procedure mainly comprised the fusion based on the Dempster-Shafer theory of evidence (DSET),
accuracy assessment and data inter-comparison, as shown in Fig. 4. The basic of this study is the fusion
of multi-source products based on DSET. The fusion method was performed at the pixel level and it
involves the following three main steps: (1) Construct the basic probability assignment (BPA) function
of each pixel that belongs to each LC class considering the accuracy assessment of various products; (2)
calculate the combined probability mass for each class per pixel using the Dempster's rule of combination;
and (3) determine the finally accepted LC class per pixel by a decision rule. Afterwards, pixels with a
determined LC class were integrated to generate a new map. For large-scale or global land cover mapping,
previous researchers divided the study area into a lot of sub-regions and conducted classification in each
sub-region on GEE (Gong et al., 2020; Liu et al., 2020c; Huang et al., 2021; Jin et al., 2022; Zhang et al.,
2021; Zhao et al., 2021). The shape and size of sub-region vary in previous work, such as hexagons with
a side length of 2°, geographical grids with a size of 1°×1°, 3.5°×3.5°, 5°×5°, or 10°×10°. When deciding
on the size of sub-regions, two important factors should be considered. The size of samples in each sub-
region should be sufficient so that the rare land cover classes will not be missed. On the other hand, it is
impossible to implement mapping work at a sub-region as larger as we want due to memory constraints.
To determine the appropriate size, we tested different sizes of the sub-region (see Table S1). Result shows
that dividing the study area into 4°×4° grids performed best. Therefore, we split the world's terrestrial
area into 1507 4°×4° geographical grids. The entire framework was implemented in all 4° × 4°
geographical grids on the GEE platform.

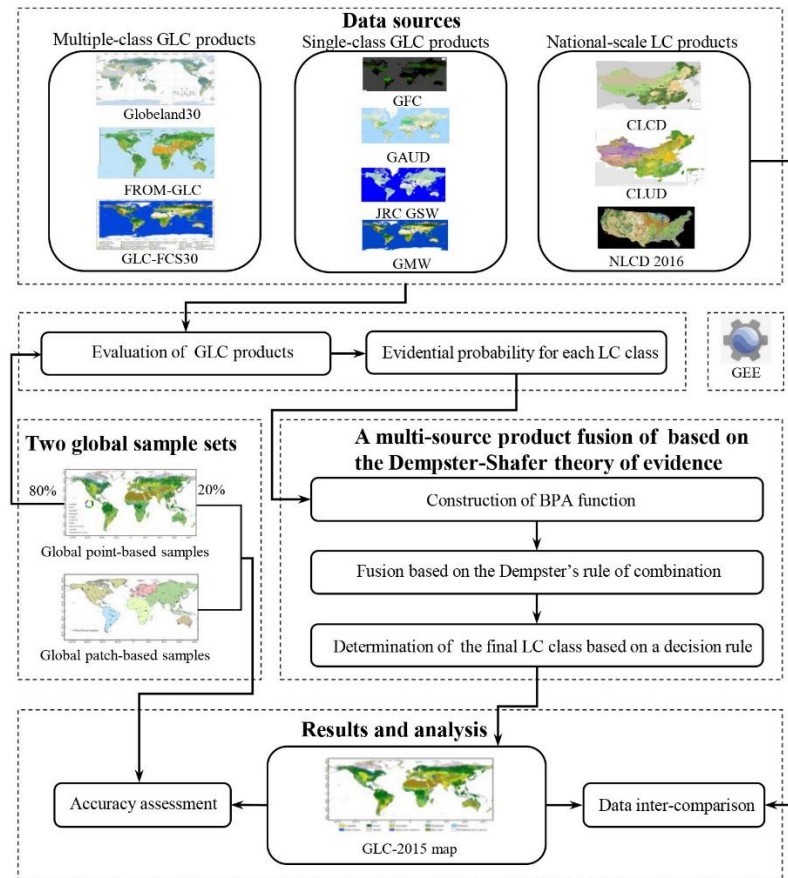


**Figure 4. The framework for generating the GLC-2015 map using a multi-source product fusion approach**
**based on DEST.**
**3.1 Definition of the classification system**
In this study, we adopted the classification system with 10 LC classes, including cropland, forest,
grassland, shrubland, wetland, water bodies, tundra, impervious surfaces, bare land, and permanent snow
and ice (Chen et al., 2015), as listed in Table 2. Due to the applications for different social needs, the
existing GLC products and national-scale LC products were produced with different classification
systems (Tables S2-S3). The GlobeLand30 used a simple classification system that only contained 10
first-level classes. Unlike the GlobeLand30, the FROM_GLC and GLC_FCS30 were classified with a
two-level classification scheme. Through analysis of these systems, we found that the classification
systems are not the same, but they have some agreements. There are both 10 major classes in the
GlobeLand30 and FROM_GLC despite that the definition of some classes differs. Additionally, in
contrast to the GlobeLand30 and FROM_GLC, the level-0 classification system of the GLC_FCS30
lacks tundra. However, in the level-2 detailed LC classes of the GLC_FCS30, lichens and mosses has
little distinction with tundra.
**Table 2. Classification system adopted in this paper.**

| Id | LC class | Definition |
|---|---|---|
| 10 | Cropland | Land areas used for food production and animal feed. |
| 20 | Forest | Land areas dominated by trees with tree canopy cover over 30%, and sparse trees with tree canopy cover between 10%-30%. |
| 30 | Grassland | Land areas dominated by natural grass with a cover over 10%. |
| 40 | Shrubland | Land areas dominated by shrubs with a cover over 30%, including mountain shrubs, deciduous shrubs, evergreen shrubs and desert shrubs with a cover over 10%. |
| 50 | Wetland | Land areas dominated by wetland plants and water bodies. |
| 60 | Water bodies | Land areas covered with accumulated liquid water. |
| 70 | Tundra | Land areas dominated by lichen, moss, hardly perennial herb and shrubs in the polar regions. |
| 80 | Impervious surfaces | Land areas covered with artificial structures. |
| 90 | Bare land | Land areas with scarce vegetation with a cover lower than 10%. |
| 100 | Permanent snow and ice | Land areas dominated by permanent snow, glacier and icecap. |

According to the LC translation tables (Tables S2-S3), the original LC classes of FROM_GLC and
GLC_FCS30, CLUD for 2015, and NLCD 2016 for 2016 were converted into the 10 target land cover
classes based on the similarity of LC definition. Note that cropland in our classification system was
defined as land areas for food production and animal feed. Therefore, pasture in level-2 classes of the
FROM_GLC was converted into cropland rather than grassland. In addition, lichens/mosses in the level-
2 detailed classification system of GLC_FCS30 was converted into tundra.
**3.2 A multi-source product fusion for the GLC-2015 mapping**
The DSET is an effective method widely applied for the fusion of multi-source data. To generate a new
high-quality GLC map, a multi-source product fusion method using DSET was proposed. In the
remainder of the section 3.2, We introduced the overview on the theory and presented the application of
DSET in our mapping process.
**3.2.1 Dempster-Shafer theory of evidence**
The DSET is developed by Dempster and Shafer, which is an extension of Bayesian probability theory.
This theory treats information from different data sources as independent evidence and integrated these
evidences with no requirements regarding the prior knowledge. In the fusion, we assume a classification
process in which all the input data are to be classified into mutually exclusive classes. Let the set $\Omega$ of
these classes be a frame of discrimination. $2^{\Omega}$ is the power set of $\Omega$ that includes all the classes and
their possible unions. We defined the function $m: 2^{\Omega} \to [0,1]$ as the basic probability assignment (BPA)
function if and only if it satisfies $m(\Phi) = 0$ and $\sum_{A \subseteq 2^{\Omega}} m(A) = 1$ with Ø denotes an empty set. For
each class $A \subseteq 2^{\Omega}$, m(A) is called the basic probability mass which can be computed from the BPA
function and represents the degree of support for class A or confidence in class A.

The purpose of fusion is to evaluate and integrate information from multiple sources. In the DSET,

these multi-source data are regarded as different evidence and provide different assessments. To generate
all the evidences, Dempster-Shafer theory of evidence offers a rule. Suppose $m_i(B_j)$ is the basic
probability mass computed from the BPA function for each input data $i$ with $1 \leq i \leq n$ for all classes
$B_j \in 2^{\Omega}$. Dempster's rule of combination is provided to calculate a combined probability mass from
different evidences. The fusion rules are given in equation (1) and (2).
$$m(C) = \frac{\sum_{B_1 \cap B_2 \ldots \cap B_n = C} \prod_{1 \leq i \leq n} m_i(B_j)}{1 - k} \tag{1}$$

$$k = \sum_{B_1 \cap B_2 \ldots \cap B_n = \emptyset} \prod_{1 \leq i \leq n} m_i(B_j) \tag{2}$$
Where $k$ represents the basic probability mass associated with conflicts among the sources of evidence.
$C$ is the intersection of all classes $B_j$ and carries the joint information from all the input data. After the
combination, we took a decision rule to decide the class we finally accept. There are several ways to
decide the final class by simply choosing the class with the maximum belief, plausibility, support, or
commonality.
**3.2.2 Mapping based on DSET**
Here, we presented our implementation for the GLC-2015 mapping in the framework of DSET. All the
GLC products and national-scale products described in Sect. 2 were selected as input maps to be
combined. In the integration of multi-source products, since all the LC classes in our classification system
are known, the frame of discrimination was defined to be our classification system:
$$\Omega = \left\{ \begin{array}{l} \text{cropland, forest, grassland, shrubland, wetland, water bodies,} \\ \text{tundra, impervious surfaces, bare land, permanent snow and ice} \end{array} \right\} \quad (3)$$

The definition of BPA function is the critical point in applying DSET (Rottensteiner et al., 2005).
In the fusion, we wanted to achieve a per-pixel classification into one of ten LC classes: cropland, forest,
grassland, shrubland, wetland, water bodies, tundra, impervious surfaces, bare land, and permanent snow
and ice. For each product, the accuracy for each LC class was calculated and used as evidential
probability to construct the BPA. Given that the local accuracy for a 4°×4° grid was not able to adequately
reflect the actual land cover landscape, especially for the rare LC classes, the global accuracy was
incorporated into the construction of the BPA to avoid uncertainties from a local point of view. Since the
assessment based on local samples plays a more critical role in BPA construction for a local grid, a higher
weight should be assigned to the local accuracy. To identify the best weight, we tested different weights
of the local accuracy (see Fig. S1). The result shows that using 75% performed robustly and obtained
relatively higher overall accuracy. Therefore, we chose 75% as the weight for local accuracy and 25%
for global accuracy. Here, we defined the BPA function as follow:
$$m_i(T_j) = \frac{PA_{local_{(ij)}} + UA_{local_{(ij)}}}{2} \times 75\% + \frac{PA_{global_{(ij)}} + UA_{global_{(ij)}}}{2} \times 25\% \quad (4)$$

Where $m_i(T_j)$ represents the BPA function of evidence source $i$ for the LC class $T_j$ ; $PA_{local_{(ij)}}$,
$UA_{local_{(ij)}}$ denote producer's accuracy and user's accuracy of evidence source $i$ for the LC class $T_j$ for
each  4° × 4° geographical grid, respectively; $PA_{global_{(ij)}}$, $UA_{global_{(ij)}}$ denote producer's accuracy and
user's accuracy of evidence source $i$ for LC class $T_j$ at the global scale.
To estimate the exact values of $PA_{local_{(ij)}}$, $UA_{local_{(ij)}}$, $PA_{global_{(ij)}}$ and $UA_{global_{(ij)}}$, we used 80%
of the global point-based samples more than 160,000 points derived in Sect 2.3. As soon as we obtained
the measurements of $m_i(T_j)$, the combined probability masses $m(T_j)$ were evaluated based on
Dempster's rule of combination for each pixel classified as the LC class $T_j$ by fusing BPA values of all
the evidence sources:
$$m(T_j) = \frac{1}{1-k} \sum_{T_{1j} \cap T_{2j}\dots \cap T_{nj}=T_j} m_i(T_j) \tag{5}$$
$$k = \sum_{T_{1j} \cap T_{2j}\dots \cap T_{nj}=\emptyset} m_i(T_j) \tag{6}$$
Where $k$ represents the basic probability mass associated with conflict; $m_i(T_j)$ represents the basic
probability mass of a certain pixel belonging to the LC class $T_j$ from different GLC products.
Additionally, a belief measure (Bel) was given to measure the degree of credibility that a pixel
labeled as the finally accepted LC class when combining all the available evidences. The belief measure
was determined by
$$Bel(T_j) = \sum_{T_{ij} \subseteq T_j} m_i(T_j) \tag{7}$$
To determine the finally accepted LC class per pixel, we took the rule of maximum combined
probability mass as our decision rule and the LC class with the maximum combined probability mass is
assigned to the 30 m pixel. Pixels labeled with the LC class were integrated to generate the GLC-2015
product.
**3.3 Accuracy assessment**
To assess the accuracy of the GLC-2015 map, we utilized two validation methods: validation with the
global point-based samples and the global patch-based samples. Since the global point-based sample set
is distributed evenly across the world and its sample size for each LC class is relatively sufficient and
balanced, even for the rare classes, it can provide a representative and credible basis for estimation of the
GLC-2015 map globally. Furthermore, we used the global patch-based samples to conduct accuracy
assessment from the local landscape scale. Although the global patch-based sample set provide an
inadequate sample size for rare LC classes, it can take advantage of the spatial context information and
efficiently reflect the actual pattern of the landscape.
The confusion matrix was produced to evaluate and analyze the GLC-2015 mapping result. The
error matrix is composed of entry $A_{ij}$, which represents the number of samples with reference LC class

$j$ being classified as LC class $i$. The overall accuracy (OA), kappa coefficient, producer's accuracy (PA), and user's accuracy (UA) were generated from confusion matrix to describe the quality of the GLC-2015 map. They are defined as follows:

$$OA = \frac{\sum_i A_{ii}}{\sum_i \sum_j A_{ij}} \tag{8}$$

$$P_o = OA \tag{9}$$

$$P_e = \sum_k \frac{\sum_i A_{ik}}{\sum_i \sum_j A_{ij}} \times \frac{\sum_j A_{kj}}{\sum_i \sum_j A_{ij}} \tag{10}$$

$$kappa = \frac{P_o - P_e}{1 - P_e} \tag{11}$$

$$PA^i = \frac{A_{ii}}{\sum_k A_{ki}} \tag{12}$$

$$UA^i = \frac{A_{ii}}{\sum_k A_{ik}} \tag{13}$$

Where $UA^i$ and $PA^i$ represent UA and PA of the LC $i$, respectively; $P_o$ is the agreement between the reference and the classified data; $P_e$ is the hypothetical probability of chance agreement.

**3.4 Data inter-comparison**

To better reflect the quality of the GLC-2015 map, we inter-compared the GLC-2015 map with the existing products at multiple scales. In the accuracy assessment of different products, two global validation sets described earlier were employed.

To figure out whether the GLC-2015 map promotes accuracy in the areas with high classification difficulty and how much the improvement is compared to the other GLC products, we conducted the spatial consistency analysis between the GlobeLand30, FROM_GLC, and GLC_FCS30 and compared the mapping performance of the GLC-2015 with others in the areas of low inconsistency, moderate inconsistency, and high inconsistency. To visually present the spatial consistency between three existing GLC maps, we employed the spatial superposition method to obtain the spatial correspondence pixel-by-pixel between different maps. Based on the times of all the GLC products agreed for the same LC class, the degree of consistency for a pixel was identified as three levels with the agreement value equal to 3, 2, or 1. The areas of low inconsistency were regarded as pixels that classified as the same LC class in all three GLC maps (labeled as 3). The moderate inconsistency areas were regarded as pixels that were consistent in only two GLC maps (labeled as 2). The high inconsistency areas were regarded as pixels that were totally inconsistent in these three GLC maps (labeled as 1). For a visual comparison, all these

GLC maps were aggregated to 0.05°, in which the LC class with the largest proportion determined the
class in each 0.05° grid.
**3.5 Assessment on mapping performance of DSET and other methods**
In addition to inter-comparison between the GLC-2015 map and the existing products, we compared the
DSET method with two existing commonly used fusion methods, including the majority voting (MV)
and spatial correspondence (SC) based on two global validation sets including 20% of the global point-
based samples and the whole global patch-based samples. MV is a fusion approach that combines input
maps and adopts the LC class favored by the majority of the candidate maps. In the MV method, we
compared the GlobeLand30, FROM_GLC, and GLC_FCS30 at each pixel and chose the class that two
or three LC products agreed for. For pixels where three LC products were different, the LC class of the
product with the highest accuracy was adopted. SC method produces an integrated land cover map by
selecting the LC class of the input map that has the highest spatial correspondence with the reference
data. In this study, 80% of the global point-based samples were used as the reference data to obtain the
SC map of each global LC product. If the class of a product agreed with that of the point-based sample,
a value equal to 1 was assigned to that sample. On the contrary, a value equal to 0 was assigned to the
sample if the class of the product differed from that of the sample. In each 4° × 4° grid, we used the
Kriging method to obtain spatial correspondence maps which have the correspondence value ranging
from 0 to 1 for three products. Then, the class of the product with the highest spatial correspondence was
chosen for each pixel.
Furthermore, we compared the mapping performance of DSET with Random Forest (RF) which is
considered one of the most popular algorithms for land cover mapping. In the land cover classification
using the FR classifier, all available Level-2 Tier 1surface reflectance (SR) data of Landsat 8 OLI
(Operational Land Imager) sensors from the year 2015 and two adjacent years on GEE was employed.
All Landsat images have been atmospherically corrected. The following six bands were used as input
features: blue, green, red, NIR, SWIR1, and SWIR2. To improve the mapping performance, several
important spectral indices, including DNVI, NDWI, and NDBI were also used as auxiliary data to the
RF classifier. The RF classifier was trained on 80% of the global point-based samples since those samples
were of high quality after manual visual interpretation of high-resolution images. As the global land cover
mapping based on the RF classifier is a tough task, we randomly selected a total of 300 grids with the
size of 4° (Fig. S2) and applied corresponding local RF classifiers to these grids. Then, the mapping
results were validated by the remaining 20% of the point-based samples.

## 4.    Results and discussion

### 4.1 Mapping result of the GLC-2015 map

Using a multi-source product fusion method based on the DSET, we generated an improved 30m global
land cover map in 2015 (GLC-2015). Figure 5 illustrates the GLC-2015 map. The GLC-2015 map can
accurately describe the spatial distribution of various LC classes. For example, cropland areas are mostly
located in Central America, the region from the Hungarian plain to the Siberian plain, the eastern and
southern parts of China, and the most of India. In addition, forest, which is one of the easily
distinguishable classes from the map, is concentrated in the eastern part of North America, the Amazon
basin of South America, the northern part of Eurasia and the equatorial region of Africa.

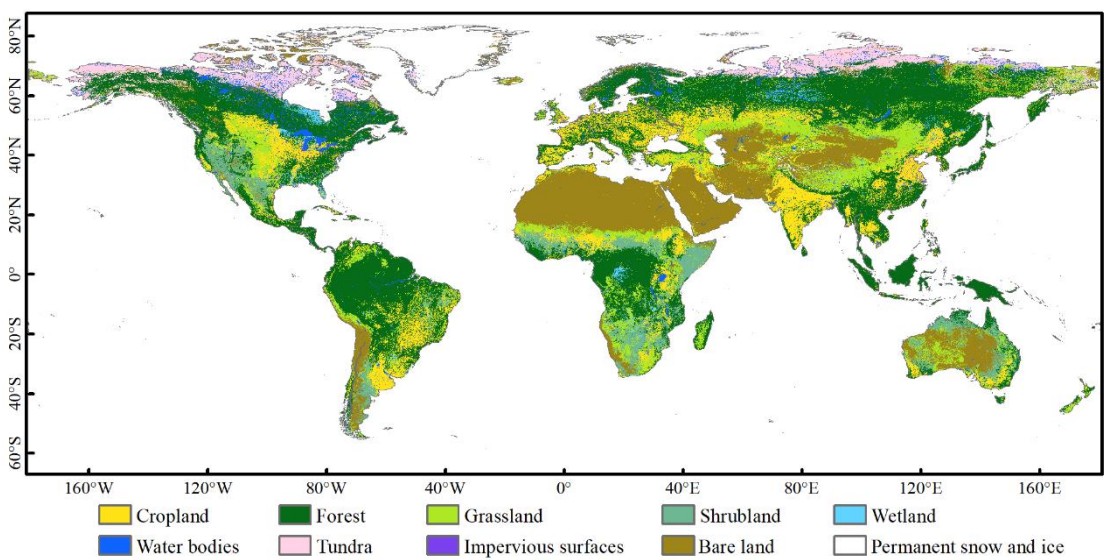

**Figure 5. Global land cover map in 2015 with 30 m resolution (GLC-2015).**

### 4.2 Accuracy assessment of the GLC-2015 map

#### 4.2.1 Accuracy assessment with the global point-based samples

The accuracy of the GLC-2015 map was first tested via the global point-based samples, and the results
of assessment are listed in Table 3. The GLC-2015 map achieved an OA of 79.5% and kappa coefficient
of 0.757 at the global scale, demonstrating the good performance of our map. Among all the LC classes,
permanent snow and ice possessed the best mapping performance, with PA and UA achieving 89.1% and

 93.7%. The accuracy of water bodies, forest and impervious surfaces was also high, where PA and UA

exceeded 80.0%. Grassland, shrubland, and wetland had relatively low accuracy, with PA below 75.0%.
Among them, grassland and shrubland were mainly confused with forest, which might be because these
classes are both vegetation, thus causing difficulty in recognition by spectral information. Due to the
complex spectral characteristics, wetland is often mixed with vegetation (Ludwig et al., 2019).
**Table 3. The confusion matrix for the GLC-2015 map based on the global point-based samples.**

| | Cropland | Forest | Grassland | Shrubland | Wetland | Water bodies | Tundra | Impervious surfaces | Bare land | Permanent snow and ice | Total | PA |
|---|---|---|---|---|---|---|---|---|---|---|---|---|
| Cropland | 3623 | 387 | 356 | 61 | 27 | 48 | 2 | 71 | 81 | 0 | 4656 | 0.778 |
| Forest | 155 | 8813 | 186 | 141 | 232 | 16 | 43 | 43 | 53 | 3 | 9685 | 0.910 |
| Grassland | 10 | 337 | 1920 | 19 | 24 | 13 | 47 | 36 | 184 | 9 | 2599 | 0.739 |
| Shrubland | 155 | 438 | 656 | 1469 | 39 | 29 | 70 | 78 | 442 | 4 | 3380 | 0.435 |
| Wetland | 47 | 287 | 82 | 14 | 1067 | 64 | 22 | 18 | 110 | 4 | 1715 | 0.622 |
| Water bodies | 27 | 90 | 15 | 1 | 73 | 1936 | 17 | 10 | 44 | 3 | 2216 | 0.874 |
| Tundra | 1 | 242 | 119 | 6 | 29 | 19 | 1411 | 2 | 269 | 17 | 2115 | 0.667 |
| Impervious surfaces | 74 | 41 | 11 | 3 | 8 | 11 | 1 | 1295 | 45 | 0 | 1489 | 0.870 |
| Bare land | 36 | 59 | 237 | 32 | 44 | 91 | 55 | 60 | 4909 | 38 | 5561 | 0.883 |
| Permanent snow and ice | 0 | 11 | 8 | 0 | 4 | 18 | 13 | 1 | 86 | 1154 | 1295 | 0.891 |
| Total | 4128 | 10705 | 3590 | 1746 | 1547 | 2245 | 1681 | 1614 | 6223 | 1232 | 34711 | |
| UA | 0.878 | 0.823 | 0.535 | 0.841 | 0.690 | 0.862 | 0.839 | 0.802 | 0.789 | 0.937 | | |
| OA | | | | | | 0.795 | | | | | | |
| Kappa | | | | | | 0.757 | | | | | | |

The regional accuracies are presented in Fig. 6. The OA of the GLC-2015 ranged from 66.4% to
93.4%, and kappa coefficient from 0.552 to 0.813. From the perspective of OA, Water regions lead,
followed by Tropical desert, Temperate continental forest, and Polar. These are areas with homogeneous
land cover and have low difficulty in mapping. Boreal tundra woodland, Tropical dry forest, Tropical
shrubland, and Subtropical desert are the regions with low OA. The first one may be related to the high
latitudes. The followed two may be because they belong to areas with complicated and mixed LC classes
which is not easily classified. The last one may be the consequence of sparse vegetation in desert areas.
For the kappa coefficient, the ranking was similar with those for OA.

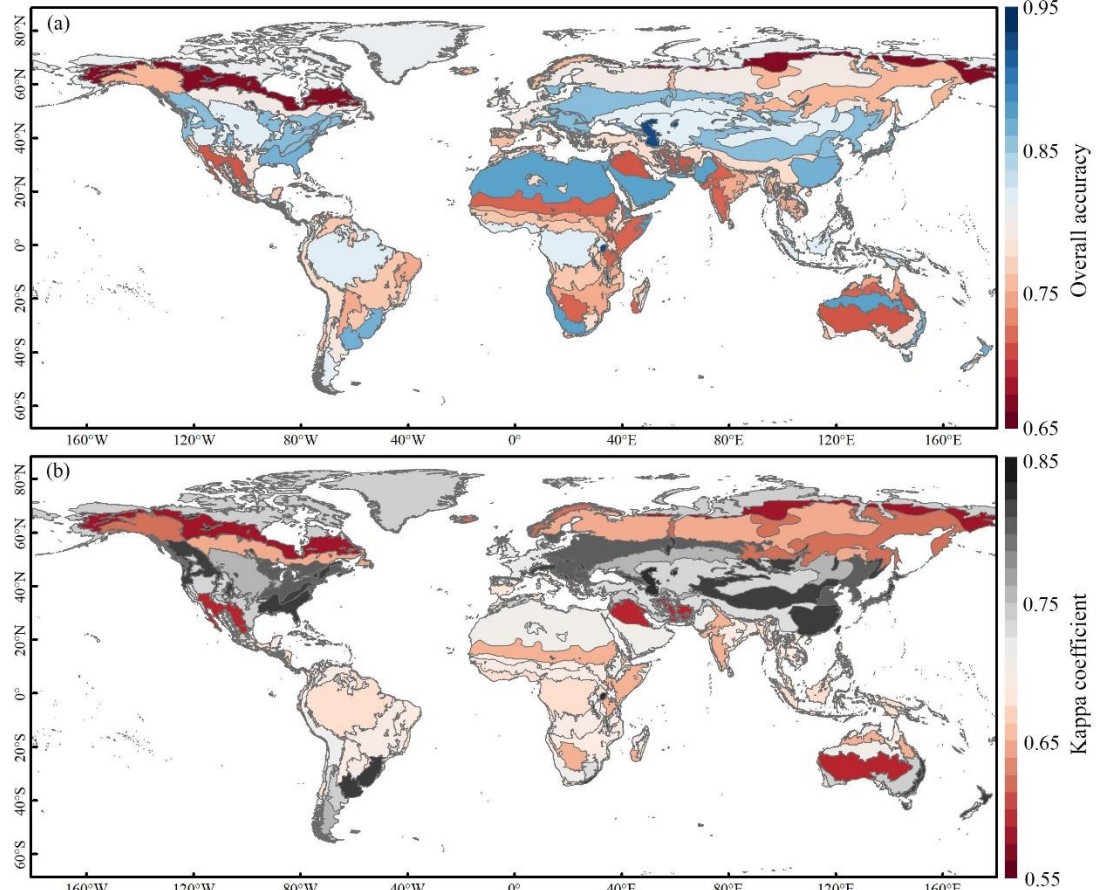

**Figure 6. Regional accuracy of the GLC-2015 map according to ecoregions. (a)overall accuracy, (b) kappa**
**coefficient. The ecoregion boundaries are obtained from the Food and Agriculture Organization of the United**
**Nations (FAO).**
**4.2.2 Accuracy assessment with the global patch-based samples**
The accuracy assessment of the GLC-2015 map was also conducted with the global patch-based samples.
Table 4 summarizes the results for accuracy assessment of each LC class in the GLC-2015 map. From
the assessment results, it can be found that the OA of the GLC-2015 map reached 83.6%, which was
higher than 79.5% tested with the global point-based samples. The kappa coefficient of the GLC-2015
map was 0.566, which was 0.191 lower than the result calculated with the global point-based samples.
In both accuracy assessment results based on two different validation data sets, water bodies, forest, and
permanent snow and ice were validated to have high accuracy, and grassland, shrubland, and wetland
were validated to have low accuracy. Nevertheless, the ranking of accuracy for each LC class had a slight
difference. For example, in assessment based on the global point-based samples, impervious surfaces
and permanent snow and ice ranked higher than that based on the global patch-based samples. This may
be because a LC map can easily show where one LC class is distributed but hardly describe its actual
shape. In addition to the accuracy assessment on a pixel scale, validation on a patch scale is equally
important because it can reflect the shape consistency between the GLC-2015 map and the actual
landscape, even if the size of global patch-based samples is relatively small. Overall, no matter from the
respective of the global point-based samples or the global patch-based samples, the mapping accuracies
of the GLC-2015 map are satisfactory.
**Table 4. Mapping accuracy via the global patch-based samples for the GLC-2015 map**

| | Cropland | Forest | Grassland | shrubland | Wetland | Water bodies | Tundra | Impervious surfaces | Bare land | Permanent snow and ice |
|---|---|---|---|---|---|---|---|---|---|---|
| PA | 0.887 | 0.895 | 0.629 | 0.589 | 0.301 | 0.939 | 0.701 | 0.757 | 0.682 | 0.825 |
| UA | 0.916 | 0.844 | 0.617 | 0.714 | 0.511 | 0.917 | 0.872 | 0.713 | 0.599 | 0.767 |
| OA | | | | | | 0.836 | | | | |
| Kappa | | | | | | 0.566 | | | | |

**4.3 Inter-comparison with existing GLC products**
**4.3.1 Inter-comparison based on the global point-based samples**
Based on the global point-based samples, the inter-comparison of the GLC-2015 map with the
GlobeLand30, FROM_GLC, and GLC_FCS30 were conducted. The accuracy assessment results for all
GLC maps are listed in Table 5. It can be found that the GLC-2015 map achieved the highest OA of 79.5%
compared with GlobeLand30 of 65.3%, FROM_GLC of 61.7%, and GLC_FCS30 of 65.5%, respectively.
The accuracy gap between the GLC-2015 map and other existing ones was 14.0%-17.8%. Also, the GLC-
2015 map possessed a better kappa coefficient than other products. For all classes except tundra, the
GLC-2015 map outperformed the other three maps in terms of PA. For cropland, grassland, shrubland,
wetland, and tundra, the GLC-2015 map also exhibited better performance regarding UA than the
GlobeLand30, FROM_GLC, and GLC_FCS30. Overall, for the PA or UA, the GLC-2015 map ranked
first or second in nearly all LC classes, which demonstrated that the GLC-2015 map had smaller omission
and commission errors against the other three products.
**Table 5. Mapping accuracy of the GLC products with the global point-based samples.**

| | Cropland | Forest | Grassland | Shrubland | Wetland | Water bodies | Tundra | Impervious surfaces | Bare land | Permanent snow and ice | OA (Kappa coefficient) |
|---|---|---|---|---|---|---|---|---|---|---|---|

| | | | | | | | | | | | | |
|---|---|---|---|---|---|---|---|---|---|---|---|---|
| GLC-2015 | PA | 0.778 | 0.910 | 0.739 | 0.435 | 0.622 | 0.874 | 0.667 | 0.870 | 0.883 | 0.891 | 0.795 |
| | UA | 0.878 | 0.823 | 0.535 | 0.841 | 0.690 | 0.862 | 0.839 | 0.802 | 0.789 | 0.937 | (0.757) |
| Globeland30 | PA | 0.752 | 0.719 | 0.713 | 0.245 | 0.540 | 0.680 | 0.769 | 0.688 | 0.609 | 0.821 | 0.653 |
| | UA | 0.786 | 0.818 | 0.255 | 0.428 | 0.573 | 0.869 | 0.577 | 0.809 | 0.868 | 0.905 | (0.598) |
| FROM_GLC | PA | 0.389 | 0.694 | 0.707 | 0.411 | 0.307 | 0.607 | 0.712 | 0.732 | 0.731 | 0.881 | 0.617 |
| | UA | 0.671 | 0.859 | 0.278 | 0.422 | 0.289 | 0.742 | 0.686 | 0.661 | 0.761 | 0.773 | (0.558) |
| GLC_FCS30 | PA | 0.757 | 0.775 | 0.452 | 0.399 | 0.455 | 0.604 | 0.228 | 0.777 | 0.809 | 0.726 | 0.655 |
| | UA | 0.616 | 0.816 | 0.384 | 0.405 | 0.515 | 0.808 | 0.688 | 0.774 | 0.645 | 0.947 | (0.591) |

Further quantitative accuracy assessments of different GLC products were performed in 4° × 4°
grids using the global point-based samples, and box plots were produced for each product for all grids
within different ecoregions, as shown in Fig. 7. It can be found that the GLC-2015 map outperformed
other existing products with the best OA and kappa coefficient across different ecoregions. Also, the
mean overall accuracy of the GLC-2015 map exceeded 65.0% in all ecoregions, showing the high quality
of our mapping results. It is worth noting that the GLC-2015 map showed shorter boxes except in
Subtropical dry forest and Subtropical desert, which means the GLC-2015 map had relatively small
fluctuation than other ones. In Subtropical desert, Tropical dry forest, and Boreal tundra woodland, the
OA and kappa coefficient of the four products were relatively low. However, the GLC-2015 map
exceeded the highest of others and greatly improved the mean OA in these regions.

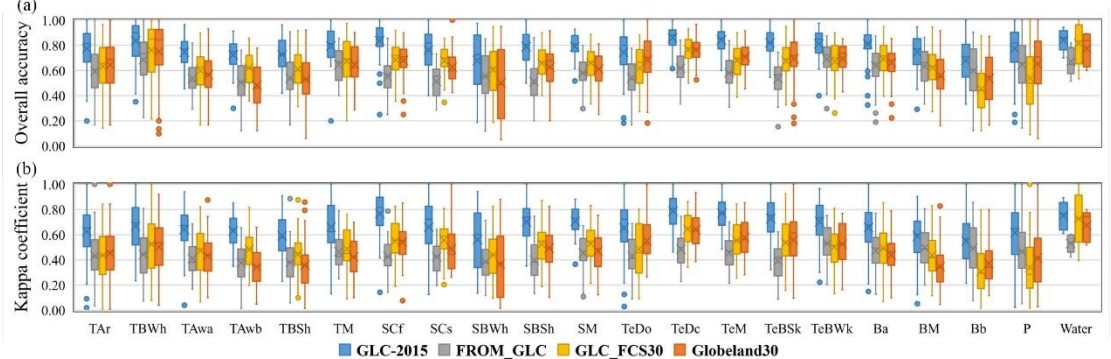


**Figure 7. The box-plot of the accuracy for twenty-one ecoregion zones. (a) overall accuracy, (b)kappa**
**coefficient. Ecoregion abbreviation and corresponding ecoregion is described in Table S4.**
**4.3.2 Inter-comparison based on the global patch-based samples**
Although the global point-based samples are adequate and even across the globe, the distribution of
points in each 4° × 4° geographical grid is too sparse to reflect the actual spatial pattern of the landscape.
Focusing on LC pattern at the local scale, we also used the global patch-based samples which can provide
spatial context information to conduct the accuracy assessment of the GLC-2015 map and compare
difference GLC products. Table 6 lists the accuracies of the GLC-2015 map and the other three GLC
products. Obviously, the GLC-2015 map achieved the best OA and kappa coefficient among these four
GLC maps. The overall accuracy gap between the GLC-2015 product and others was 5.9%-24.5%, which
presented a more significant variation compared with the result based on the global point-based samples.
In terms of PA and UA, the GLC-2015 map was higher than the other three ones in most LC classes.
Specifically, all the products had lower accuracy for grassland, shrubland, and wetland, similar to that in
the accuracy assessment based on the global point-based samples. It is evident that the FROM_GLC had
the lowest mapping accuracy for grassland, shrubland, and wetland, implying that the classification
method of FROM_GLC is not robust for these three LC classes.
**Table 6. Mapping accuracy of the GLC products with the global patch-based samples**

|  |  | Cropland | Forest | Grassland | Shrubland | Wetland | Water bodies | Tundra | Impervious surfaces | Bare land | Permanent snow and ice | OA |
|---|---|---|---|---|---|---|---|---|---|---|---|---|
| GLC-2015 | PA | 0.887 | 0.895 | 0.629 | 0.589 | 0.301 | 0.939 | 0.701 | 0.757 | 0.682 | 0.825 | 0.836 |
|  | UA | 0.916 | 0.844 | 0.617 | 0.714 | 0.511 | 0.917 | 0.872 | 0.713 | 0.599 | 0.767 | (0.566) |
| Globeland30 | PA | 0.896 | 0.698 | 0.765 | 0.539 | 0.455 | 0.824 | 0.752 | 0.643 | 0.492 | 0.831 | 0.777 |
|  | UA | 0.891 | 0.906 | 0.444 | 0.527 | 0.157 | 0.893 | 0.500 | 0.703 | 0.829 | 0.705 | (0.437) |
| FROM_GLC | PA | 0.485 | 0.714 | 0.640 | 0.254 | 0.032 | 0.904 | 0.760 | 0.506 | 0.681 | 0.501 | 0.591 |
|  | UA | 0.872 | 0.809 | 0.193 | 0.139 | 0.186 | 0.884 | 0.696 | 0.808 | 0.496 | 0.703 | (0.360) |
| GLC_FCS30 | PA | 0.865 | 0.779 | 0.398 | 0.565 | 0.363 | 0.869 | 0.051 | 0.648 | 0.658 | 0.742 | 0.748 |
|  | UA | 0.857 | 0.832 | 0.509 | 0.330 | 0.132 | 0.942 | 0.573 | 0.643 | 0.462 | 0.752 | (0.418) |

Accuracy assessment was calculated in each patch-based sample, and box plots were produced for
each GLC product at the continental scale, as shown in Fig. 8. The GLC-2015 map showed a robust
performance in each continent, with the highest OA and kappa coefficient among all the maps. Also, in
all continents, the GLC-2015 map had the shortest boxes in terms of OA, which denoted that it had a
more minor variation in accuracy at the continental scale. Among four products, the GLC_FCS30 and
Globeland30 achieved similar accuracies in most continents. Obviously, the FROM_GLC showed lowest
accuracy across different continents, especially in Oceania, where the OA of most patch-based samples
was below 40.0%, namely most of the pixels in Oceania were incorrectly classified. We further compared
mapping accuracies for each LC class in different continents (Figs. S3 and S4). Since tundra and
permanent snow and ice are rare and only existent in certain regions, they were not included in the
comparison. As for PA across different continents, the GLC-2015 map outperformed other maps in forest,
water bodies, and bare land. As for UA across different continents, the GLC-2015 map outperformed
other maps in cropland, grassland, shrubland and wetland, and achieved similar accuracies with the
GLC_FCS30 and Globeland30 in forest. Overall, the GLC-2015 map outperformed others regarding
mapping accuracy at continental scale. In addition, all GLC products showed significant variation and
low mean accuracy in grassland, shrubland, and wetland over most continents.

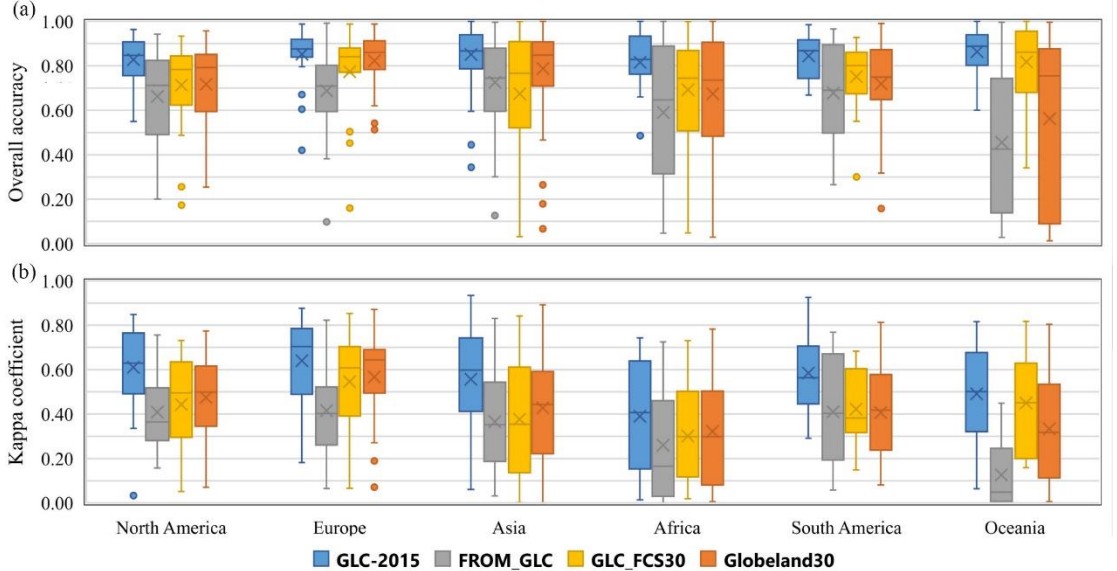


**Figure 8. The box-plot of the accuracy for different continents. (a) overall accuracy, (b)kappa coefficient.**

Furthermore, to compare the OA of the GLC-2015 map with other GLC products, scatter plots were

used to describe the relationship between the overall accuracy of the GLC-2015 map and one other
product in each patch-based sample, as displayed in Fig. 9. Most of the points were above the 1:1 line,
implying that the GLC-2015 map surpassed other GLC products in terms of OA. The distribution of
points was more dispersed from the 1:1 line in the plot of the GLC-2015 map against FROM_GLC
compared to other plots. It indicated that these two products had a more significant difference, which
was also proved in Table 6.

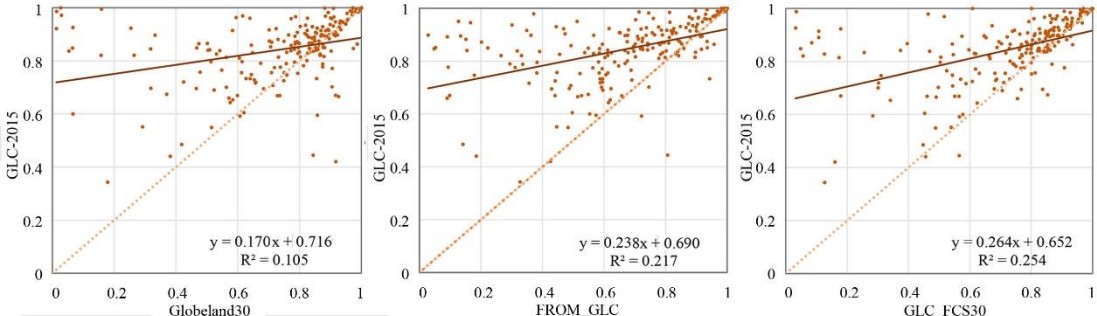


**Figure 9. Scatter plots between the GLC-2015 map and other products obtained using the global patch-based**
**samples.**
**4.3.3 Areal comparison for individual classes**
To assess the similarities and discrepancies between the GLC-2015 and other GLC products, we
compared the area of various LC classes at multiple scales, including global, continental, national, and
ecoregional scales.
The areal comparison for various classes of different GLC products over the globe is shown in Fig.
10. Generally, the areas of water bodies and permanent snow and ice of four GLC products were very
similar, which may be related to the similar LC definitions. In contrast, the areas of cropland, forest,
grassland, and shrubland in GLC-2015 differed significantly from those in other GLC products. The area
of forest in GLC-2015 is much higher than other products. This may be because FROM_GLC and
GLC_FCS30 defined forest with tree cover over 15%, while GLC-2015 used a threshold of over 10%.
The cropland areas in GLC-2015 and Globeland30 were close, higher than FROM_GLC but lower than
GLC_FCS30. Moreover, the FROM_GLC underestimated the cropland area as it had a low producer's
accuracy for cropland (see Table 5), which was also demonstrated in previous researches (Liu and Xu,
2021; Zhang et al., 2021). FROM_GLC and Globeland30 shared similar grassland areas since a similar
accuracy for grassland was found in these two products (see Table 5). However, the FROM_GLC and
Globeland30 significantly overestimated grassland extent, with much bare land misclassified as
grassland (Hu et al., 2014). The GLC_FCS30 showed the smallest area for grassland, which might be
related to its higher threshold in vegetation cover for grassland. For shrubland, the area difference
between GLC-2015 and Globeland30 was minimal, and the areas in FROM_GLC and GLC_FCS30 were
similar. Furthermore, the wetland area in FROM_GLC was the lowest among all the products, with a
total area of 0.168 million $km^2$. In contrast, the Globeland30 and GLC_FCS30 exhibited greater wetland
extent than GLC-2015 since these two products classified non-wetlands sensitive to water as wetlands
(Zhang et al., 2023). In particular, the tundra area in GLC_FCS30 was much smaller than other products.
This is mainly because only lichens/mosses in the original classification system of GLC_FCS30 was
converted into tundra in the classification system we used, which leads to the omission of tundra. The
areas of impervious surfaces in GLC-2015, Globeland30, and GLC_FCS30 were very close and higher
than FROM_GLC. For bare land, there was large difference between Globeland30 and other products,
while the area in GLC-2015 and GLC_FCS30 was very close.

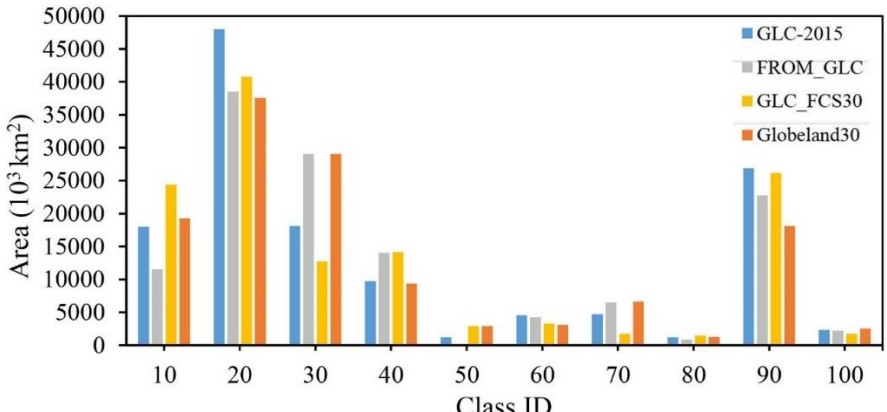


**Figure 10. Areal comparison of various land cover classes among GLC products at the global scale. Class IDs**
**10, 20, 30, 40, 50, 60, 70, 80, 90, and 100 denote cropland, forest, grassland, shrubland, wetland, water bodies,**
**tundra, impervious surfaces, bare land, and permanent snow and sea ice, respectively.**
The area similarity and difference for various classes of different GLC products were also compared
over six continents, the top 40 countries ranked by area, and 21 ecoregions (Figs. S5- S7). Overall, the
four products showed a similar distribution trend of different classes. For most LC classes, the continental,
national, and ecoregional rankings of four products agreed with their ranking at the global scale. Whereas,
for grassland and shrubland, the area ranking of four products varied at three different regional scales.
**4.3.4 Visual inter-comparison for individual classes**
The visual comparison of cropland in GLC-2015, Globeland30, FROM_GLC, GLC_FCS30, Global
Food Security-Support Analysis Data (GSFAD30) (Xiong et al., 2017; Teluguntla et al., 2018), and other
national-scale maps was conducted in three local regions (Fig. S8). In the Egyptian agricultural area,
GLC-2015, FROM_GLC, and GLC-FCS30 shared similar delineation of the cropland and had a good
representation of cropland with fine spatial details. Since the date time of the Google Earth image is 2015,
Globeland30 missed the newly cultivated cropland. GFASD30 had the largest cropland area among five
products but misclassified bare land as cropland. In the agricultural area of Southeastern China, GLC-
2015 had an agreement with GFSAD30 and CLCD. Globeland30 and GLC_FCS30 overestimated the
area of cropland. As for FROM_GLC, it failed to depict the spatial distribution of cropland and had many
omissions. In cropland-dominated areas of the United States, FROM_GLC significantly underestimated
the extent of cropland. The other five products exhibited a similar delineation of cropland, but there were
little differences in some small areas. For example, Globeland30 misclassified some grassland into
cropland, and NLCD 2016 had a good ability to distinguish the farm rack.
We also compared the performance in the forest of different products in three forest-prevalent
regions of Congo, China, and the United States (Fig. S9). Overall, GLC-2015 and Globeland30 showed
accurate delineation in three regions. FROM_GLC also had good performance for the forest in Congo
and USA but overestimated the forest in China, mislabeling shrubland and grassland as forest.
Furthermore, GFC tended to miss sparse trees in China, and GLC_FCS30 underestimated the extent of
forest in both three regions. As for national-scale products, CLCD and NLCD 2016 had a good ability to
identify the details of forest, while CLUD dramatically missed both dense and sparse woodlands.
Furthermore, to compare the performance in the wetland of GLC-2015 with other global and
national-scale products, three wetland regions in South-central Canada, coastal America, and Sundarbans
were selected. It can be found that GLC-2015 and Globeland30 had similar representation and performed
well in identifying the wetland over three regions (Fig. S10). Unexpectedly, FROM_GLC performed
poorly in each region, with almost no wetlands captured. GLC_FCS30 also showed unstable quality in
three regions. For example, it highly underestimated the wetland area in coastal America and completely
mislabeled the mangroves as cropland in Sundarbans. NLCD 2016 and GMW accurately demonstrated
the spatial pattern of the wetland, while the CA_wetlands map underestimated the wetland extent because
it defined wetlands by wetland frequency of no less than 80% from 2000 to 2016 (Wulder et al., 2018).
To understand the spatial distribution of impervious surfaces in different products, a comparison of
mapping results for three megacities, including Tokyo, Shanghai, and New York, was shown in Fig. S11.
In Tokyo, a high consistency was found between GLC-2015, FROM_GLC, and GAUD, and both
successfully captured the impervious surfaces in peri-urban areas. GLC_FCS30 showed the largest area
for impervious surfaces because it misclassified many croplands into impervious surfaces. In Shanghai,
GLC_FCS30 underestimated the central city, and CLUD lost the details of impervious surfaces because
it was developed using the visual interpretation method. Other products generally had the similar
representation and accurately demonstrated the spatial distribution of the city. For New York, the
FROM_GLC, GLC_FCS30, and GAUD agreed well with GLC-2015, while Globeland30 and NLCD
2016 had high impervious areas than others.

**4.3.5 Visual inter-comparison at the local scale**

We selected six typical geographical tiles covering six continents and different landscape environments
to further present the mapping performance of the GLC-2015 map, Globeland30, FROM_GLC, and
GLC_FCS30, as shown in Fig. 11. Overall, from a local point of view, the GLC-2015 map tended to be
more diverse in LC classes and had better identification performance in various classes. In flattened
cropland areas (Fig. 11a and b), the GLC-2015 map revealed diverse LC classes and accurately
distinguished impervious surfaces; however, the Globeland30 exaggerated the extent of impervious
surfaces, and the FROM_GLC failed to delineate impervious surfaces with small size. In addition, the
FROM_GLC misclassified some cropland pixels as grassland (Fig. 11a) and had an abnormal "stamp"
(Fig. 11b). As for mountain areas (Fig. 11c and d), the GLC-2015 map uncovered the spatial pattern of
natural and planted forest, cropland, and grassland. There were large confusions between cropland and
grassland in the results of the FROM_GLC and GLC_FCS30, and some impervious surfaces and
cropland areas were wrongly labeled as bare land by the FROM_GLC. The areas (Fig. 11c), which were
classified as forest, were misidentified as cropland and grassland in three other products. For the
rainforest areas where a large number of trees were reclaimed for cropland (Fig. 11e), the GLC-2015
map, Globeland30, and GLC_FCS30 had similarities in cropland areas; but the FROM_GLC recognized
some reclaimed areas as grassland. Additionally, the GLC-2015 map accurately presented the spatial
distribution of impervious surfaces while other products had omission or commission errors. In the
cropland-dominated areas (Fig. 11f), the GLC-2015 map and Globeland30 showed a higher agreement,
and both of them mapped the undulating areas as grassland. Unlike the aforementioned two products, the
FROM_GLC misclassified large tracts of croplands as grasslands, and the GLC_FCS30 did not capture
the grassland in undulating areas. Figure 11 also shows the belief measure of the fused result in different
geographical tiles. Although it does not directly evaluate the mapping accuracy, it serves as a degree of
support for the hypothesis of an accepted LC class being true, it can still reflect the quality of the GLC-
2015 map. Overall, Bel of the GLC-2015 map exceeded 80% in most areas of each tile, demonstrating
the credibility and high quality of our mapping result.

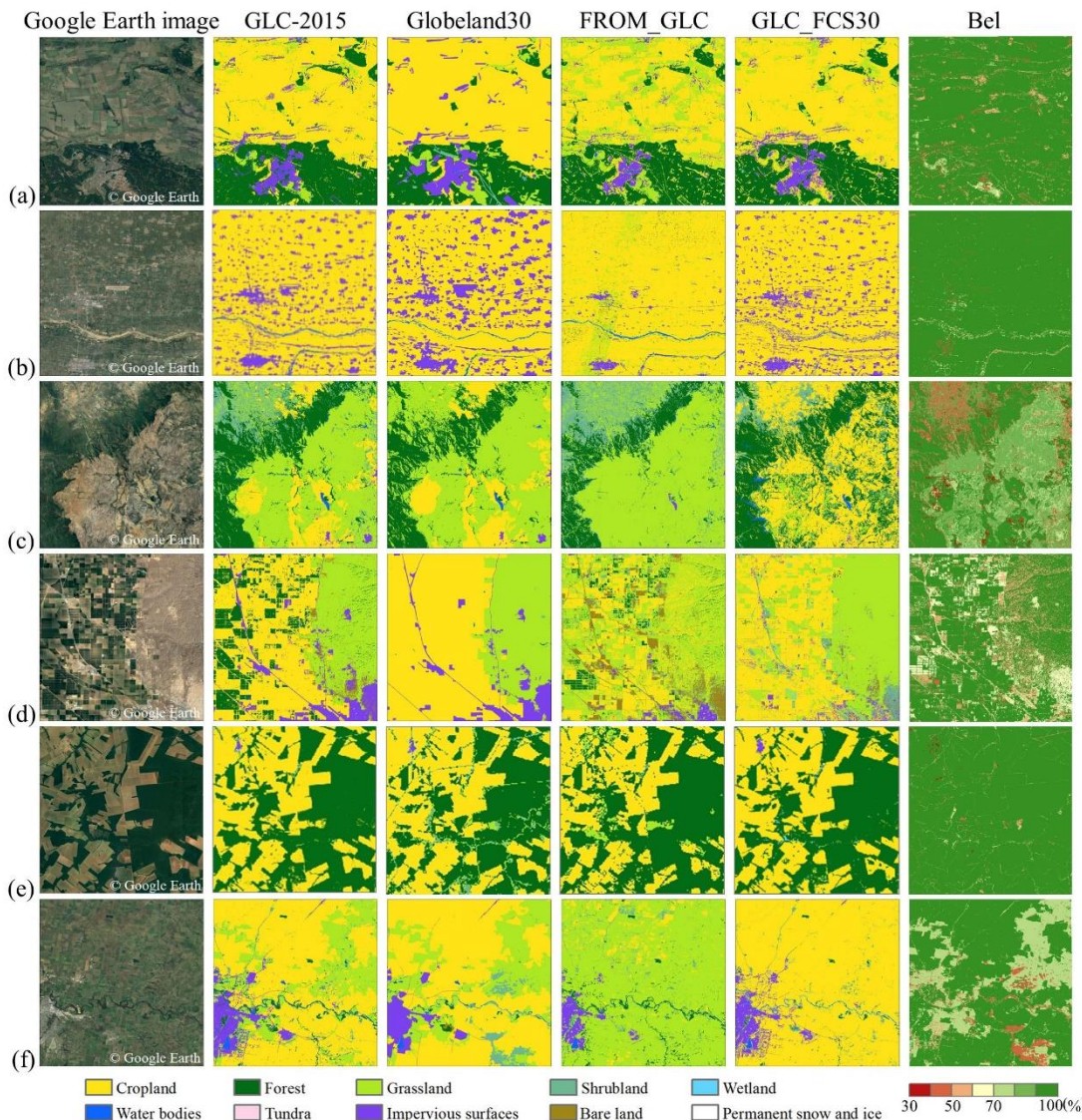

| | Google Earth image | GLC-2015 | Globeland30 | FROM_GLC | GLC_FCS30 | Bel |
| --- | --- | --- | --- | --- | --- | --- |

Cropland Forest Grassland Shrubland Wetland
Water bodies Tundra Impervious surfaces Bare land Permanent snow and ice
30 50 70 100(%)

**Figure 11. Visual comparison between the GLC-2015 map and three other products for different continents.**

**(a) to (f) are examples for Europe, Asia, Africa, North America, South America, and Oceania, respectively.**

**4.4 Inter-comparison with national-scale products**

Except for comparison with the existing GLC products, the GLC-2015 was also compared with three national-scale products (CLCD, CLUD, and NLCD 2016 over CONUS). We first compared the accuracy of the GLC-2015 with NLCD, CLCD, and CLUD using the point-based samples (Tables S5-S6). It can be found that the GLC-2015 obtained an overall accuracy of 88.8% in China, higher than CLCD (78.3%) and CLUD (70.2%). Specifically, the GLC-2015 achieved the highest PA and UA in all LC classes except wetland and impervious surfaces. In the CONUS, the GLC-2015 outperformed NCLD 2016 with an OA improvement of 13.2%. Additionally, the GLC-2015 exhibited better mapping performance in nearly all LC classes.

An accuracy comparison between the GLC-2015 and three national-scale products was also

performed using patch-based samples (Tables S7-S8). Overall, the GLC-2015 achieved a better OA of

85.7% in China, with respect to CLCD (83.6%) and CLUD (75.4%). In terms of PA and UA, the GLC-

2015 ranked first and second in most LC classes. In the CONUS, the GLC-2015 possessed an OA of

85.4% and a kappa coefficient of 0.787, outperforming NLCD 2016. Although the GLC-2015 had lower

PA or UA in cropland, forest, and impervious surfaces compared to NLCD 2016, the GLC-2015

outperformed NLCD 2016 in other LC classes.

We further performed an areal comparison for each LC class of GLC-2015 and three national-scale

products (Figs. S12 and S13). Generally, the GLC-2015, CLCD, and CLUD exhibited similar areas in

most classes. Notably, the areas of cropland, shrubland, and wetland in GLC-2015 were very close to

CLCD but different from CLUD. In the CONUS, the areas of cropland, water bodies, and bare land in

the GLC-2015 and NLCD 2016 were close. In contrast, the areas of the remaining LC classes in the

GLC-2015 showed a large difference from NLCD 2016. The area differences in forest, grassland and

shrubland between GLC-2015 and NLCD 2016 were mainly related to different LC definitions. For

example, the minimum fraction of tree cover in the forest is 10% in GLC-2015, whereas NLCD 2016

used a minimum fraction of 20%. NCLD 2016 had higher area of impervious surfaces than the GLC-

2015 because open urban in NLCD 2016 includes too much vegetation.

**4.5 Improvement of the GLC-2015 map compared to existing GLC products**

The spatial distribution of inconsistency between three GLC products at the global scale is illustrated in

Fig. 12. From the inconsistency map, we found that areas of low inconsistency mainly corresponded to

homogeneous regions with simple LC classes. For example, the northern part of Africa was mainly

classified as bare land, the northern part of South America was mainly classified as forest, and the

Greenland was classified as permanent snow and ice. On the contrary, areas of high inconsistency were

located in regions with complicated LC classes, especially in mixed vegetation regions or sparse

vegetation regions, such as northern Asia, South Africa, Sahel region, Australia, northern and southern

North America, and eastern and southern South America.

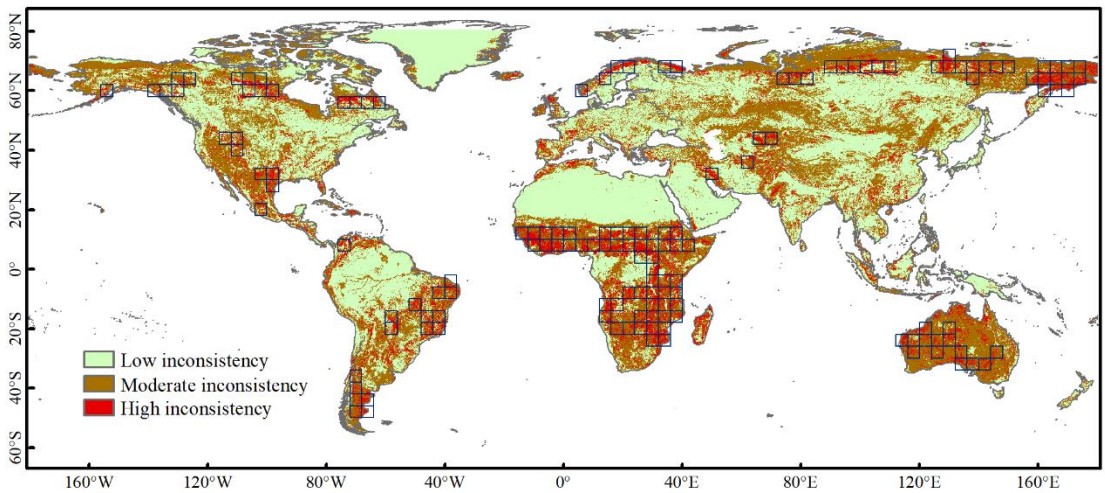

Figure 12. Distribution of inconsistency between the Globeland30, FROM_GLC, and GLC_FCS30.The blue
rectangles are high-inconsistency grids that the area of pixels with value equal to 1 account for more than 20%
of the total area.

Based on the global point-based samples, we assessed the accuracies of the GLC-2015 map,

Globeland30, FROM_GLC, and GLC_FCS30, in the aforementioned areas of low inconsistency,
moderate inconsistency, and high inconsistency, as shown in Table 7. Overall, the GLC-2015 map had
the highest accuracies against the other three ones in three areas. For each product, areas of low
inconsistency obtained the highest accuracies, followed by areas of moderate inconsistency and then high
inconsistency, which demonstrated that inconsistency of the existing products could indicate the quality
of maps. In areas of low inconsistency, the overall accuracy gap between the GLC-2015 map and
previous ones was as small as 0.1%-0.6%. However, for areas of moderate and high inconsistency, the
comparison accuracy gap expanded to 19.3%-28.0% and 27.5%-29.7%, respectively. It proved the
outperformance of the GLC-2015 map over the other three products in the areas of high identification
difficulty.
Table 7. Accuracy assessments of the GLC products in three areas.

| | GLC-2015 | | Globeland30 | | FROM_GLC | | GLC_FCS30 | |
|---|---|---|---|---|---|---|---|---|
| | OA | Kappa | OA | Kappa | OA | Kappa | OA | Kappa |
| Areas of low inconsistency | 0.951 | 0.938 | 0.945 | 0.929 | 0.950 | 0.936 | 0.951 | 0.937 |
| Areas of moderate inconsistency | 0.760 | 0.723 | 0.561 | 0.498 | 0.480 | 0.411 | 0.567 | 0.495 |
| Areas of high inconsistency | 0.567 | 0.498 | 0.292 | 0.204 | 0.286 | 0.198 | 0.270 | 0.160 |

We further provided a comparative analysis of three previous GLC products and the GLC-2015 map

in areas of high inconsistency. We calculated the area of pixels with a value equal to 1 in 4° × 4° grids.
The grids that the area of pixels with a value equal to 1 account for more than 20% of the total area was
selected as grids of high inconsistency. Finally, a total number of 147 grids were selected (Fig. 12). To
compare the accuracy of the GLC-2015 map and other ones, we utilized scatter plots to represent the
relationship between the overall accuracy of one previous product and the GLC-2015 map in each grid
of high inconsistency based on the global point-based samples (Fig. 13). Most of the points were above
the 1:1line, namely the values of y-axes corresponding to those points were larger than the values of x-
axes, which demonstrated that the GLC-2015 map performed better than other GLC products in most
grids of high inconsistency. It can be found that the fitting line in each scatter plot had the intercept
exceeding 0.40, the slope less than 0.55, and the $R^2$ less than 0.35, showing that the GLC-2015 map had
a large difference with other ones.

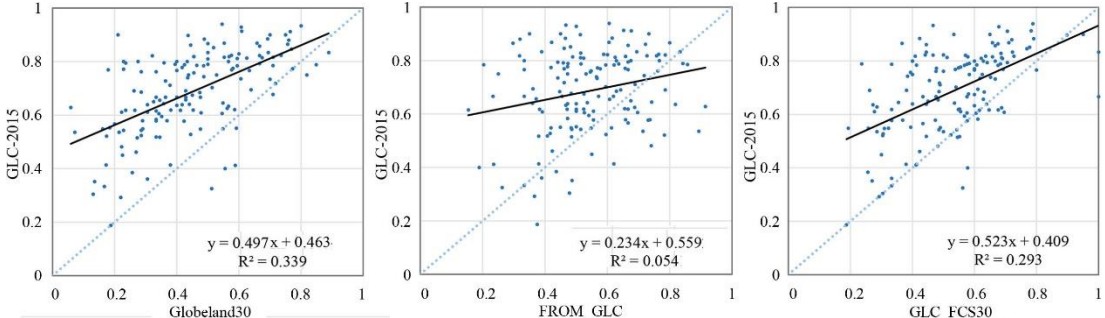


**Figure 13. Overall accuracy relationship between the GLC-2015 map and other products in grids of high**
**inconsistency.**
To intuitively compare the mapping result of the GLC-2015 map and three existing ones in areas of
high inconsistency, we focused on visual inspection in various areas based on four 5 km×5km patch-
based samples and conducted accuracy statistics, as shown in Fig. 14. In the detailed display, it is apparent
that three previous products had a large difference in four areas. As can be seen from the four visual cases,
the typical confusions between LC classes in areas of high inconsistency were as follows: (1) shrubland
was easily misclassified as forest and grassland; (2) cropland, grassland, and shrubland were heavily
confused with each other; (3) bare land was likely to be mixed with shrubland and grassland. Overall,
the GLC-2015 map surpassed other products in the local accuracy assessment. In Western Australian
mulga shrublands (Fig. 14a), the GLC-2015 map and GLC_FCS30 showed similar spatial distribution
and shape of bare land and forest, which was consistent with the real landscape. While the Globeland30
classified bare land as grassland and the FROM_GLC under-classified bare land. As for Zambezian and
mopane woodlands (Fig. 14b), the GLC-2015 map performed best with OA reaching 82.6%, followed
by the FROM_GLC. In contrast, other products mixed shrubland with forest or grassland. In agricultural
land of Western United States (Fig. 14c), the GLC-2015 and Globeland30 exhibited similar mapping
results with the ground truth while the FROM_GLC had large difference with other products. When it
comes to Guinean forest-savanna mosaic (Fig. 14d), the GLC-2015 map and Globeland30 showed high
spatial consistency, and both had accurate classification profile for cropland, forest, and impervious
surfaces, while other products misidentified cropland as other LC classes.

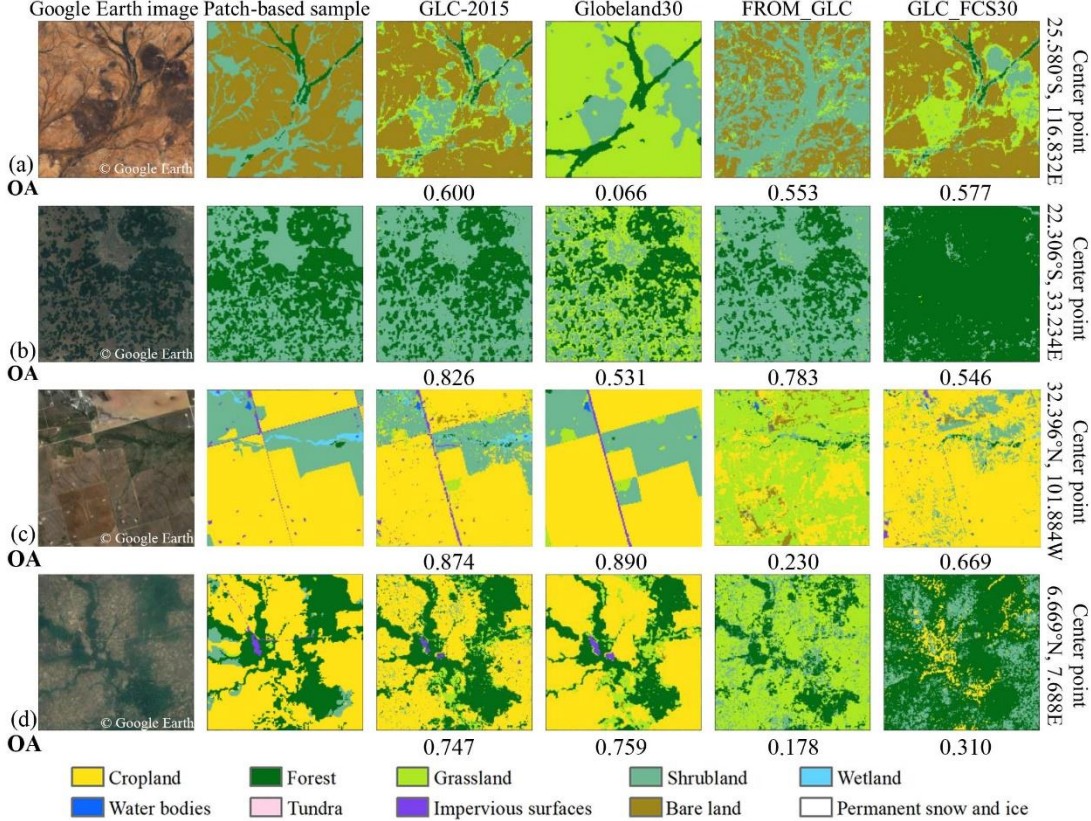


**Figure 14. Visual comparison between the GLC-2015 map and three other products based on 5km × 5km**
**patch-based samples and Google Earth images for four areas of high inconsistency (a-d). The OA for each**
**product was calculated by the corresponding patch-based sample.**
**4.6 Comparison between DSET and other methods**
**4.6.1 Inter-comparison with other data fusion methods**
The accuracy assessments on GLC-2015 obtained by DSET and global mapping results from two other
data fusion methods were conducted based on two global validation sample sets. The confusion matrixes
with the global point-based samples are shown in Table S9 and S10. The OA of the global land cover
classification obtained by the MV and SC was 72.1% and 71.8%, respectively. As shown in Table 3, the
OA of the GLC-2015 map obtained by the DSET method was 79.5%, which had an improvement of 7.4%
and 7.7% compared to mapping results from the MV and SC. In addition, the GLC-2015 map obtained
higher PA and UA for most LC classes.
When evaluating GLC maps obtained by different data fusion approaches using the global patch-
based samples, the DSET method obtained the highest OA of 83.6% and kappa coefficient of 0.566,
compared with 80.1% and 0.497 for MV, and 71.8% and 0.391 for SC (Table S11). Here, the DSET
method achieved an accuracy improvement of 3.5% and 11.8%. Compared to the two other methods, the
DSET improved the accuracy for nearly all the LC classes, especially for grassland, shrubland, and
wetland. We also compared the overall accuracy relationship between the DSET and other methods. From
the scatter plots (Fig. 15), we found that the majority of points were above the 1:1 line, implying DSET
had better mapping performance than others in most regions across the globe.

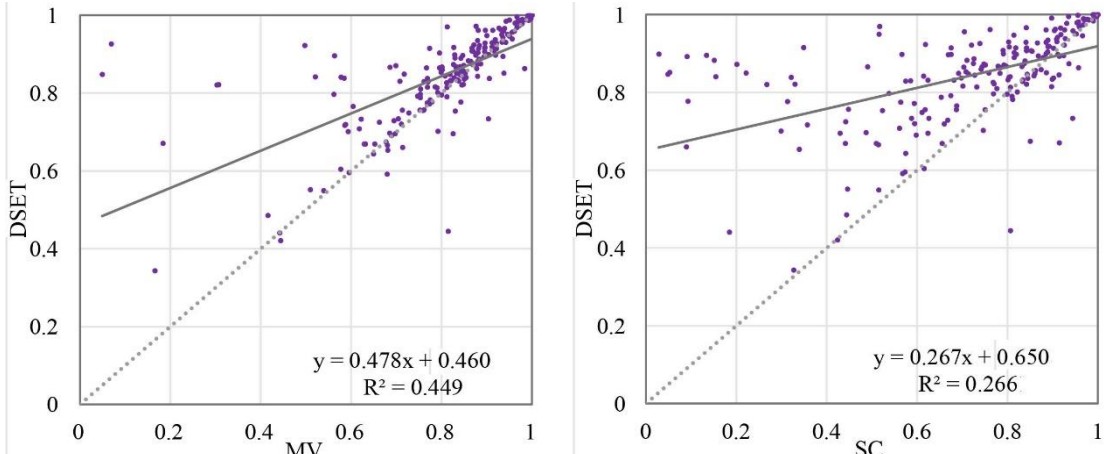


**Figure 15. Scatter plots between the DSET and other data fusion methods based on the global patch-based**
**samples.**
Land cover mapping results from the DSET and other methods were also visually illustrated in six
tiles with size of the 0.25° covering different continents, as displayed in Fig. S14. Despite that mapping
results from the DSET and MV depicted similar spatial distribution of LC classes in all tiles except the
tile in North America, the DSET more accurately delineated the impervious surfaces of small size which
scattered in cropland-dominated (Fig. S14a) or arid areas (Fig. S14c). Notably, the mapping results from
the SC method presented significant differences from that obtained by the DSET and MV. For example,
the SC method failed to capture scattered rural residential areas (Fig. S14b) and misclassified grassland
as cropland (Fig. S14d). Overall, the DSET method possessed better recognition performance in various
LC classes than the other two methods.
**4.6.2 Inter-comparison with the Random Forest**
Based on the validation data from 20% of the global point-based samples, we evaluated the quality of
the GLC-2015 map obtained by the DSET method and mapping results classified by the RF classifier for
a total of 300 grids. The DSET method obtained a mean OA of 80.9% across six continents, while the
RF achieved a lower accuracy of 69.9%. From the scatter plots which compared the OA and kappa
coefficient between the DSET and RF grid by grid, it was found that the DSET possessed higher accuracy
in most grids (Fig. S15). Especially, the points were clustered in the upper right corner of the plot (Fig.
S15a), which indicated that the RF classifier trained with the global point-based samples performed well
in those selected grids though it was inferior to the DSET method. Fig. S16 shows the OA of the DSET
and RF across six continents. We found that the DSET method outperformed RF classifier for each
continent. Especially, the mapping results of both two methods presented the lowest accuracy in Oceania.
It may be because the selected grids are located in regions with heterogeneous landscape. As for the box
plot for the RF classifier, the low hinge exceeded 60.00% in all continents except Oceania, demonstrating
the reliability of the RF classifier trained by the global point-based samples. Nevertheless, the
performance of the RF classifier was worse than the DSET method. This highlights the feasibility of the
DSET method in integrating the existing maps for a better one.
**4.7 Advancement and Limitations**
To address the problem that current 30m GLC products have great inconsistency in heterogeneous
areas and low mapping accuracy for spectral similar LC classes, this study adopted a multi-source
product fusion approach based on DSET to create an improved global land cover map (GLC-2015). The
results show that the GLC-2015 had good mapping performance with OA reaching 79.5% and 83.6%
based on two different validation sets. Compared with those existing products, the GLC-2015 greatly
improved the accuracy across the globe, especially in areas of high inconsistency with a significant
improvement of 27.5%-29.7%. Compared with other commonly used data fusion methods, the adopted
DSET approach provided higher OA and kappa coefficient which showed the benefit of the DEST in
integrating various land cover data. No matter from the respective of the global point-based samples or
the global patch-based samples, the GLC-2015 showed relatively low accuracy for grassland, shrubland,
and wetland compared to other LC classes. Those LC classes are challenging to map at the global scale
duo to their spectral similarity to other classes, ambiguous definitions, or variety with regions. However,
compared to other existing 30m GLC products, the GLC-2015 map performed better with the PA and OA
ranking first or second for grassland, shrubland, and wetland, which indicated the improvement of the
GLC-2015 in poorly-mapped LC classes. It was found that the GLC-2015 map had worse performance
in areas with more disagreements (Table 7). However, the GLC-2015 map surpassed other products in
the areas with different agree of inconsistency. Moreover, the accuracy gap between the GLC-2015 map
and other ones in areas of high inconsistency was larger than that in areas with fewer disagreements,
implying that the GLC-2015 map provides a more accurate characterization of land cover in poorly-
mapped areas. Although the GLC-2015 map was not capable of avoiding all the wrong mapping results
caused by the disagreements from the candidate GLC products, it outperformed the existing products
from the aspects of mapping accuracy for the easily misclassified classes and areas with great
inconsistency.

Although the GLC-2015 map can evidently improve mapping accuracy in inconsistent areas, there

are still some uncertainties. First, we used three multiple-class GLC maps and four single-class GLC
maps as the source data for integration. Since those products provided information of land cover at the
global scale, classification errors inevitably exist in some specific regions. The multisource product
fusion method based on DEST depends highly on the quality of those candidate maps such that the
inconsistency between those source maps might lead to incorrect classification. Second, the date time of
the GlobeLand30 is different from that of other maps. Because of the five-year time interval, there are
changes in land cover, which inevitably distort the fusion results. However, the changed areas are tiny
compared to the world's terrestrial area. The uncertainties caused by the LC changes are minor than those
from classification errors. In addition, the global point-based samples were used to evaluate the reliability
of each product. The accuracy of GlobeLand30 was lower than the other products for areas with LC
changes. In this case, the fusion depended more on other maps to avoid the errors caused by LC changes.
Third, due to the different LC definitions, uncertainties in classification system conversion are inevitable
(Zhang et al., 2017), which might cause problems for the fusion based on the DSET method. However,
we conducted a reliability evaluation of the candidate maps to reduce the influence of uncertainties in
classification system conversion on the fusion. The point-based samples used for reliability evaluation
were labeled referring to the LC definitions in our classification system so that all the maps were
evaluated under the criterion of the classification system we used. By the reliability evaluation, the
candidate maps were assessed to have lower accuracy for areas with mismatched information. When
integrating all the maps grid by grid, the mismatched information would contribute less to the fusion.
Lastly, most candidate LC products used a simple classification system without a level-2 classification
system, so they made no contributions to a more detailed classification system when they served as source
data for data fusion. Although some maps provided detailed LC classification results, such as the
GLC_FCS30 and FROM_GLC for 2015, there might be several challenges in the standardization and
uniformity of level-2 classification systems due to the large discrepancies in the definition and criteria.
Therefore, the GLC-2015 adopted a simple classification system containing 10 major LC classes. In
future work, measures will be taken to meet the expectation of a more detailed classification system for
GLC mapping. An improved GLC product with a detailed classification system rather than a simple one-
level classification system can be further developed based on the highly applicable and general DSET
method whenever more products with diverse LC classes are available. Additionally, a feasible
framework for the conversion of different level-2 classification systems into a uniform system should be
developed.
**5. Data availability**
The improved global land cover map in 2015 with 30 m resolution is available at
https://doi.org/10.6084/m9.figshare.22358143.v2 (Li et al., 2022). The GLC-2015 product is organized
by a total of 1507 4° × 4° geographical grids in GeoTIFF format across the world's terrestrial area. Each
image of the GLC-2015 product is named as "GLC-2015_lon_lat" (lon and lat represent the longitude
and latitude and of the grid's lower left corner, respectively).
**6. Conclusions**
GLC information at fine spatial resolution is vital for the global environment and climate studies which
can capture the footprint of human activity. Resulting from the differences in classification scheme,
satellite sensor data, classification algorithms and sampling strategies, the existing GLC products have
high inconsistency in some parts of the world, especially in fragmented areas and transition zones. More
accurate and reliable data with accuracy improved in areas of high mapping inconsistency is very
desirable. In this study, with the help of the GEE platform, we developed the GLC-2015 map by
integrating multiple existing GLC maps based on the DSET. The GLC-2015 map can significantly
increase the mapping accuracy and possess good recognition performance in various LC classes.
The GLC-2015 map was validated by both the global point-based samples and the global patch-
based samples. Accuracy assessments show that the GLC-2015 map achieved an OA of 79.5%, a kappa
coefficient of 0.757 using a total of 34,117 global point-based samples, and an OA of 83.6%, a kappa
coefficient of 0.566 using a total of 201 global patch-based samples. Data inter-comparison indicated
that the GLC-2015 map surpassed other three products both visually and quantitatively, by OA
improvement of 14.0%-17.8% validated with the global point-based samples and 5.9%-24.5% with the
global patch-based samples. Compared to other products, there are fewer misclassifications in the GLC-
2015 map for most LC classes, such as forest, cropland, shrubland, and water bodies. Meanwhile, the
GLC-2015 map outperformed others in terms of OA and kappa coefficient across different ecoregions
and different continents. Notably, the GLC-2015 map showed better performance than others by an
increment of 0.1%-0.6%. in overall accuracy for areas of low inconsistency, 19.3%-28.0% for areas of
moderate inconsistency, and 27.5%-29.7% for areas of high inconsistency. In addition, the mapping
results obtained by the DSET surpassed other data fusion methods with OA improvement of 7.4%-7.7%
via the global point-based samples and 3.5%-11.8% via the global patch-based samples. Therefore, it can
be concluded that the GLC-2015 map is a robust and reliable map that can significantly improve mapping
accuracy compared to previous GLC products and mapping results from other common data fusion
methods.
**Author contributions**
XL and XX conceived the research. BL and XX designed and carried out the experiments. QS and DH
provided data. BL wrote the original manuscript. XX, HZ and YC reviewed the writing.
**Competing interests**
The authors declare that they have no conflict of interest.
**Financial support**
This research has been supported by the National Key Research & Development Program of China (Grant
No. 2019YFA0607203), the National Natural Science Foundation of China (Grant No. 42001326,
42171409), and the Natural Science Foundation of Guangdong Province of China (Grant No.
2022A1515012207).

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
