# Peer review of "An improved global land cover mapping in 2015 with 30"

_Earth System Science Data, 2022_

## Referee Comment (RC1)

The authors seek to generate a new global LC map by fusing the existing ones using Dempster-Shafer theory of evidence. The manuscript is easy to follow and language is always understandable. The technique is routine and lack of innovations. By examining their result, I accidently found an error which suggests the data and method they used may not robust (see my comments below). The new map achieves higher accuracy but also shares the common problems in the existing data, such as unstable performance for specific LCs or in certain areas. As such, I am not convinced that this dataset and manuscript could be candidates for ESSD.

The GlobeLand30 in 2010 has a 5-year temporal gap between the other datasets. The LC changes in this 5 five years will bias you result. How did you deal with it? Please clarify this in the manuscript.

To my knowledge, the GSW data has multiply layers and its historical data is provided monthly. Therefore, how did you derive the water bodies from GSW for 2015. This should be explicitly clarified in the manuscript.

For single-class datasets (e.g., GSW), how did you deal with the background? Did you just ignore it or treat it as non-water? I think the latter is more useful.

Why different number of blocks were chosen for patch-based samples while the number of pixel-based samples seems to follow an equal allocation? Grids with more blocks will have more weights in the validation.

When I try to download your result, I found it was labeled by grid id. It would be better to label it with latitude and longitude (e.g., upper left corner), which is a straightforward and common way.

Echo to my comment above, I download a small file and load the smallest tile into my computer. I found a volcano (5.048 S, 151.330 E) in the Papua New Guinea was misclassified into water bodies. So, I further check the datasets you used. It turns out the error comes from the GLC_FCS30 and FROM-GLC (check the figure below). It indicates that your approach, despite the additional training samples, failed to correct such error. This may be a small problem when it comes to global mapping, and accidently found by me. But it's also a reminder for the authors to check their data and methods.

[Figure]

GLC_FCS30 adopted a detailed classification system (level-2) only in some places (seems to inherit from the ESA' CCI_LC). Therefore, I think this may lead to geographical accuracy biases even after you remap the level-2 LCs to yours. How did you deal with it, could you clarify?

The accuracy assessment of your results shows the same pattern with the existing ones, where some LCs (e.g., shrub and wetland) always possess lower accuracies. Geographically, both your results and existing ones exhibit poor performance in areas with more disagreements (Table 7). I don't see much contribution and improvements in this dataset.

---

## Author Comment (AC1)

Dear Editor and Referees:

We are particularly grateful for your careful reading, and for giving us many constructive comments on this work. According to the comments and suggestions, we have tried our best to improve the previous manuscript ESSD-2022-142 (An improved global land cover mapping in 2015 with 30 m resolution (GLC-2015) based on a multi-source product fusion approach). We believe the revised manuscript accounts for all reviewers' comments, and it was significantly improved as a result. The modified words or sentences are marked as blue color in the revised manuscript. We are providing an item-by-item response to all questions and recommendations.

Thanks very much for your time.

Best regards,

Xiaoping Liu and all co-authors

**Reviewer #1:**

**General comment:**

**The authors seek to generate a new global LC map by fusing the existing ones using Dempster Shafer theory of evidence. The manuscript is easy to follow and language is always understandable. The technique is routine and lack of innovations. By examining their result, I accidently found an error which suggests the data and method they used may not robust (see my comments below). The new map achieves higher accuracy but also shares the common problems in the existing data, such as unstable performance for specific LCs or in certain areas. As such, I am not convinced that this dataset and manuscript could be candidates for ESSD.**

Response: Thanks for the comment. These comments are very helpful for revising and improving our paper. The manuscript has been improved according to your and another reviewer's comments. The point-by-point responses are listed below in **blue**. The changes in our manuscript are marked with red.

Although some studies have adopted the Dempster-Shafer theory of evidence to create a hybrid map (Ran et al., 2012; Huang et al., 2022), they focused only on the regional scale. There are a lot of challenges to overcome when applying DSET to land cover mapping at the global scale. First, large and reliable samples are required to evaluate the reliability of the input GLC maps. Visual interpretation of a large number of samples over the globe is labor-intensive and time-consuming. Second, the application of the DSET on a global scale is restricted. Given that the characterization of the land cover landscape varies around the world, the study area must be split into sub-regions so that the quality of the existing GLC maps can be more accurately assessed for different regions. Compared to a single fusion model for regional land cover mapping, a local adaptive fusion model is demanded for global land cover mapping. Third, the traditional local computer processing method is not effective in global land cover mapping due to the lack of high computation resource and the difficulties in preparing image mosaics (Zhang et al., 2021).

To deal with the above problems, we implemented the GLC mapping using following strategies: (1) We used the interpretation-based method to generate a total of 200,000 point-based samples over the world's terrestrial area and used 80% of the samples to evaluate the reliability of each candidate map; (2) The global land was divided into 1507 4°×4° geographical grids, and the accuracy assessment on each product was performed in each grid using local samples. Meanwhile, the corresponding local adaptive fusion method based on the DSET was applied; (3) We implemented the whole LC mapping task on Google Earth Engine platform. With the help of GEE, the computer memory and image processing problems can be solved. In general, our GLC-2015 map is the first 30m-resolution land cover map that successfully overcomes the aforementioned issues in applying the DSET method on a global scale.

Based on the DSET method, the GLC-2015 map obtained better performance than any of the existing ones with an OA improvement of 12.5%-14.7% based on point-based samples and 10.9%-18.5% based on patch-based samples. Especially, our map showed the most substantial outperformance in the areas of high inconsistency, with an accuracy improvement of 21.0%-25.2% compared to 0.2%-1% for areas of low inconsistency and 17.6%-23.2% for areas of moderate inconsistency. In other words, the superiority of our map over other products is more evident in areas with more disagreements (the details can be found in our response to Comment #1-6 and Comment #1-8).

Although the GLC-2015 map provided relatively lower accuracy for grassland, shrubland, and wetland,

its accuracy for these LC classes was higher than the existing products with the PA and OA ranking first or second (the details can be found in our response for Comment #1-8).

Therefore, although there are some classification errors for some specific LC classes and regions, the GLC-2015 product can still provide a more accurate characterization of land cover than the current products and is a good complement to the existing GLC data.

References:

Huang, A., Shen, R., Li, Y., Han, H., Di, W., and Hagan, D. F.: A methodology to generate integrated land cover data for land surface model by improving Dempster-Shafer theory, Remote Sen., 14, 972, https://10.3390/rs14040972, 2022.

Ran, Y., Li, X., Lu, L., and Li, Z.: Large-scale land cover mapping with the integration of multi-source information based on the Dempster–Shafer theory, Int. J. Geogr. Inf. Sci., 26,169-191, https://doi.org/10.1080/13658816.2011.577745, 2012.

**Comment #1-1. The GlobeLand30 in 2010 has a 5-year temporal gap between the other datasets. The LC changes in this 5 five years will bias you result. How did you deal with it? Please clarify this in the manuscript.**

Response: Thanks for the comment. In the process of fusing the Globeland30, FROM_GLC and GLC_FCS30 for a new map, we just neglected the 5-year interval between the input products. The reasons why we used the Globeland30 and regarded it as reasonable are as follows:

(1) In many existing studies focusing on generating a hybrid map, multisource land cover products with different data times were used (Jung et al., 2006; Xu et al., 2014; See et al., 2015; Song et al., 2017; Tsendbazar et al., 2017). As demonstrated in previous work, the uncertainties from land cover changes are relatively more minor than that from inaccurate classification (Xu et al., 2014). In addition, the LC changes caused by the five-year gap is tiny when it comes to the LC mapping at the global scale. Thus, most spatial disagreements between the existing maps are not about LC changes over the time interval but about classification error and the integration of the maps is aimed at finding the most representative LC class (McCallum et al., 2006; See et al., 2015).

(2) When we implemented the multi-source product fusion based on the DSET method, we used a global point-based sample set verified by manual interpretation for the year 2015 to evaluate the reliability of each GLC product for each LC class in a 4° × 4° grid. If there are land cover changes in some areas from a candidate map due to the time interval from 2015, the reliability of this map is lower based on the assessment with the point-based samples. Thus, the LC classes assigned to the output map will be more likely to come from other input maps.

(3) To improve the performance of a synergetic land cover map, it is better to employ more available products with high quality (Zhong et al., 2019). The Globeland30 product has great popularity due to its good accuracy and worldwide coverage. Also, its classification scheme is almost the same as our target map (GLC-2015).

Given the considerations above, it is reasonable to use the Globeland30 product as one of the input maps though there is a 5-year temporal gap between GlobeLand30 and the two other GLC products

(FROM_GLC and GLC_FCS30).

The content on how we dealt with a 5-year temporal gap between Gloeland30 and other products had been added in our manuscript.

"Although the data time of GlobeLand30 is 2010, which has a five-year gap with other products, it was used in our project for the following reasons: (1) The changed areas of LC caused by the time interval are tiny compared to the global land area. In addition, there is relatively less uncertainty due to LC changes than due to inaccurate classification (Xu et al., 2014). Most spatial disagreements between the existing maps are about classification errors rather than LC changes over the time interval (McCallum et al., 2006; See et al., 2015); (2) We used a global point-based sample set for the year 2015 to evaluate the reliability of the input products in all 4° × 4° grids. At locations where land cover changed between 2010 and 2015, the Globeland30 was more likely to have low accuracy based on the validation and less likely to contribute to the fusion using the DSET approach. In this way, the errors due to land cover changes can be largely avoided; (3) The GlobeLand30 has great popularity due to its good accuracy. The classification system of the GlobeLand30 is almost the same as that in our study." (Revised manuscript, Line 150-160)

Furthermore, we have added the discussion about uncertainties brought by the GlobeLand30:

"Second, the date time of the GlobeLand30 is different from that of other maps. Because of the five-year time interval, there are changes in land cover, which inevitably distort the fusion results. However, the changed areas are tiny compared to the world's terrestrial area. The uncertainties caused by the LC changes are minor than those from classification errors. In addition, the global point-based samples were used to evaluate the reliability of each product. The accuracy of GlobeLand30 was lower than the other products for areas with LC changes. In this case, the fusion depended more on other maps to avoid the errors caused by LC changes." (Revised manuscript, Line 751-757)

References:

Jung, M., Henkel, K., Herold, M., and Churkina, G.: Exploiting synergies of global land cover products for carbon cycle modeling, Remote Sens. Environ., 101, 534-553, https://doi.org/10.1016/j.rse.2006.01.020, 2006.

McCallum, I., Obersteiner, M., Nilsson, S., and Shvidenko, A.: A spatial comparison of four satellite derived 1 km global land cover datasets, Int. J. Appl. Earth Obs. Geoinf., 8, 246–255, https://doi.org/10.1016/j.jag.2005.12.002, 2006.

See, L., Schepaschenko, D., Lesiv, M., McCallum, I., Fritz, S., Comber, A., Perger, C., Schill, C., Zhao, Y., Maus, V., Siraj, M. A., Albrecht, F., Cipriani, A., Vakolyuk, M. y., Garcia, A., Rabia, A. H., Singha, K., Marcarini, A. A., Kattenborn, T., Hazarika, R., Schepaschenko, M., van der Velde, M., Kraxner, F., and Obersteiner, M.: Building a hybrid land cover map with crowdsourcing and geographically weighted regression, ISPRS J. Photogramm. Remote Sens., 103, 48-56, https://doi.org/10.1016/j.isprsjprs.2014.06.016, 2015.

Song, X., Huang, C., and Townshend, J. R.: Improving global land cover characterization through data fusion, Geo-Spat. Inf. Sci., 20, 141-150, https://doi.org/10.1080/10095020.2017.1323522, 2017.

Xu, G., Zhang, H., Chen, B., Zhang, H., Yan, J., Chen, J., Che, M., Li, X., and Dong, X.: A Bayesian based method to generate a synergetic land-cover map from existing land-cover products., Remote

Sen., 6, 5589-5613, https://doi.org/10.3390/rs6065589, 2014.

Tsendbazar, N. E., Bruin, S., and Herold, M.: Integrating global land cover datasets for deriving user-specific maps, Int. J. Digit. Earth., 10, 219-237, http://dx.doi.org/10.1080/17538947.2016.1217942, 2017.

Zhong, Y., Luo, C., Hu, X., Wei, L., Wang, X., and Jin, S. Cropland product fusion method based on the overall consistency difference: A case study of China, Remote Sens., 11, 1065, https://doi:10.3390/rs11091065, 2019.

**Comment #1-2. To my knowledge, the GSW data has multiply layers and its historical data is provided monthly. Therefore, how did you derive the water bodies from GSW for 2015. This should be explicitly clarified in the manuscript.**

Response: Thanks for the comment. On the GEE platform, the JRC GSW datasets are available with multi subsets as 'Surface Water Mapping Layers', 'Monthly Water History', 'Monthly Water Recurrence', and 'Yearly Water Classification History', so that users can choose them appropriately. For our purpose, we used the GSW Yearly Water Classification History v1.1 in the GEE catalog, which provides the annual dynamics of water presence for the period of 1984 to 2019 at a 30m pixel basis. Each image of this data has a single 'waterClass' band which describes the seasonality of water throughout the year by four different types: no data, no water, seasonal water, and permanent water. Given that the seasonal water in the GSW data is not as reliable as the permanent water (Meyer et al., 2020) and might contain wetland around rivers, lakes, and ponds, we selected GSW data for the year 2015 and the permanent water was regarded as water bodies, while the seasonal water was excluded.

Correspondingly, we have added the clarification of the use of GSW in our manuscript.

"We used the GSW Yearly Water Classification History v1.1 in the GEE catalog. A single 'waterClass' band is present in each image that provides the water's seasonality throughout the year with four types: no data, no water, seasonal water, and permanent water. Since the seasonal water in GSW data is not as reliable as the permanent water (Meyer et al., 2020), we selected permanent water bodies and excluded seasonal water bodies." (Revised manuscript, Line 190-194)

References:

Meyer, M. F., Labou, S. G., Cramer, A. N., Brousil, M. R., and Luff, B. T.: The global lake area, climate, and population dataset, Sci. Data, 7, 174, https://doi.org/10.1038/s41597-020-0517-4, 2020.

**Comment #1-3. For single-class datasets (e.g., GSW), how did you deal with the background? Did you just ignore it or treat it as non-water? I think the latter is more useful.**

Response: Thanks for the comment. For those single-class datasets, we treated it as another land cover type. If the background information is regarded as a land cover type, these products provide the presence of land cover with two types. For example, the GSW contains "water" and "non-water". In this way, the quality of the GSW can be comprehensively estimated since we can provide the PA and UA for both water and non-water. In our study, "non-water" can be any of other nine class except "water". The PA and UA for the "non-water" were defaulted to 0 since the GSW did not provide information about the other nine LC classes.

In addition, the part has been added as:

"The background information of these single-class products was considered as another land cover class (e.g., non-water) participating in the fusion. The accuracy of the background information was defaulted to 0 since it did not provide information about any of the other nine categories in our classification system." (Revised manuscript, Line 170-173)

**Comment #1-4. Why different number of blocks were chosen for patch-based samples while the number of pixel-based samples seems to follow an equal allocation? Grids with more blocks will have more weights in the validation.**

Response: Thanks for the comment. In our study, the patch-based samples focused more on assessing the mapping performance of our GLC-2015 map in heterogenous landscape, such as fragmented areas and transition zones. So, we used random sampling because this method is easy to perform and capable to increase the sample size from targeted areas (Pengra et al., 2020). In our study, we randomly selected 5 km × 5km patch-based samples over the globe and across different ecoregions. Subsequently, a manual adjustment was applied to slightly increased the sample size for areas with disagreement which exists in the previous GLC maps. In this way, the sample set is more capable to verify whether the GLC-2015 makes the improvement in regions where land cover is poorly mapped by pervious maps.

As the manual interpretation of large number of 5 km × 5 km blocks is time-consuming and labor-intensive, we generated 144 samples in the previous manuscript. Based on the suggestion, we have added another 57 5km× 5km samples to make the distribution more equal. In this way, the validation of our GLC-2015 map via patch-based samples will be more reliable.

We have updated the description of patch-based samples in our manuscript.

"Simple random sampling was used to derive 5 km × 5 km blocks over the world's terrestrial area and across different ecoregions because it is easy to perform and capable to augment the sample size from target areas (Pengra et al., 2020). Since inconsistency between current GLC maps tends to appear in those heterogeneous areas, such as fragmented regions and transition zones, we slightly increased the sample size for areas with the heterogeneous landscape to better evaluate our mapping results. In total, there were 201 blocks selected as the global patch-based samples, as displayed in Figure. 3a. Then, for each block in the patch-based samples, we used ArcGIS 10.5 software to derive polygons (patches) of various sizes which captured the real landscape on the high-resolution images. Meanwhile, each polygon was manually labeled with a LC class. Four examples of producing patch-based samples are shown in Figure. 3b-c.

[Figure]

**Figure 3. Spatial distribution and selected examples of the global patch-based samples. The locations of 5 km × 5 km patch-based samples are shown as panel (a), the locations of four selected samples are remarked by red dash circles. Panels (b) and (c) illustrate the production of global patch-based samples on manual interpretation. The red lines in high-resolution images circa 2015 are results after vectorization using ArcGIS 10.5 software. Four corresponding patch-based samples are shown as (c).**" (Revised manuscript, Line 237-252)

In addition, all related validation results based on the global patch-based samples have been updated, including Table 4 and 6, Figure 9 and 10.

**Table 4. Mapping accuracy via the global patch-based samples for the GLC-2015 map**

|  | Cropland | Forest | Grassland | shrubland | Wetland | Water bodies | Tundra | Impervious surfaces | Bare land | Permanent snow and ice |
|---|---|---|---|---|---|---|---|---|---|---|
| PA | 0.862 | 0.899 | 0.626 | 0.583 | 0.232 | 0.939 | 0.701 | 0.742 | 0.757 | 0.820 |
| UA | 0.917 | 0.814 | 0.634 | 0.687 | 0.647 | 0.916 | 0.872 | 0.722 | 0.617 | 0.751 |
| OA | | | | | 0.844 | | | | | |
| Kappa | | | | | 0.564 | | | | | |

**Table 6. Mapping accuracy of the GLC products with the global patch-based samples**

|  |  | Cropland | Forest | Grassland | Shrubland | Wetland | Water bodies | Tundra | Impervious surfaces | Bare land | Permanent snow and ice | OA (Kappa coefficient) |
|---|---|---|---|---|---|---|---|---|---|---|---|---|
| GLC-2015 | PA | 0.862 | 0.899 | 0.626 | 0.583 | 0.232 | 0.939 | 0.701 | 0.742 | 0.757 | 0.820 | 0.844 |
|  | UA | 0.917 | 0.814 | 0.634 | 0.687 | 0.647 | 0.916 | 0.872 | 0.722 | 0.617 | 0.751 | (0.564) |
| Globeland30 | PA | 0.896 | 0.698 | 0.765 | 0.539 | 0.455 | 0.824 | 0.752 | 0.643 | 0.492 | 0.831 | 0.735 |

| | | | | | | | | | | | | |
|---|---|---|---|---|---|---|---|---|---|---|---|---|
| | UA | 0.891 | 0.906 | 0.444 | 0.527 | 0.157 | 0.893 | 0.500 | 0.703 | 0.829 | 0.705 | (0.434) |
| FROM_GLC | PA | 0.485 | 0.714 | 0.640 | 0.254 | 0.032 | 0.904 | 0.760 | 0.506 | 0.681 | 0.501 | 0.659 |
| | UA | 0.872 | 0.809 | 0.193 | 0.139 | 0.186 | 0.884 | 0.696 | 0.808 | 0.496 | 0.703 | (0.353) |
| GLC_FCS30 | PA | 0.865 | 0.779 | 0.398 | 0.565 | 0.363 | 0.869 | 0.051 | 0.648 | 0.658 | 0.742 | 0.712 |
| | UA | 0.857 | 0.832 | 0.509 | 0.330 | 0.132 | 0.942 | 0.573 | 0.643 | 0.462 | 0.752 | (0.414) |

[Figure]

**Figure 9. The box-plot of the accuracy for different continents. (a) overall accuracy, (b)kappa coefficient.**

[Figure]

**Figure 10. Scatter plots between the GLC-2015 map and other products obtained using the global patch-based samples.**

References:

Pengra, B. W., Stehman, S. V., Horton, J. A., Dockter, D. J., Schroeder, T. A., Yang, Z., Cohen, W. B., Healey, S. P., and Loveland, T. R.: Quality control and assessment of interpreter consistency of annual land cover reference data in an operational national monitoring program, Remote Sens. Environ., 238, 111261, https://doi.org/10.1016/j.rse.2019.111261, 2020.

**Comment #1-5. When I try to download your result, I found it was labeled by grid id. It would be better to label it with latitude and longitude (e.g., upper left corner), which is a straightforward and common way.**

Response: Thanks for the suggestion. We have named the mapping result in each grid with latitude and longitude of its lower left corner and re-uploaded our results. Correspondingly, we have changed the description of our data as well as the access.

"The improved global land cover map in 2015 with 30 m resolution is available at https://doi.org/10.6084/m9.figshare.21371304.v1 (Li et al., 2022). The GLC-2015 product is organized by a total of 1507 4° × 4° geographical grids in GeoTIFF format across the world's terrestrial area. Each image of the GLC-2015 product is named as "GLC-2015_lon_lat" (lon and lat represent the longitude and latitude and of the grid's lower left corner, respectively)." (Revised manuscript, Line 771-775)

**Comment #1-6. Echo to my comment above, I download a small file and load the smallest tile into my computer. I found a volcano (5.048 S, 151.330 E) in the Papua New Guinea was misclassified into water bodies. So, I further check the datasets you used. It turns out the error comes from the GLC_FCS30 and FROM-GLC (check the figure below). It indicates that your approach, despite the additional training samples, failed to correct such error. This may be a small problem when it comes to global mapping, and accidently found by me. But it's also a reminder for the authors to check their data and methods.**

[Figure]

Response: Thanks for pointing out the error. Since the accuracy of our data reached 76.4% assessed with the global point-based samples and 84.4% with the global patch-based samples, it is inevitable that inaccurate classification exists, especially for small land cover. Although the GLC-2015 map was not capable of avoiding all the wrong mapping results, it proved to be superior to the existing products from the aspects of mapping accuracy for the easily misclassified classes and areas with great inconsistency. As advocated by previous work, the accuracy of the integrated map is expected to be improved with more high-quality data adopted (Fritz et al., 2011; Huang et al., 2022). To our knowledge, there are several LC products with 30m resolution at the national scale, such as the National Land Cover Database (NLCD) (Yang et al., 2018) and China's land-use/cover datasets (CLUDs) (Liu et al., 2014). These national LC maps are more likely to offer higher accuracy because they were produced by experts who have good knowledge of land cover classes nationally. Future work with these reliable products employed will help to avoid inaccurate classification of the fused product.

We have added the discussion about the uncertainties caused by the source data and the further work to improve the quality of our map. The detailed revision can be seen below.

"Although the GLC-2015 map can evidently improve mapping accuracy in inconsistent areas, there are still some problems. First, we used three multiple-class GLC maps and four single-class GLC maps as the source data for integration. Since those products provide information of land cover at the global scale, classification errors inevitably exist in some specific regions. The multisource product fusion method based on DEST depends highly on the quality of those candidate maps such that the inconsistency between those source maps might lead to incorrect classification." (Revised manuscript, Line 745-750)

"As advocated by researchers that the accuracy of the integrated map is expected to be improved with more high-quality data adopted in the mapping task (Fritz et al., 2011; Huang et al., 2022). Several land cover products which focus on a national scale are more likely to offer higher accuracy because they are produced by experts who have good knowledge of land cover classes nationally. Thus, more reliable national land cover products, such as the National Land Cover Database for the year 2016 (NLCD2016) (Yang et al., 2018) and China's land-use/cover datasets (CLUDs) in 2015 (Liu et al., 2014), can further be integrated by our proposed method to develop a more accurate GLC map." (Revised manuscript, Line 763-769)

References:

Fritz, S., You, L., Bun, A., See, L., McCallum, I., Schill, C., Perger, C., Liu, J., Hansen, M., and Obersteiner, M.: Cropland for sub-Saharan Africa: A synergistic approach using five land cover data sets, Geophy. Res. Lett., 38, L04404, https://doi.org/10.1029/2010GL046213, 2011.

Huang, A., Shen, R., Li, Y., Han, H., Di, W., and Hagan, D. F.: A methodology to generate integrated land cover data for land surface model by improving Dempster-Shafer theory, Remote Sen., 14, 972, https://10.3390/rs14040972, 2022.

Liu, J., Kuang, W., Zhang, Z., Xu, X., Qin, Y., Ning, J., Zhou, W., Zhang, S., Li, R., Yan, C., Wu, S., Shi, X., Jiang, N., Yu, D., Pan, X., and Chi, W.: Spatiotemporal characteristics, patterns and causes of land use changes in China since the late 1980s, Dili Xuebao/Acta Geogr. Sin., 69, 3-14, https://doi.org/10.11821/dlxb201401001, 2014.

Yang, L., Jin, S., Danielson, P., Homer, C., Gass, L., Bender, S. M., Case, A., Costello, C., Dewitz, J., Fry, J., Funk, M., Granneman, B., Liknes, G. C., Rigge, M., and Xian, G.: A new generation of the United States National Land Cover Database: Requirements, research priorities, design, and implementation strategies, ISPRS J. Photogramm. Remote Sens., 146, 108-123, https://doi.org/10.1016/j.isprsjprs.2018.09.006, 2018.

**Comment #1-7. GLC_FCS30 adopted a detailed classification system (level-2) only in some places (seems to inherit from the ESA CCI_LC). Therefore, I think this may lead to geographical accuracy biases even after you remap the level-2 LCs to yours. How did you deal with it, could you clarify?**

Response: Thanks for the comment. We agree that there may be geographical accuracy biases from GLC_FCS30. We are sorry that we just remapped level-2 classes to match the land cover classes in our classification system without dealing with the biases because we have no effective strategy to address this problem. In our study, we collected a total number of 20,000 point-based samples over the globe and

used 80% of the samples to evaluate the accuracy of each GLC product. If the GLC_FCS30 has lower quality than other products in some regions, the LC classes from it will not be assigned to the output map. In this way, the uncertainties brought by the geographical accuracy biases of the GLC_FCS30 can be decreased. In the future, efforts will be made to solve this problem.

We have discussed this issue in the manuscript as follows:

"Third, there might be geographical accuracy biases from the GLC_FCS30 since it adopted a detailed level-2 classification system only for some areas. In this study, we used sufficient point-based samples to assess the accuracy of different GLC products. Based on the evaluation, LC classes could be selected from other more reliable candidate maps if the GLC_FCS30 provided low accuracy. In this way, the uncertainty brought by GLC_FCS30 could be reduced to some extent." (Revised manuscript, Line 758-762)

**Comment #1-8. The accuracy assessment of your results shows the same pattern with the existing ones, where some LCs (e.g., shrub and wetland) always possess lower accuracies. Geographically, both your results and existing ones exhibit poor performance in areas with more disagreements (Table 7). I don't see much contribution and improvements in this dataset.**

Response: Thanks for the comment. Our map showed relatively low accuracy for some land cover classes, such as shrubland, grassland, and wetland. However, it still provided more accurate information than the current 30m-resolution GLC maps with the PA and OA ranking first or second for those LC classes (see Table 5 and Table 6, the accuracy of the GLC-2015 ranks first is underlined with purple and the second with green).

The grassland is easy to be misclassified with cropland in some specific regions due to the high phenological similarity between them. Shrubland is mainly confused with forest due to similar spectral information and ambiguous definition. As for wetland, it is often mixed with vegetation and water bodies due to their complex spectral characteristics. It is a great challenge to accurately map those LC classes when generating a multiple-class GLC product (Liu et al., 2021; Zhang et al.,2021). With more reliable products for these three LC classes available, we can improve the mapping performance for them using our multi-source product fusion method.

**Table 5. Mapping accuracy of the GLC products with the global point-based samples.**

| | | Cropland | Forest | Grassland | Shrubland | Wetland | Water bodies | Tundra | Impervious surfaces | Bare land | Permanent snow and ice | OA (Kappa coefficient) |
|---|---|---|---|---|---|---|---|---|---|---|---|---|
| GLC-2015 | PA | 0.741 | 0.917 | 0.658 | 0.358 | 0.399 | 0.856 | 0.667 | 0.857 | 0.857 | 0.881 | 0.760 |
| | UA | 0.854 | 0.783 | 0.440 | 0.762 | 0.673 | 0.839 | 0.832 | 0.780 | 0.772 | 0.932 | (0.715) |
| Globeland30 | PA | 0.749 | 0.712 | 0.651 | 0.208 | 0.508 | 0.681 | 0.770 | 0.681 | 0.591 | 0.806 | 0.635 |
| | UA | 0.770 | 0.805 | 0.220 | 0.386 | 0.521 | 0.870 | 0.575 | 0.790 | 0.864 | 0.907 | (0.576) |
| FROM_GLC | PA | 0.385 | 0.694 | 0.705 | 0.389 | 0.347 | 0.592 | 0.705 | 0.751 | 0.723 | 0.875 | 0.613 |
| | UA | 0.647 | 0.862 | 0.269 | 0.418 | 0.282 | 0.753 | 0.687 | 0.646 | 0.774 | 0.763 | (0.554) |
| GLC_FCS30 | PA | 0.744 | 0.764 | 0.389 | 0.354 | 0.439 | 0.600 | 0.227 | 0.777 | 0.783 | 0.712 | 0.635 |
| | UA | 0.596 | 0.798 | 0.314 | 0.385 | 0.471 | 0.804 | 0.688 | 0.758 | 0.637 | 0.948 | (0.568) |

**Table 6. Mapping accuracy of the GLC products with the global patch-based samples**

| | | Cropland | Forest | Grassland | Shrubland | Wetland | Water bodies | Tundra | Impervious surfaces | Bare land | Permanent snow and ice | OA (Kappa coefficient) |
|---|---|---|---|---|---|---|---|---|---|---|---|---|
| GLC-2015 | PA | 0.862 | 0.899 | 0.626 | 0.583 | 0.232 | 0.939 | 0.701 | 0.742 | 0.757 | 0.820 | 0.844 |
| | UA | 0.917 | 0.814 | 0.634 | 0.687 | 0.647 | 0.916 | 0.872 | 0.722 | 0.617 | 0.751 | (0.564) |
| Globeland30 | PA | 0.896 | 0.698 | 0.765 | 0.539 | 0.455 | 0.824 | 0.752 | 0.643 | 0.492 | 0.831 | 0.735 |
| | UA | 0.891 | 0.906 | 0.444 | 0.527 | 0.157 | 0.893 | 0.500 | 0.703 | 0.829 | 0.705 | (0.434) |
| FROM_GLC | PA | 0.485 | 0.714 | 0.640 | 0.254 | 0.032 | 0.904 | 0.760 | 0.506 | 0.681 | 0.501 | 0.659 |
| | UA | 0.872 | 0.809 | 0.193 | 0.139 | 0.186 | 0.884 | 0.696 | 0.808 | 0.496 | 0.703 | (0.353) |
| GLC_FCS30 | PA | 0.865 | 0.779 | 0.398 | 0.565 | 0.363 | 0.869 | 0.051 | 0.648 | 0.658 | 0.742 | 0.712 |
| | UA | 0.857 | 0.832 | 0.509 | 0.330 | 0.132 | 0.942 | 0.573 | 0.643 | 0.462 | 0.752 | (0.414) |

When it comes to the mapping performance of our GLC-2015 map in areas with different-level disagreements, our map had worse performance in areas with more disagreements, as shown in Table 7. However, our map outperformed the other three in both areas of low inconsistency, moderate inconsistency, and high inconsistency. Especially, the accuracy gain of our map against other products was **21.0%-25.2%** for areas of high inconsistency, which was larger compared to 17.6%-23.2% for areas of moderate inconsistency and 0.2%-1% for areas of low inconsistency. That is to say, the superiority of our map over others is more evident in areas with more disagreements. So, we can conclude that the GLC-2015 map obtains great improvement and provides a more accurate characterization of land cover in poorly-mapped areas.

**Table 7. Accuracy assessments of the GLC products in three areas.**

| | GLC-2015 | | Globeland30 | | FROM_GLC | | GLC_FCS30 | |
|---|---|---|---|---|---|---|---|---|
| | OA | Kappa | OA | Kappa | OA | Kappa | OA | Kappa |
| Areas of low inconsistency | 0.939 | 0.922 | 0.931 | 0.912 | 0.929 | 0.909 | 0.937 | 0.919 |
| Areas of moderate inconsistency | 0.717 | 0.671 | 0.534 | 0.467 | 0.485 | 0.416 | 0.541 | 0.464 |
| Areas of high inconsistency | 0.509 | 0.430 | 0.285 | 0.196 | 0.299 | 0.212 | 0.257 | 0.144 |

In addition, based on the suggestion of another reviewer, we have added the comparison of the accuracy of the GLC-2015 map and mapping results from other data fusion methods and RF classifier. The details can be found in the **Comment #2-1** and **Comment #2-2**.

**Reviewer #2:**

**This paper developed an improved global land cover map at 30m resolution in 2015 by fusing multi-source products of land covers and other thematic mappers. Two sets of global samples with points and patches have been developed and used to evaluate the performance of derived GLC-2015. This work is high-intensive in terms of the labor involved, and the evaluation is sound with clear logic. Before recommending it for publication, I raised several concerns below, which might be helpful to improve this paper.**

**Response:** We thank the reviewer for the comments. These comments are very helpful for revising and improving our paper. The manuscript has been improved according to your and another reviewer's comments. The point-by-point responses are listed below in **blue**. The changes in our manuscript are marked with red.

**Comment #2-1. Although the authors adopted the DSET approach to generate the GLC-2015 product and compared it with similar products such as FROM-GLC and GLC_FCS30, the improvements gained from the DSET approach should be highlighted in those common approaches such as major voting and other common approaches. Otherwise, the highlights of the DSET in the manuscript should be reconsidered.**

Response: Thanks for the comment. Based on the suggestion, we have highlighted the advantage of the DSET as follows: (1) The DSET method can discount evidence form inaccurate information with a probability mass that reflects the degree of belief rather than a binary decision (Razi et al., 2019); (2) The DSET can integrate evidence from a variety of sources without the requirement of prior knowledge (Chen and Venkataramanan, 2005); (3) The DSET method can provide a total degree of belief to reflect the reliability of the final fused results.

Correspondingly, we have added this part in Introduction Section:

"Several attempts have been made to produce an accurate and consistent LC map using various methods, such as majority voting (MV), fuzzy agreement and Bayesian theory. Iwao et al. (2011) created a GLC map based on a simple majority voting method. Jung et al. (2006) generated a 1km GLC map by combination of MODIS, GLC2000 and GLCC data based on fuzzy agreement scoring. Subsequently, Fritz et al. (2011) extended the synergy method of Jung et al. (2006) by ranking LC maps and mapped the cropland extent in Sub-Saharan Africa. See et al. (2015) generated two GLC products by integrating medium resolution LC products with geographically weighted regression (GWR). Gengler and Bogaert (2018) proposed a Bayesian data fusion method and applied it to the LC mapping for a specific region in Belgium. All these researches have demonstrated that fusion method can create an integrated LC product where the mapping accuracy is greatly improved by combing the best of candidate maps. However, the MV method is sensitive to the quality of the candidate maps and has significant uncertainties when the input products exhibit great disagreement(Chen and Venkataramanan, 2005). The fuzzy agreement is highly subjective since it depends on expert assessment, while the Bayesian theory requires a prior knowledge or conditional probabilities and fails to handle the states of ignorance(Liu and Xu, 2021).

The Dempster-Shafer theory of evidence (DSET) is an evidence-based approach to reason with uncertainties. Unlike the majority voting, the DSET method can discount evidence form inaccurate information with a probability mass that reflects the degree of belief rather than a binary decision (Razi

et al., 2019). In contrast to the Bayesian theory, the DSET can integrate evidence from a variety of sources without the requirement of prior knowledge (Chen and Venkataramanan, 2005). Moreover, the reliability of the final fused results is measured by the DSET method with a total degree of belief. Although previous literature focused on the application of the DSET method in multisource data aggregation, very little research has been conducted at a global scale due to the lack of accurate and sufficient samples and the demand for adequate computing resources." (Revised manuscript, Line 93-115)

Furthermore, we have compared the DSET and other data fusion methods in our revised manuscript. Comparison shows the superiority of the DSET over other methods. First, we added how we conducted the comparison in Section 3 as follows:

**"3.5 Assessment on mapping performance of DSET and other methods**

In addition to inter-comparison between the GLC-2015 map and three existing GLC products, we compared the DSET method with two existing commonly used fusion methods, including the majority voting (MV) and spatial correspondence (SC) based on two global validation sets including 20% of the global point-based samples and the whole global patch-based samples. MV is a fusion approach that combines input maps and adopts the LC class favored by the majority of the candidate maps. In the MV method, we compared the GlobeLand30, FROM_GLC, and GLC_FCS30 at each pixel and chose the class that two or three LC products agreed for. For pixels where three LC products were different, the LC class of the product with the highest accuracy was adopted. SC method produces an integrated land cover map by selecting the LC class of the input map that has the highest spatial correspondence with the reference data. In this study, 80% of the global point-based samples were used as the reference data to obtain the SC map of each global LC product. If the class of a product agreed with that of the point-based sample, a value equal to 1 was assigned to that sample. On the contrary, a value equal to 0 was assigned to the sample if the class of the product differed from that of the sample. In each $4° \times 4°$ grid, we used the Kriging method to obtain spatial correspondence maps which have the correspondence value ranging from 0 to 1 for three products. Then, the class of the product with the highest spatial correspondence was chosen for each pixel." (Revised manuscript, Line 402-418)

Then we have added the assessment on the DSET and other two methods in Section 4 as well as supplementary material.

"4.5.1 Inter-comparison with other data fusion methods

The accuracy assessments on GLC-2015 obtained by DSET and global mapping results from two other data fusion methods were conducted based on two global validation sample sets. The error matrices with the global point-based samples are shown in Table S3 and S4. The OA of the global land cover classification obtained by the MV and SC was 69.9% and 71.9%, respectively. As shown in Table 3, the OA of the GLC-2015 map obtained by the DSET method was 76.0%, which had an improvement of 6.1% and 4.1% compared to mapping results from the MV and SC. In addition, the GLC-2015 map obtained higher PA and UA for most LC classes." (Revised manuscript, Line 670-677)

**"Table S3. The error metric for the land cover classification obtained by MV method based on the global point-based samples.**

| | Cropland | Forest | Grassland | Shrubland | Wetland | Water bodies | Tundra | Impervious | Bare land | Permanent | Total | PA |
|---|---|---|---|---|---|---|---|---|---|---|---|---|

| | Cropland | Forest | Grassland | Shrubland | Wetland | Water bodies | Tundra | Impervious surfaces | Bare land | Permanent snow and ice | Total | PA |
|---|---|---|---|---|---|---|---|---|---|---|---|---|
| Cropland | 3491 | 200 | 572 | 169 | 29 | 23 | 6 | 80 | 80 | 0 | 4650 | 0.751 |
| Forest | 541 | 7642 | 357 | 568 | 279 | 24 | 37 | 78 | 119 | 2 | 9847 | 0.792 |
| Grassland | 173 | 206 | 1634 | 176 | 57 | 10 | 16 | 45 | 134 | 1 | 2452 | 0.666 |
| Shrubland | 426 | 469 | 937 | 1184 | 82 | 10 | 38 | 45 | 431 | 0 | 3622 | 0.327 |
| Wetland | 113 | 296 | 88 | 81 | 809 | 70 | 32 | 10 | 143 | 3 | 1645 | 0.492 |
| Water bodies | 140 | 123 | 38 | 56 | 131 | 1501 | 65 | 17 | 176 | 4 | 2251 | 0.667 |
| Tundra | 0 | 134 | 173 | 115 | 10 | 17 | 1300 | 1 | 331 | 3 | 2084 | 0.624 |
| Impervious surfaces | 123 | 12 | 24 | 19 | 5 | 3 | 3 | 1264 | 42 | 0 | 1495 | 0.854 |
| Bare land | 163 | 26 | 486 | 210 | 61 | 48 | 67 | 88 | 4476 | 9 | 5634 | 0.794 |
| Permanent snow and ice | 3 | 8 | 27 | 10 | 9 | 37 | 17 | 2 | 121 | 902 | 1136 | 0.794 |
| Total | 5173 | 9116 | 4336 | 2588 | 1472 | 1743 | 1581 | 1630 | 6053 | 924 | 34616 | |
| UA | 0.675 | 0.838 | 0.377 | 0457 | 0.550 | 0.861 | 0.822 | 0.775 | 0.739 | 0.976 | | |
| OA | 0.699 | | | | | | | | | | | |
| Kappa | 0.646 | | | | | | | | | | | |

**Table S4. The error metric for the land cover classification obtained by SC method based on the global point-based samples.**

| | Cropland | Forest | Grassland | Shrubland | Wetland | Water bodies | Tundra | Impervious surfaces | Bare land | Permanent snow and ice | Total | PA |
|---|---|---|---|---|---|---|---|---|---|---|---|---|
| Cropland | 3144 | 133 | 869 | 243 | 69 | 24 | 10 | 67 | 91 | 2 | 4652 | 0.676 |
| Forest | 285 | 7524 | 628 | 737 | 155 | 19 | 149 | 80 | 47 | 23 | 9647 | 0.780 |
| Grassland | 99 | 84 | 1864 | 150 | 43 | 14 | 70 | 40 | 78 | 12 | 2454 | 0.760 |
| Shrubland | 216 | 181 | 1043 | 1603 | 66 | 30 | 86 | 74 | 318 | 11 | 3628 | 0.442 |
| Wetland | 41 | 252 | 383 | 117 | 703 | 38 | 58 | 16 | 32 | 5 | 1645 | 0.427 |
| Water bodies | 64 | 95 | 75 | 36 | 249 | 1556 | 47 | 33 | 83 | 8 | 2246 | 0.693 |
| Tundra | 9 | 46 | 102 | 55 | 53 | 13 | 1711 | 2 | 80 | 14 | 2085 | 0.821 |
| Impervious surfaces | 66 | 6 | 44 | 13 | 9 | 7 | 5 | 1261 | 80 | 5 | 1496 | 0.843 |
| Bare land | 51 | 30 | 447 | 131 | 132 | 104 | 91 | 66 | 4544 | 42 | 5638 | 0.806 |
| Permanent snow and ice | 1 | 4 | 17 | 1 | 11 | 18 | 26 | 0 | 46 | 1008 | 1132 | 0.890 |
| Total | 3976 | 8355 | 5472 | 3086 | 1490 | 1823 | 2253 | 1639 | 5399 | 1130 | 34623 | |
| UA | 0.791 | 0.901 | 0.341 | 0.519 | 0.472 | 0.854 | 0.759 | 0.769 | 0.842 | 0.892 | | |
| OA | 0.719 | | | | | | | | | | | |
| Kappa | 0.674 | | | | | | | | | | | |

” (Supplementary material with change)

“When evaluating GLC maps obtained by different data fusion approaches using the global patch-based samples, the DSET method obtained the highest OA of 84.4% and kappa coefficient of 0.564, compared with 80.1% and 0.497 for MV, and 71.8% and 0.391 for SC (Table S5). Here, the DSET method achieved an accuracy improvement of 4.3% and 12.6%. Compared to the two other methods, the DSET improved the accuracy for nearly all the LC classes, especially for grassland, shrubland, and wetland. We also compared the overall accuracy relationship between the DSET and other methods. From the scatter plots (Figure 15), we found that the majority of points were above the 1:1 line, implying DSET had better mapping performance than others in most regions across the globe.

[Figure]

**Figure 15. Scatter plots between the DSET and other data fusion methods based on the global patch-based samples.**" (Revised manuscript, Line 678-688)

"**Table S5. Mapping accuracy of different data fusion methods with the global patch-based samples**

|  |  | Cropland | Forest | Grassland | Shrubland | Wetland | Water bodies | Tundra | Impervious surfaces | Bare land | Permanent snow and ice | OA (Kappa coefficient) |
|---|---|---|---|---|---|---|---|---|---|---|---|---|
| DSET | PA | 0.862 | 0.899 | 0.626 | 0.583 | 0.232 | 0.939 | 0.701 | 0.742 | 0.757 | 0.820 | 0.844 |
|  | UA | 0.917 | 0.814 | 0.634 | 0.687 | 0.647 | 0.916 | 0.872 | 0.722 | 0.617 | 0.751 | (0.564) |
| MV | PA | 0.891 | 0.872 | 0.580 | 0.452 | 0.172 | 0.930 | 0.831 | 0.709 | 0.620 | 0.779 | 0.801 |
|  | UA | 0.890 | 0.882 | 0.569 | 0.448 | 0.166 | 0.944 | 0.827 | 0.717 | 0.612 | 0.779 | (0.497) |
| SC | PA | 0.877 | 0.856 | 0.276 | 0.177 | 0.178 | 0.870 | 0.803 | 0.690 | 0.472 | 0.675 | 0.718 |
|  | UA | 0.885 | 0.869 | 0.268 | 0.171 | 0.180 | 0.883 | 0.769 | 0.707 | 0.473 | 0.675 | (0.391) |

" (Supplementary material with change)

"Land cover mapping results from the DSET and other methods were also visually illustrated in six tiles with size of the 0.25° covering different continents, as displayed in Figure S4. Despite that mapping results from the DSET and MV depicted similar spatial distribution of LC classes in all tiles except the tile in North America, the DSET more accurately delineated the impervious surfaces of small size which scattered in cropland-dominated (Figure S4a) or arid areas (Figure S4c). Notably, the mapping results from the SC method presented significant differences from that obtained by the DSET and MV. For example, the SC method failed to capture scattered rural residential areas (Figure S4b) and misclassified grassland as cropland (Figure S4d). Overall, the DSET method possessed better recognition performance in various LC classes than the other two methods." (Revised manuscript, Line 689-697)

"

[Figure]

**Figure S4. Visual comparison between mapping results from DSET and other data fusion methods for different continents. (a) to (f) are examples for Europe, Asia, Africa, North America, South America, and Oceania, respectively.**" (Supplementary material with change)

"In summary, from the respective of both two global validation sets, the LC map from DSET (GLC-2015) obtained higher OA and performed better in identifying different classes related to those from two others, which demonstrated that the DSET method we adopted is robust to generate a new LC map from the existing products. Especially, the OA of the MV and SC was also higher than the Globeland30, FROM_GLC, and GLC_FCS30, confirming that higher accuracy could be achieved by integrating various LC maps." (Revised manuscript, Line 698-703)

Third, we emphasized the superiority of the DSET over other methods in the Section 6.

"In addition, the mapping results obtained by the DSET surpassed other data fusion methods with OA

improvement of 4.1%-6.1% via the global point-based samples and 4.3%-12.6% via the global patch-based samples. Therefore, it can be concluded that the GLC-2015 map is a robust and reliable map that can significantly improve mapping accuracy compared to previous GLC products and mapping results from other common data fusion methods." (Revised manuscript, Line 796-801)

References:

Chen, T. M. and Venkataramanan, V.: Dempster-Shafer theory for intrusion detection in ad hoc networks, IEEE Internet computing, 9, 35-41, https://doi.org/10.1109/MIC.2005.123, 2005.

Liu, K. and Xu, E.: Fusion and correction of multi-source land cover products based on spatial detection and uncertainty reasoning methods in Central Asia, Remote Sen., 13, 244, https://doi.org/10.3390/rs13020244, 2021.

Gengler, S. and Bogaert, P.: Combining land cover products using a minimum divergence and a Bayesian data fusion approach, Int. J. Geogr. Inf. Sci. , 32, 806-826, https://doi.org/10.1080/13658816.2017.1413577, 2018.

Iwao, K., Nasahara, K. N., Kinoshita, T., Yamagata, Y., Patton, D., and Tsuchida, S.: Creation of new global land cover map with map integration, J. Geogr. Inf. Syst., 3, 160-165, https://doi.org/10.4236/jgis.2011.32013, 2011.

Jung, M., Henkel, K., Herold, M., and Churkina, G.: Exploiting synergies of global land cover products for carbon cycle modeling, Remote Sens. Environ., 101, 534-553, https://doi.org/10.1016/j.rse.2006.01.020, 2006.

Razi, S., Karami Mollaei, M. R., and Ghasemi, J.: A novel method for classification of BCI multi-class motor imagery task based on Dempster–Shafer theory, Inf. Sci., 484, 14-26, https://doi.org/10.1016/j.ins.2019.01.053, 2019.

See, L., Schepaschenko, D., Lesiv, M., McCallum, I., Fritz, S., Comber, A., Perger, C., Schill, C., Zhao, Y., Maus, V., Siraj, M. A., Albrecht, F., Cipriani, A., Vakolyuk, M. y., Garcia, A., Rabia, A. H., Singha, K., Marcarini, A. A., Kattenborn, T., Hazarika, R., Schepaschenko, M., van der Velde, M., Kraxner, F., and Obersteiner, M.: Building a hybrid land cover map with crowdsourcing and geographically weighted regression, ISPRS J. Photogramm. Remote Sens., 103, 48-56, https://doi.org/10.1016/j.isprsjprs.2014.06.016, 2015.

**Comment #2-2. How about the mapping performance if using these samples (80%) do the classification directly? Because these samples have been manually visualized and are qualified for the classification task. Please add some test results or discuss this issue in the manuscript.**

Response: Thanks for the comment. Based on the suggestion, we have added the test results of the classification using the Random Forest classifier trained on the 80% of the global point-based samples in our revised manuscript. The detailed revision can be seen below.

"In addition to the comparison between DSET and two other fusion methods, we compared the mapping performance of DSET with Random Forest (RF) which is considered one of the most popular algorithms for land cover mapping. In the land cover classification using the FR classifier, all available Level-2 Tier 1surface reflectance (SR) data of Landsat 8 OLI (Operational Land Imager) sensors from the year 2015 and two adjacent years on GEE was employed. All Landsat images have been atmospherically corrected.

The following six bands were used as input features: blue, green, red, NIR, SWIR1, and SWIR2. To improve the mapping performance, several important spectral indices, including DNVI, NDWI, and NDBI were also used as auxiliary data to the RF classifier. The RF classifier was trained on 80% of the global point-based samples since those samples were of high quality after manual visual interpretation of high-resolution images. As the global land cover mapping based on the RF classifier is a tough task, we randomly selected a total of 300 grids with the size of 4° (Figure S1) and used corresponding local classifiers to these grids. Then, the mapping results were validated by the remaining 20% of the global-point samples." (Revised manuscript, Line 419-431)

"

**Figure S1. The spatial distribution of the selected 4° × 4° grids where the comparison between DSET and RF classifier was implemented.**" (Supplementary material with change)

"4.5.2 Inter-comparison with the Random Forest

Based on the validation data from 20% of the global point-based samples, we evaluated the quality of the GLC-2015 map obtained by the DSET method and mapping results classified by the RF classifier for a total of 300 grids. The DSET method obtained an average OA of 77.7% across six continents, while the RF achieved a lower accuracy of 69.8%. From the scatter plots which compared the OA and kappa coefficient between the DSET and RF grid by grid, we can see that the DSET possessed higher accuracy in most grids (Figure S5). Especially, the points were clustered in the upper right corner of the plot (Figure S5a), which indicated that the RF classifier trained with the global point-based samples performed well in those selected grids though it was inferior to the DSET method. Figure S6 shows the OA of the DSET and RF across six continents. We found that the DSET method outperformed RF classifier for each continent. Additionally, the DSET was similar to the RF in terms of the ranking of accuracy over the continents. Especially, the mapping results of both two methods presented the lowest accuracy in Oceania. It may be because the selected grids were located in regions with heterogeneous landscape. As for the box plot for the RF classifier, the low hinge exceeded 60.00% in all continents except Oceania, demonstrating the reliability of the RF classifier trained by the global point-based samples. Nevertheless, the performance of the RF classifier was worse than the DSET method. This highlights the feasibility of the DSET method in integrating the existing maps for a better one." (Revised manuscript, Line 704-720)

"

[Figure]

**Figure S5. Scatter plots between the DSET and RF based on the global point-based samples.**

[Figure]

**Figure S6. The box plot of the overall accuracy comparison between the DSET and RF for different continents.**" (Supplementary material with change)

**Comment #2-3. The proposed work can also be applied in regions with adequate and high-quality data, such as NLCD in the US and China. This can be improved or discussed in the revised manuscript.**

Response: Thanks for the suggestion. We have added the section 4.6 "Advancement and Limitations" in our manuscript and discussed future work based on more high-quality data.

"Although the GLC-2015 map can evidently improve mapping accuracy in inconsistent areas, there are still some uncertainties. First, we used three multiple-class GLC maps and four single-class GLC maps as the source data for integration. Since those products are created for providing information of land cover at the global scale, classification errors will inevitably exist in some specific regions. The multisource product fusion method based on DEST depends highly on the quality of those candidate maps such that the inconsistency between those source maps might lead to incorrect classification." (Revised manuscript, Line 745-750)

"As advocated by researchers that the accuracy of the integrated map is expected to be improved with more high-quality data adopted in the mapping task (Fritz et al., 2011; Huang et al., 2022). Several land cover products which focus on a national scale are more likely to offer higher accuracy because they are produced by experts who have good knowledge of land cover classes nationally. Thus, more reliable

national land cover products, such as the National Land Cover Database for the year 2016 (NLCD2016) (Yang et al., 2018) and China's land-use/cover datasets (CLUDs) in 2015 (Liu et al., 2014), can further be integrated by our proposed method to develop a more accurate GLC map." (Revised manuscript, Line 763-769)

References:

Fritz, S., You, L., Bun, A., See, L., McCallum, I., Schill, C., Perger, C., Liu, J., Hansen, M., and Obersteiner, M.: Cropland for sub-Saharan Africa: A synergistic approach using five land cover data sets, Geophys. Res. Lett., 38, L04404, https://doi.org/10.1029/2010GL046213, 2011.

Huang, A., Shen, R., Li, Y., Han, H., Di, W., and Hagan, D. F.: A methodology to generate integrated land cover data for land surface model by improving Dempster-Shafer theory, Remote Sen., 14, 972, https://doi.org/10.3390/rs14040972, 2022.

Liu, J., Kuang, W., Zhang, Z., Xu, X., Qin, Y., Ning, J., Zhou, W., Zhang, S., Li, R., Yan, C., Wu, S., Shi, X., Jiang, N., Yu, D., Pan, X., and Chi, W.: Spatiotemporal characteristics, patterns and causes of land use changes in China since the late 1980s, Dili Xuebao/Acta Geogr. Sin., 69, 3-14, https://doi.org/10.11821/dlxb201401001, 2014.

Yang, L., Jin, S., Danielson, P., Homer, C., Gass, L., Bender, S. M., Case, A., Costello, C., Dewitz, J., Fry, J., Funk, M., Granneman, B., Liknes, G. C., Rigge, M., and Xian, G.: A new generation of the United States National Land Cover Database: Requirements, research priorities, design, and implementation strategies, ISPRS J. Photogramm. Remote Sens., 146, 108-123, https://doi.org/10.1016/j.isprsjprs.2018.09.006, 2018.

**Minor Comments:**

**Comment #2-4. Page 108: BPA function. This term should be fully spelled when it first appears in the main text.**

Response: Thanks for the comment. The full name of the term has been added at the place it first appears in the revised manuscript.

"To fulfill the purpose, we first performed reliability evaluation, where the accuracy of each GLC product for each LC class in each $4° \times 4°$ geographical grid is regarded as the evidential probability to create the basic probability assignment (BPA) function." (Revised manuscript, Line121-123)

**Comment #2-5. Page 172: The selection of 4°×4° should be discussed.**

Response: Thanks for the comment. For large-scale or global land cover mapping, previous researchers divided the study area into a lot of sub-regions (Gong et al., 2020; Huang et al.,2021; Jin et al., 2022; Liu et al., 2020; Zhang et al., 2020,2021; Zhao et al.,2021). When applying the DSET method to generate a global hybrid map, it is more useful to use the local adaptive fusion model for each sub-regions rather than a single model for the whole globe. We divided the global land into 4°×4° sub-regions with the following two considerations:

(1) Sufficiency of samples for land cover classes. If we generate the samples in a small spatial grid such as a Landsat scene, the size of samples might be insufficient and it was also difficult to obtain samples

for the rare land cover classes.

(2) Computation capacity and memory of the GEE platform. The GEE platform provides unprecedented opportunities for global land cover classification tasks due to the access to numerous analysis-ready earth observations datasets and high-performance, intrinsically parallel computation (Gorelick et al., 2017). However, GEE has computation capacity limitations. It is impossible to implement mapping work at a sub-region as large as we want because of the issue of running out of memory.

In our study, to balance the mapping efficiency on the GEE platform and the sufficiency for land cover classes in sub-regions, we split the globe into 1507 4º×4° geographical grids and then conducted land cover mapping at the regional scale.

Correspondingly, we have added the explanation for dividing the world's terrestrial area into 4º×4° grids.

"For large-scale or global land cover mapping, previous researchers divided the study area into a lot of sub-regions and conducted classification in each sub-region on GEE (Gong et al., 2020; Liu et al., 2020; Huang et al., 2021; Jin et al., 2022; Zhang et al., 2021; Zhao et al., 2021). The shape and size of sub-region vary in previous work, such as hexagons with a side length of 2°, geographical grids with a size of 1°×1°, 3.5º×3.5°, 5º×5°, or 10º×10°. When deciding on the size of sub-regions, two important factors should be considered. The size of samples in each sub-region should be sufficient so that the rare land cover classes will not be missed. On the other hand, it is impossible to implement mapping work at a sub-region as large as we want due to memory constraints. To balance the mapping efficiency on the GEE platform and the sufficiency for land cover classes, we split the world's terrestrial area into 1507 4º×4° geographical grids." (Revised manuscript, Line 262-271)

References:

Gong, P., Li, X., Wang, J., Bai, Y., Chen, B., Hu, T., Liu, X., Xu, B., Yang, J., Zhang, W., and Zhou, Y.: Annual maps of global artificial impervious area (GAIA) between 1985 and 2018, Remote Sens. Environ., 236, 111510, https://doi.org/10.1016/j.rse.2019.111510, 2020.

Gorelick, N., Hancher, M., Dixon, M., Ilyushchenko, S., Thau, D., and Moore, R.: Google Earth Engine: Planetary-scale geospatial analysis for everyone, Remote Sens, Environ., 202, 18–27, https://doi.org/10.1016/j.rse.2017.06.031, 2017.

Huang, X., Li, J., Yang, J., Zhang, Z., Li, D., and Liu, X.: 30m global impervious surface area dynamics and urban expansion pattern observed by Landsat satellites: From 1972 to 2019., Sci. China Earth Sci, 64, 1922–1933, https://doi.org/10.1007/s11430-020-9797-9, 2021.

Jin, Q.; Xu, E.; and Zhang, X.: A fusion method for multisource land cover products based on superpixels and statistical extraction for enhancing resolution and improving accuracy, Remote Sens, 14, 1676, https://doi.org/10.3390/rs14071676, 2022.

Liu, X., Huang, Y., Xu, X., Li, X., Li, X., Ciais, P., Lin, P., Gong, K., Ziegler, A. D., Chen, A., Gong, P., Chen, J., Hu, G., Chen, Y., Wang, S., Wu, Q., Huang, K., Estes, L., and Zeng, Z.: High-spatiotemporal-resolution mapping of global urban change from 1985 to 2015, Nature Sustainability, 3, 564-570, https://doi.org/10.1038/s41893-020-0521-x, 2020.

Zhang, X., Liu, L., Chen, X., Gao, Y., Xie, S., and Mi, J.: GLC_FCS30: global land-cover product with fine classification system at 30 m using time-series Landsat imagery, Earth Syst. Sci. Data, 13, 2753-2776, https://doi.org/10.5194/essd-13-2753-2021, 2021.

Zhao, J., Yu, L., Liu, H., Huang, H., Wang, J., and Gong, P.: Towards an open and synergistic framework for mappingglobal land cover, PeerJ, 9, e11877, https://doi.org/10.7717/peerj.11877, 2021.

**Comment #2-6. Page 178-179: I wonder why the initial samples generated from the FROM_GLC were used in this study, not other land cover products. Explanations about this topic should be discussed in the manuscript.**

Response: Thanks for the comment. Collecting global point-based samples is a key step in our study. We employed stratified random sampling which generates samples for land cover classes based on area proportion from reference land cover product. In this method, a classification map served as prior knowledge. This map was only used to derive proper size for each class. The reasons why we chose the FROM_GLC as the reference map other than Globeland30 and GLC_FCS30 are as follows:

(1) From the respective of time, the data time of Globeland30 product is 2010 which has a 5-year interval from our samples. If we used Globeland30, there would be some land cover change between 2010 and 2015 and the size of samples for each class would be affected.

(2) For the three existing 30m global land cover products (Globeland30, FROM_GLC, and GLC_FCS30), the classification system used for FROM_GLC level-1 has the same land cover classes used in the Globeland30, while the GLC_FCS30 has differences with others in the classification scheme and definition of land cover classes due to the inheritance from the CCI-LC classification system(Gao et al., 2020; Liu et al., 2021). As we adopted the same classification scheme as Globeland30, it was reasonable to choose the FROM_GLC rather than GLC_FCS30.

Considering both the data time and classification system, the FROM_GLC was used in our study. Given that there are inevitably errors in samples generated from the FROM_GLC, the class label from the FROM_GLC was not assigned to our samples. Instead, we checked all the points according to Google Earth high-resolution images and labeled them. Through the manual interpretation, we can guarantee the global samples are accurate.

Correspondingly, the related description of generating the global point-based samples has been updated in our revised manuscript.

"The stratified random sampling depends on area ratio of LC classes from a LC product. We used the FROM_GLC as prior knowledge rather than the Globeland30 and GLC_FCS30 with two considerations: (1) the FROM_GLC has the same data time as our target map (GLC-2015) while the Globeland30 has a 5-year interval from our samples, which affects the size of samples for each LC class; (2)the 10 level-1 land cover classes of the FROM_GLC is similar to that in the classification system of the GLC-2015, while the GLC_FCS30 has differences with the GLC-2015 in the classification scheme and definition of land cover classes." (Revised manuscript, Line 210-217)

"The FROM_GLC shows low accuracy for some LC classes, especially for cropland and forest (Gao et al., 2020; Liu et al., 2021b; Zhang et al., 2021; Zhang et al., 2022). If the global samples were extracted with LC class label from the FROM_GLC, there would be inevitable errors. Therefore, the FROM_GLC was only used to determine the size and location of samples for each LC class. Instead, all the points were manually labeled according to Google Earth high-resolution images." (Revised manuscript, Line 222-226)

References:

Gao, Y., Liu, L., Zhang, X., Chen, X., Mi, J., and Xie, S.: Consistency Analysis and Accuracy Assessment of Three Global 30-m Land-Cover Products over the European Union using the LUCAS Dataset, Remote Sen., 12, 3479, https://doi.org/10.3390/rs12213479, 2020.

Liu, L., Zhang, X., Gao, Y., Chen, X., Shuai, X., and Mi, J.: Finer-resolution mapping of global land cover: Recent developments, consistency analysis, and prospects, Journal of Remote Sensing, 5289697, https://doi.org/10.34133/2021/5289697, 2021.

Zhang, X., Liu, L., Chen, X., Gao, Y., Xie, S., and Mi, J.: GLC_FCS30: global land-cover product with fine classification system at 30 m using time-series Landsat imagery, Earth Syst. Sci. Data, 13, 2753-2776, https://doi.org/10.5194/essd-13-2753-2021, 2021.

Zhang, C., Dong, J., and Ge, Q.: Quantifying the accuracies of six 30-m cropland datasets over China: A comparison and evaluation analysis, Comput. Electron. Agric., 197, 106946, https://doi.org/10.1016/j.compag.2022.106946, 2022.

**Comment #2-7. Page 198: Why select these 1507 samples randomly? Can they be determined according to their ecoregions or cover types? It is better to explain it here clearly.**

Response: Thanks for the comment. It is a widely-used strategy to divide the study area into sub-regions for the large-scale or global land cover mapping. In previous work, the size of a sub-region ranged from 1°×1° to 10º×10° (Gong et al., 2020; Liu et al., 2020; Huang et al., 2021; Jin et al., 2022; Zhang et al., 2021; Zhao et al., 2021). Considering the limited memory resource on GEE and the sufficiency for land cover classes, the global land was split to 4°×4° grids. In total, there were 1507 grids across the globe.

We suspect that the reviewer is asking why we select 93 grids from a total of 1507 grids. Deriving samples according to ecoregions or cover types is suitable to increase the size of rare classes (Olofsson et al., 2014). While simple random sampling is easy to perform and capable to increase the sample size from targeted areas (Pengra et al., 2020). In our study, the patch-based samples focused more on assessing the mapping performance of our GLC-2015 map in the heterogenous landscape, such as fragmented areas and transition zones. In our previous manuscript, we just used simple random sampling to selected 93 grids and derive 5 km × 5 km patch-based samples from those grids. Then, a manual adjustment was applied to slightly increased the sample size for areas with disagreement which exists in the previous GLC maps. In this way, the sample set was more capable to verify whether the GLC-2015 improved in regions where land cover was poorly mapped by previous maps. Considering that the previous 144 patch-based samples may not enough for each ecoregion, we have added another 57 samples in the revised manuscript.

We have updated the description of collecting global path-based samples in the manuscript as follows:

"Simple random sampling was used to derive 5 km × 5 km blocks over the world's terrestrial areas because it is easy to perform and capable to augment the sample size from target areas (Pengra et al., 2020). Since inconsistency between current GLC maps tends to appear in those heterogeneous areas, such as fragmented regions and transition zones, we slightly increased the sample size for areas with the heterogeneous landscape to better evaluate our mapping results. In total, there were 201 blocks selected as the global patch-based samples, as displayed in Figure. 3a.

[Figure]

(a)

(b) High-resolution image

(c) Path-based sample

(1)    (2)    (3)    (4)

**Figure 3. Spatial distribution and selected examples of the global patch-based samples. The locations of 5 km × 5 km patch-based samples are shown as panel (a), the locations of four selected samples are remarked by red dash circles. Panels (b) and (c) illustrate the production of global patch-based samples on manual interpretation. The red lines in high-resolution images circa 2015 are results after vectorization using ArcGIS 10.5 software. Four corresponding patch-based samples are shown as (c).**" (Revised manuscript, Line 237-252)

References:

Gong, P., Li, X., Wang, J., Bai, Y., Chen, B., Hu, T., Liu, X., Xu, B., Yang, J., Zhang, W., and Zhou, Y.: Annual maps of global artificial impervious area (GAIA) between 1985 and 2018, Remote Sens. Environ., 236, 111510, https://doi.org/10.1016/j.rse.2019.111510, 2020.

Huang, X., Li, J., Yang, J., Zhang, Z., Li, D., and Liu, X.: 30m global impervious surface area dynamics and urban expansion pattern observed by Landsat satellites: From 1972 to 2019., Sci. China Earth Sci, 64, 1922–1933, https://doi.org/10.1007/s11430-020-9797-9, 2021.

Jin, Q.; Xu, E.; and Zhang, X.: A fusion method for multisource land cover products based on superpixels and statistical extraction for enhancing resolution and improving accuracy, Remote Sens, 14, 1676, https://doi.org/10.3390/rs14071676, 2022.

Liu, X., Huang, Y., Xu, X., Li, X., Li, X., Ciais, P., Lin, P., Gong, K., Ziegler, A. D., Chen, A., Gong, P., Chen, J., Hu, G., Chen, Y., Wang, S., Wu, Q., Huang, K., Estes, L., and Zeng, Z.: Highspatiotemporal-resolution mapping of global urban change from 1985 to 2015, Nature Sustainability, 3, 564-570, https://doi.org/10.1038/s41893-020-0521-x, 2020.

Olofsson, P., Foody, G. M., Herold, M., Stehman, S. V., Woodcock, C.E., and Wulder, M. A. : Good practices for estimating area and assessing accuracy of land change, Remote Sens. Environ., 148, 42-57, https://doi.org/10.1016/j.rse.2014.02.015, 2014.

Pengra, B. W., Stehman, S. V., Horton, J. A., Dockter, D. J., Schroeder, T. A., Yang, Z., Cohen, W. B., Healey, S. P., and Loveland, T. R.: Quality control and assessment of interpreter consistency of annual land cover reference data in an operational national monitoring program, Remote Sens. Environ., 238, 111261, https://doi.org/10.1016/j.rse.2019.111261, 2020.

Zhang, X., Liu, L., Chen, X., Gao, Y., Xie, S., and Mi, J.: GLC_FCS30: global land-cover product with fine classification system at 30 m using time-series Landsat imagery, Earth Syst. Sci. Data, 13, 2753-2776, https://doi.org/10.5194/essd-13-2753-2021, 2021.

Zhao, J., Yu, L., Liu, H., Huang, H., Wang, J., and Gong, P.: Towards an open and synergistic framework for mappingglobal land cover, PeerJ, 9, e11877, https://doi.org/10.7717/peerj.11877, 2021.

**Comment #2-8. Figure. 3 and Figure.4 can be combined.**

Response: Thanks for the suggestion. Figure. 3 and Figure.4 have been combined as suggested in the revised manuscript. Moreover, we added 57 patch-based samples over the globe and three examples of the production of patch-based samples on manual interpretation. The improved figure is shown as below.

"

[Figure]

(a)

(b) High-resolution image

(c) Path-based sample

(1)    (2)    (3)    (4)

**Figure 3. Spatial distribution and selected examples of the global patch-based samples. The locations of 5 km × 5 km patch-based samples are shown as panel (a), the locations of four selected samples are remarked by red dash circles. Panels (b) and (c) illustrate the production of global patch-based samples on manual interpretation. The red lines in high-resolution images circa 2015 are results after vectorization using ArcGIS 10.5 software. Four corresponding patch-based samples are shown as (c).**" (Revised manuscript, Line 247-252)

**Also, the main text uses "Figure" whereas the figure caption uses "Figure". Please make them consistent.**

Response: Thanks for the suggestion. We have changed the "Fig" in main text as "Figure".

**Comment #2-9. Page 281: how to determine these two thresholds: 25% and 75% in Eq. (4). Please explain.**

Response: Thanks for the comment. We used the local adaptive fusion model to combine the existing products for each grid. To avoid the inequacy in the size of local samples for rare land cover classes, we also used global samples to evaluate products' reliability. Since the local samples plays a more critical role in the local accuracy assessment, higher weight should be assigned to the local samples in the construction of the BPA for each grid. Through the tests in several girds, it was found that when the local

samples counted for 75% of the whole sample set and the global samples counts for 25%, the fusion method exhibited robust performance and achieved relatively high accuracy. We have explained why we used 75% and 25% as two thresholds in our manuscript.

"Given that the local accuracy for a 4°×4° grid was not able to adequately reflect the actual land cover landscape, especially for the rare LC classes, global accuracy was incorporated into the construction of the BPA to avoid uncertainties from a local point of view. Since assessment based on local samples plays a more critical role in BPA construction for a local grid, higher weight should be assigned to local accuracy. In this case, we chose 75% as the weight for local accuracy and 25% for global accuracy as this ratio could achieve robust performance for different regions." (Revised manuscript, Line 331-337)

---

## Referee Report (RR1)

General comment:

The work put into developing the GLC-2015 product can fulfill the global land cover data pool and may give us more precise information about the Earth system. Before being published, the manuscript still has to go through revisions. Authors should include more product comparisons to further prove the advancement of their product.

Suggestion and comments

1. The main concern is the results given in Tables 5 and 6, where it is shown that the accuracy of the GLC-2015 product is not much improved than the other products. It is advised that the authors quantify the area differences for each land cover type at multiple scales, including global, continental, national, and ecoregional scales. Also, the authors should provide more visual comparisons regarding each land cover type with current products, including global-scale data, national-scale data, and other prevalence-used data. The visual comparisons ought to be focused on various vegetation types and climatic zones. For instance, the Amur basin, the Tibetan Plateau, Canada, and coastal mangroves should be taken into account when comparing mapping results for Wetlands. With these comparisons, the authors can state that their product is more robust than other products regarding what land cover types in what regions.

2. Why not use national-scale land cover data (e.g., CLUD, CLCD, NLCD) as candidates if the authors have acknowledged their value in lines 756 to 762? National-scale land cover products with the participation of local experts are more accurate than global products and should therefore be considered for inclusion in the development of GLC-2015.

3. One of the motivations behind the development of global land use land cover data is the expectation of a more detailed classification scheme (e.g., GLC_FCS30's scheme). The authors may provide insights on how to meet this expectation with the proposed fusion method.

4. The format of the reference needs to be standardized.

---

## Author Response (AR2)

Dear Editor and Referees:

We are particularly grateful for your careful reading, and for giving us many constructive comments on this work. According to the comments and suggestions, we have tried our best to improve the previous manuscript ESSD-2022-142 (An improved global land cover mapping in 2015 with 30 m resolution (GLC-2015) based on a multi-source product fusion approach). We believe the revised manuscript accounts for all reviewers' comments, and it was significantly improved as a result. The modified words or sentences are marked as blue color in the revised manuscript. We are providing an item-by-item response to all questions and recommendations.

Thanks very much for your time.

Best regards,
Xiaoping Liu and all co-authors

**Reviewer #1:**

**General comment:**

**The manuscript has been improved in terms of additional patch-based validation and asset name prefixes. However, due to the following concerns, I am still not convinced that this dataset and manuscript could be candidates for ESSD.**

**The method is defective when input GLC maps contain errors (see my comments below) and their reliability could not be fully evaluated by the samples. In such a case, the accuracy of some LC maps was overestimated, leading to the wrong contribution to the fusion. This can be confirmed by the misclassification of bare land from water (check my previous comments), where FROM-GLC and GLC_FCS30 identified a volcano as water. This also indicates that input samples are not representative.**

Response: Thanks for the comment. These comments are very helpful for revising and improving our paper. The manuscript has been improved according to your and another reviewer's comments. The point-by-point responses are listed below in **blue**. The changes in our manuscript are marked with red.

The DSET method can discount evidence from inaccurate information with a probability mass that reflects the degree of belief (Razi et al., 2019). In this study, we collected over 200,000 point-based samples. We think these samples are representative and enough to evaluate the reliability of each product. Based on the reliability assessment, the accuracy of each product can be appropriately estimated, and the errors from maps can be reduced in the fusion process. The efficiency of the DSET method in discounting wrong information can be demonstrated by the visual comparison in Sundarbans where mangroves are prevalent (see Figure R1). It can be found that GLC-2015 accurately depicted the spatial distribution of mangroves, although the FROM_GLC and GLC_FCS30 performed poorly, with almost no wetlands captured.

[Figure]

**Figure R1. The wetland extent from GLC-2015, Globeland30, FROM_GLC, GLC_FCS30, and the GMW.**

GLC mapping at the global scale is challenging, and there are inevitable errors, such as the misclassification of bare land from water. To reduce the error from the input maps, we employed several

reliable products to improve our mapping results (the details can be found in our response for Comment #1-5). Accuracy assessment shows the improvement of the GLC-2015 in various LC classes when three national-scale products are used.

References:

Razi, S., Karami Mollaei, M. R., and Ghasemi, J.: A novel method for classification of BCI multi-class motor imagery task based on Dempster–Shafer theory, Inf. Sci., 484, 14-26, https://doi.org/10.1016/j.ins.2019.01.053, 2019.

**Comment #1-1. The 4° grid is large in the case of global mapping, as many studies have employed smaller grid (such as the 1° grid in GAUD). Generally, smaller grids will help assess each input LC map's accuracy (e.g., geographical variations). Therefore, it's suggested to investigate the relationship between grid size and mapping performance (e.g., global OA as a function of grid size), and find a proper size.**

Response: Thanks for the comment. For large-scale or global land cover mapping, previous researchers divided the study area into a lot of sub-regions (Gong et al., 2020; Huang et al.,2021; Jin et al., 2022; Liu et al., 2020; Zhang et al., 2020,2021; Zhao et al.,2021). The shape and size of sub-region vary in previous work, for example, hexagons with a side length of 2°, geographical grids with a size of 1°×1°, 3.5°×3.5°, 5°×5°, or 10°×10°. When applying the DSET method to generate a global hybrid map, the following two factors should be taken into consideration when we decided the size of the sub-region:

(1) Sufficiency of samples for land cover classes. If we generate the samples in a small spatial grid such as a Landsat scene, the size of samples might be insufficient and it was also difficult to obtain samples for the rare land cover classes.

(2) Computation capacity and memory of the GEE platform. The GEE platform provides unprecedented opportunities for global land cover classification tasks due to the access to numerous analysis-ready earth observations datasets and high-performance, intrinsically parallel computation (Gorelick et al., 2017). However, GEE has computation capacity limitations. It is impossible to implement mapping work at a sub-region as large as we want because of the issue of running out of memory.

In the study, we found that when the size was larger than 4°, the execution aborted due to the complex computation exceeding available memory on the GEE platform. To investigate the relationship between grid size and mapping performance of the DSET method, we performed some tests in randomly selected areas. We generated five 4°×4° grids and divided each grid into sub-grids of 2° and 1°. Then, the fusion process was performed in grids of 1°, 2° and 4°. The accuracy comparison shows that the fusion method obtained the highest OA with 4° gird (Table S1). Therefore, we split the globe into 1507 4°×4° geographical grids and then conducted land cover mapping at the regional scale.

**Table S1. Relationship between the overall accuracy of the fusion method and the size of sub-regions.**

| Grid ID | 1 degree | 2 degree | 4 degree |
|---------|----------|----------|----------|
| 0119 | 0.647 | 0.757 | **0.844** |
| 0317 | 0.641 | 0.589 | **0.872** |
| 0603 | 0.735 | 0.765 | **0.971** |
| 0817 | 0.515 | 0.636 | **0.723** |
| 1206 | 0.929 | 0.952 | **0.976** |

Correspondingly, we have added the reason why we divided the world's terrestrial area into 4º×4° grids.

"For large-scale or global land cover mapping, previous researchers divided the study area into a lot of sub-regions and conducted classification in each sub-region on GEE (Gong et al., 2020; Liu et al., 2020; Huang et al., 2021; Jin et al., 2022; Zhang et al., 2021; Zhao et al., 2021). The shape and size of sub-region vary in previous work, such as hexagons with a side length of 2°, geographical grids with a size of 1°×1°, 3.5º×3.5°, 5º×5°, or 10º×10°. When deciding on the size of sub-regions, two important factors should be considered. The size of samples in each sub-region should be sufficient so that the rare land cover classes will not be missed. On the other hand, it is impossible to implement mapping work at a sub-region as large as we want due to memory constraints. To determine the appropriate size, we tested different sizes of the sub-region (see Table S1). Result shows that dividing the study area into 4º×4° grids performed best. Therefore, we split the world's terrestrial area into 1507 4º×4° geographical grids." (Revised manuscript, Line 278-288)

References:

Gong, P., Li, X., Wang, J., Bai, Y., Chen, B., Hu, T., Liu, X., Xu, B., Yang, J., Zhang, W., and Zhou, Y.: Annual maps of global artificial impervious area (GAIA) between 1985 and 2018, Remote Sens. Environ., 236, 111510, https://doi.org/10.1016/j.rse.2019.111510, 2020.

Gorelick, N., Hancher, M., Dixon, M., Ilyushchenko, S., Thau, D., and Moore, R.: Google Earth Engine: Planetary-scale geospatial analysis for everyone, Remote Sens, Environ., 202, 18–27, https://doi.org/10.1016/j.rse.2017.06.031, 2017.

Huang, X., Li, J., Yang, J., Zhang, Z., Li, D., and Liu, X.: 30m global impervious surface area dynamics and urban expansion pattern observed by Landsat satellites: From 1972 to 2019., Science China Earth Sciences, 64, 1922–1933, https://doi.org/10.1007/s11430-020-9797-9, 2021.

Jin, Q.; Xu, E.; and Zhang, X.: A fusion method for multisource land cover products based on superpixels and statistical extraction for enhancing resolution and improving accuracy, Remote Sens, 14, 1676, https://doi.org/10.3390/rs14071676, 2022.

Liu, X., Huang, Y., Xu, X., Li, X., Li, X., Ciais, P., Lin, P., Gong, K., Ziegler, A. D., Chen, A., Gong, P., Chen, J., Hu, G., Chen, Y., Wang, S., Wu, Q., Huang, K., Estes, L., and Zeng, Z.: High-spatiotemporal-resolution mapping of global urban change from 1985 to 2015, Nature Sustainability, 3, 564-570, https://doi.org/10.1038/s41893-020-0521-x, 2020.

Zhang, X., Liu, L., Chen, X., Gao, Y., Xie, S., and Mi, J.: GLC_FCS30: global land-cover product with fine classification system at 30 m using time-series Landsat imagery, Earth Syst. Sci. Data, 13, 2753-2776, https://doi.org/10.5194/essd-13-2753-2021, 2021.

Zhao, J., Yu, L., Liu, H., Huang, H., Wang, J., and Gong, P.: Towards an open and synergistic framework for mappingglobal land cover, PeerJ, 9, e11877, https://doi.org/10.7717/peerj.11877, 2021.

**Comment #1-2. I understand the high quality of GlobeLand30. But I don't think the LC changes caused by the 5-year gap of GlobeLand30 can be largely avoided through the input samples. Due to the simple stratified sampling, these samples could rarely capture the LC changes during the interval. Thus, the accuracy of GlobeLand30 (against the actual LC in 2015) will be overestimated. In such a way, the contribution of Globeland30 (when LC changes) to fusion would not be as small as the authors suggested.**

Response: Thanks for the comment. As you are concerned, the LC changes caused by the 5-year interval of the GlobeLand30 might bring uncertainties for the fusion. When we evaluated the reliability of the source maps, we only wanted to know whether these maps provided accurate information about land cover classes for the year 2015 because the data time of our target map (GLC-2015) is 2015. Therefore, for the GlobeLand30, there is no difference between the mismatched information caused by the LC changes and the errors caused by inaccurate classification since neither is consistent with the actual landscape in 2015. In other words, we can assume that the data time of GlobeLand30 is 2015. In this assumption, any mapping result of the GlobeLand30 that was inconsistent with the point-based samples for 2015 was defined as wrong, no matter whether it belongs to inaccurate mapping results due to LC changes or the classification method. In addition, as we mentioned in the previous manuscript, the changed areas of LC caused by the time interval are tiny compared to the global land area. Thus, the GlobeLand30 is still a good choice for the mapping task in 2015 regardless of LC changes.

We performed some experiments in China, a developing country that has undergone significant land cover changes (Yang et al., 2021), to figure out whether the errors from LC changes could be avoided in the fusion process. First, the China's land-use/cover datasets (CLUDs) for 2010 and 2015 were used to derive changed areas in the 5-year temporal interval since the CLUDs had good quality with OA exceeding 90.0% (Liu et al., 2014). The CLUDs datasets show that 2.9% of land cover in China had been changed during the 5 years. Then, a total of 13295 point-based samples in China were filtered from the whole sample set used in the study. Among point-based samples in China, there were 547 samples located in changed areas, accounting for 4.1% of all samples in China. Therefore, the point-based samples we used are enough to capture LC changes between 2010 and 2015 in China.

Furthermore, an accuracy comparison between the GLC-2015 and GlobeLand30 was conducted in changed areas of China using 144 validation samples for 2015 (Table R1). Results show that the GLC-2015 performed better in changed areas with OA reaching 74.3%, compared to OA of 43.5% for the GlobeLand30. This indicated to some extent that the GLC-2015 did not inherit inaccurate information about the GlobeLand30 due to LC changes. Especially, the GlobeLand30 showed low PA of 23.8% and 32.1% for impervious surfaces and forest. This is because these two LCs changed significantly from 2010 to 2015 (Liu et al., 2014), so the GlobeLand30 in 2010 failed to describe the changed land cover classes during five years. However, the GLC-2015 improved PA and UA for LCs except for permanent snow and ice compared to the GlobeLand30. This improvement in the changed areas of China is because, with the help of the reliability evaluation of each input map, the fusion depended more on other maps than GlobeLand30. Therefore, it is reasonable to believe that LC changes caused by the 5-year interval of the GlobeLand30 can be largely avoided in the fusion based on the DSET method.

**Table R1. Comparison of mapping accuracy for the GLC-2015 and GlobeLand30 in changed areas of China.**

| | | Cropland | Forest | Grassland | Shrubland | Wetland | Water bodies | Impervious surfaces | Bare land | Permanent snow and ice | OA |
|---|---|---|---|---|---|---|---|---|---|---|---|
| GLC-2015 | PA | 0.750 | 0.793 | 0.809 | 0.50 | 0.588 | 1.00 | 0.826 | 0.500 | 1.00 | 0.743 |
| | UA | 0.857 | 0.821 | 0.531 | 1.00 | 0.588 | 0.714 | 0.864 | 0.833 | 1.00 | |
| Globeland30 | PA | 0.633 | 0.321 | 0.583 | 0 | 0.250 | 0.400 | 0.238 | 0.500 | 1.00 | 0.435 |
| | UA | 0.413 | 0.750 | 0.212 | 0 | 1.00 | 0.666 | 1.00 | 0.400 | 1.00 | |


Response: Thanks for the comment. We agree that the GlobeLand30 has high quality and has been used to collect samples in many researches (Ma et al., 2017; Zhang et al., 2020; Zhang et al., 2023). However, it was too time-consuming to re-select more than 200,000 point-based samples over the glob using the GlobeLand30 as the reference map. In the study, the FROM_GLC was selected because it has the same date time and similar classification with our target map (GLC-2015). Given that the FROM_GLC shows relatively low accuracy for some special LCs, it was only used to provide approximate information about size and location of samples for each LC class. **Notably, the point-based samples did not inherit the land cover class from the FROM_GLC. Instead, we visually interpreted all the points according to Google Earth high-resolution images and labeled them**.

Moreover, we compared the overall accuracy of the fusion method with samples collected from FROM_GLC and Globeland30 in four grids, as listed in Table R2. It can be found that the accuracy difference between Method 1 and Method 2 was small in most grids. In addition, the fusion method with samples derived from Globeland30 (Method 2) did not perform better in every grid.

**Table R2. Comparison the overall accuracy of the two fusion methods. Method 1 and 2 denote the fusion method with samples derived from FROM_GLC and Globleland30, respectively.**

| Grid ID | Method 1 | Method 2 |
|---------|----------|----------|
| 0059 | 0.785 | 0.643 |
| 0258 | 0.709 | 0.730 |
| 0608 | 0.769 | 0.754 |
| 0671 | 0.968 | 0.972 |


**Table R3. Comparison of mapping accuracy for the previous GLC-2015 and updated GLC-2015 in China.**

| | | Cropland | Forest | Grassland | Shrubland | Wetland | Water bodies | Impervious surfaces | Bare land | Permanent snow and ice | OA (Kappa coefficient) |
|---|---|---|---|---|---|---|---|---|---|---|---|
| **Previous** | PA | 0.795 | 0.949 | 0.802 | 0.263 | 0.334 | 0.844 | 0.818 | 0.873 | 0.810 | **0.805** |
| | UA | 0.862 | 0.811 | 0.738 | 0.657 | 0.682 | 0.730 | 0.918 | 0.856 | 0.870 | **(0.763)** |
| **Updated** | PA | 0.844 | 0.965 | 0.968 | 0.316 | 0.598 | 0.896 | 0.905 | 0.891 | 0.793 | **0.888** |
| | UA | 0.930 | 0.928 | 0.803 | 0.923 | 0.870 | 0.741 | 0.899 | 0.962 | 0.958 | **(0.864)** |

**Table R4. Comparison of mapping accuracy for the previous GLC-2015 and updated GLC-2015 in USA.**

| | | Cropland | Forest | Grassland | Shrubland | Wetland | Water bodies | Impervious surfaces | Bare land | Permanent snow and ice | OA (Kappa coefficient) |
|---|---|---|---|---|---|---|---|---|---|---|---|
| **Previous** | PA | 0.858 | 0.972 | 0.865 | 0.556 | 0.685 | 0.935 | 0.767 | 0.875 | 1.00 | **0.794** |
| | UA | 0.921 | 0.742 | 0.665 | 0.975 | 0.804 | 0.921 | 0.891 | 0.467 | 0.667 | **(0.754)** |
| **Updated** | PA | 0.890 | 0.958 | 0.917 | 0.869 | 0.903 | 0.935 | 0.867 | 0.911 | 1.00 | **0.910** |
| | UA | 0.944 | 0.932 | 0.815 | 0.972 | 0.878 | 0.977 | 0.903 | 0.689 | 1.00 | **(0.893)** |

When comparing the mapping performance at global scale (Table R5), it was found that **the OA of the updated GLC-2015 was 1.6% higher than the previous one**. In addition, the PA and OA of the updated GLC-2015 had a slightly improvement for almost all the land cover classes.

**Table R5. Comparison of mapping accuracy for the previous GLC-2015 and updated GLC-2015 at global scale.**

| | | Cropland | Forest | Grassland | Shrubland | Wetland | Water bodies | Tundra | Impervious surfaces | Bare land | Permanent snow and ice | OA (Kappa coefficient) |
|---|---|---|---|---|---|---|---|---|---|---|---|---|
| **Previous** | PA | 0.755 | 0.925 | 0.713 | 0.412 | 0.395 | 0.874 | 0.669 | 0.857 | 0.881 | 0.891 | **0.780** |
| | UA | 0.864 | 0.797 | 0.504 | 0.815 | 0.708 | 0.852 | 0.833 | 0.795 | 0.776 | 0.928 | **(0.739)** |
| **Updated** | PA | 0.778 | 0.910 | 0.739 | 0.435 | 0.622 | 0.875 | 0.667 | 0.869 | 0.883 | 0.891 | **0.795** |
| | UA | 0.878 | 0.823 | 0.535 | 0.841 | 0.690 | 0.863 | 0.839 | 0.802 | 0.789 | 0.937 | **(0.757)** |

The quantitative comparison was also conducted in the areas of low inconsistency, moderate inconsistency, and high inconsistency, as listed in Table R6. The updated GLC-2015 performed better than the previous GLC-2015 in three areas, with an accuracy improvement of 0.3% in areas of low inconsistency, 0.7% in areas of moderate inconsistency, and 3.9% in areas of high inconsistency.

**Table R6. Comparison of mapping accuracy for the previous GLC-2015 and updated GLC-2015 in three areas.**

| | Previous | | Updated | |
|---|---|---|---|---|
| | OA | Kappa | OA | Kappa |
| Areas of low inconsistency | 0.948 | 0.931 | 0.951 | 0.938 |
| Areas of moderate inconsistency | 0.743 | 0.701 | 0.760 | 0.723 |
| Areas of high inconsistency | 0.528 | 0.450 | 0.567 | 0.498 |

Overall, the updated GLC-2015 obtained higher overall accuracy and had better performance in nearly all the LC classes than the previous one in multiple scales. Therefore, we can conclude that it is helpful to use more national-scale products to improve the performance of the GLC-2015 in some LC classes and regions.

We have employed three national-scale products in the fusion process. The description of three national-scale products had been added in Section 2.1 as follows:

"Land cover products which focus on a national scale are more likely to possess higher accuracy because they were produced by experts who have good knowledge of land cover classes nationally. Thus, the National Land Cover Database 2016 (NLCD 2016) for the year 2016 (Yang et al., 2018), China's land-use/cover datasets (CLUDs) (Liu et al., 2014) for 2015, and the annual China land cover dataset (CLCD)

(Yang and Huang, 2021) for 2015 were also included in the fusion. NLCD 2016 database, which provides continuous and accurate information about land cover and change from 2001 to 2016 at an interval of 2 or 3 years, was produced based on a pixel- and object-based approach and an effective post-classification process (Yang et al., 2018). The level-1 and level-2 overall accuracy of NLCD 2016 database for 2016 was 90.6% and 86.4%, respectively (Wickham et al., 2021). CLUDs, developed by the digital interpretation method using Landsat images, provide land cover information over China from 1980s to 2015. The overall accuracy of CLUDs reached 94.3% and 91.2% for level-1 and level-2 land cover classes, respectively (Liu et al., 2014). CLCD was generated with stable training samples derived from CLUDs and Landsat time series. Assessed with 5463 validation samples, CLCD obtained an overall accuracy of 79.31% (Yang and Huang, 2021).

**Table 1. Detailed information of GLC products and national-scale LC products used in this paper.**

| Product name | Satellite sensors | Year of reference | Access | Literature |
|---|---|---|---|---|
| Globeland30 | Landsat TM/ETM+ HJ-1 A/B | 2010 | http://www.globallandcover.com/ | (Chen et al., 2015) |
| FROM_GLC | Landsat TM/ETM+/OLI | 2015 | http://data.ess.tsinghua.edu.cn/ | (Gong et al., 2013) |
| GLC_FCS30 | Landsat OLI | 2015 | https://doi.org/10.5281/zenodo.3986872 | (Zhang et al., 2021) |
| GAUD | Landsat TM/ETM+/OLI | 2015 | https://doi.org/10.6084/m9.figshare.11513178.v1 | (Liu et al., 2020) |
| GFC | Landsat TM/ETM+ | 2015 | http://earthenginepartners.appspot.com/science-2013-global-forest | (Hansen et al., 2013) |
| JRC GSW | Landsat TM/ETM+/OLI | 2015 | http://global-surface-water.appspot.com/ | (Pekel et al., 2016) |
| GMW | ALOS PALSAR Landsat TM/ETM+ | 2015 | https://data.unep-wcmc.org/datasets/45 | (Bunting et al., 2018) |
| NLCD 2016 | Landsat TM /OLI | 2016 | https://www.mrlc.gov/data/nlcd-2016-land-cover-conus | (Yang et al., 2018) |
| CLUDs | Landsat TM HJ-1 CBERS-1 | 2015 | / | (Liu et al., 2014) |
| CLCD | Landsat TM/ETM+/OLI | 2015 | https://doi.org/10.5281/zenodo.4417810 | (Yang and Huang, 2021) |

" (Revised manuscript, Line 163-177)

The relationship between our classification system and the classification systems of three national-scale land cover products has been added in supplementary material:

**"Table S3. Relationship between our classification system and the classification systems of the three national-scale LC products.**

| Id | GLC-2015 | CLCD | CLUDs | NLCD 2016 |
|---|---|---|---|---|
| 10 | Cropland | Cropland | Rice paddy | Pasture |
|  |  |  | Bare farmland | Cropland |
|  |  |  | Orchard |  |
| 20 | Forest | Forest | Wooden land | Deciduous forest |

| | | | | Evergreen forest |
| --- | --- | --- | --- | --- |
| | | | | Mixed forest |
| 30 | Grassland | Grassland | Grassland, highly-covered | Grassland |
| | | | Grassland, medium-covered | |
| | | | Grassland, lowly-covered | |
| 40 | Shrubland | Shrub | Shrubland | Shrubland |
| 50 | Wetland | Wetland | Marshland | Woody wetlands |
| | | | Tidal flat | Herbaceous wetlands |
| | | | Salt marsh | |
| | | | Flooded flat | |
| 60 | Water bodies | Water | Rivers | Water |
| | | | Lakes | |
| | | | Reservoir and ponds | |
| 70 | Tundra | | | |
| 80 | Impervious surfaces | Impervious | Urban | Urban, open space |
| | | | Rural | Urban, low intensity |
| | | | Other construction sites | Urban, med. Intensity |
| | | | | Urban, high intensity |
| 90 | Bare land | Barren | Sandy land | Barren |
| | | | Gobi desert | |
| | | | Barren | |
| | | | Bare rocky land | |
| 100 | Permanent snow and ice | Snow/ice | Permanent snow and ice | Ice/snow |

” (Supplementary material with change)

Lastly, since we employed three national-scale products in the fusion process, all the related results about the GLC-2015 were updated in the revised manuscript.


Response: Thanks for the comment. These comments are very helpful for revising and improving our paper. The manuscript has been improved according to your and another reviewer's comments. The point-by-point responses are listed below in **blue**. The changes in our manuscript are marked with red.

**Comment #2-1. The main concern is the results given in Tables 5 and 6, where it is shown that the accuracy of the GLC-2015 product is not much improved than the other products. It is advised that the authors quantify the area differences for each land cover type at multiple scales, including global, continental, national, and ecoregional scales. Also, the authors should provide more visual comparisons regarding each land cover type with current products, including global-scale data, national-scale data, and other prevalence-used data. The visual comparisons ought to be focused on various vegetation types and climatic zones. For instance, the Amur basin, the Tibetan Plateau, Canada, and coastal mangroves should be taken into account when comparing mapping results for Wetlands. With these comparisons, the authors can state that their product is more robust than other products regarding what land cover types in what regions.**

Response: Thanks for the comment. Based on your suggestion, we have quantified the area difference for each land cover class at multiple scales in our revised manuscript.

"**4.3.3 Areal comparison for individual classes**

[revised manuscript text omitted]

"

[Figure]

**Figure S5. Areal comparison of various land cover classes among GLC products over six continents.**

[Figure]

**Figure S6. Areal comparison of various land cover classes among GLC products over the top40 countries.**

[Figure]

**Figure S7. Areal comparison of various land cover classes among GLC products over different ecoregions.**
(Supplementary material with change)

Also, we have added the visual comparison for individual classes between GLC-2015 and other widely-used products in our revised manuscript.

"**4.3.4 Visual inter-comparison for individual classes**

The visual comparison of cropland in GLC-2015, Globeland30, FROM_GLC, GLC_FCS30, GSFAD30 (Xiong et al., 2017; Teluguntla et al., 2018), and other national-scale maps was conducted in three local regions (Figure S8). In the Egyptian agricultural area, GLC-2015, FROM_GLC, and GLC-FCS30 shared similar delineation of the cropland and had a good representation of cropland with fine spatial details. Since the date time of the Google Earth image is 2015, Globeland30 missed the newly cultivated cropland. GFASD30 had the largest cropland area among five products but misclassified bare land as cropland. In the agricultural area of Southeastern China, GLC-2015 had an agreement with GFSAD30 and CLCD. Globeland30 and GLC_FCS30 overestimated the area of cropland. As for FROM_GLC, it failed to depict the spatial distribution of cropland and had many omissions. In cropland-dominated areas of the United States, FROM_GLC significantly underestimated the extent of cropland. The other five products exhibited a similar delineation of cropland, but there were little differences in some small areas. For example, Globeland30 misclassified some grassland into cropland, and NLCD 2016 had a good ability to distinguish the farm rack." (Revised manuscript, Line 613-626)

"

**Figure S8. Comparing the crop extent from GLC-2015 and other widely used products in three agricultural regions.**" (Supplementary material with change)

"We also compared the performance in the forest of different products in three forest-prevalent regions of Congo, China and the United States (Figure S 9). Overall, GLC-2015 and Globeland30 showed accurate delineation in three regions. FROM_GLC also had good performance for the forest in Congo and USA but overestimated the forest in China, mislabeling shrubland and grassland as forest. Furthermore, GFC tended to miss sparse trees in China, and GLC_FCS30 underestimated the extent of forest in both three regions. As for national-scale products, CLCD and NLCD 2016 had a good ability to identify the details of forest, while CLUD dramatically missed both dense and sparse woodlands."
(Revised manuscript, Line 627-633)

"

[Figure]

**Figure S9. Comparing the forest extent from GLC-2015 and other widely used products in three forest-**

**dominated regions.**" (Supplementary material with change)

"Furthermore, to compare the performance in the wetland of GLC-2015 with other global and national-scale products, three wetland regions in South-central Canada, coastal America, and Sundarbans were selected. It can be found that GLC-2015 and Globeland30 had similar representation and performed well in identifying wetlands over three regions (Figure S10). Unexpectedly, FROM_GLC performed poorly in each region, with almost no wetlands captured. GLC_FCS30 also showed unstable quality in three regions. For example, it highly underestimated the wetland area in coastal America and completely mislabeled the mangrove as cropland in Sundarbans. NLCD 2016 and GMW accurately demonstrated the spatial pattern of wetlands, while the CA_wetlands map underestimated the wetland extent because it defined wetlands by wetland frequency of no less than 80% from 2000 to 2016 (Wulder et al., 2018). " (Revised manuscript, Line 634-642)

"

[Figure]

**Figure S10. Comparing the wetland extent from GLC-2015 and other widely used products in three wetland-dominated regions.**" (Supplementary material with change)

"To understand the spatial distribution of impervious surfaces in different products, a comparison of mapping results for three megacities, including Tokyo, Shanghai, and New York, was shown in Figure S11. In Tokyo, a high consistency was found between GLC-2015, FROM_GLC, and GAUD, and both successfully captured the impervious surfaces in peri-urban areas. GLC_FCS30 showed the largest area for impervious surfaces because it misclassified many croplands into impervious surfaces. In Shanghai, GLC_FCS30 underestimated the central city, and CLUD lost the details of impervious surfaces because it was developed using the visual interpretation method. Other products generally had the similar representation and accurately demonstrated the spatial distribution of the city. For New York, the FROM_GLC, GLC_FCS30, and GAUD agreed well with GLC-2015, while Globeland30 and NLCD 2016 had high impervious areas than others." (Revised manuscript, Line 643-652)

"

[Figure]

**Figure S11. Comparing the impervious extent from GLC-2015 and other widely used products in three megacities.**" (Supplementary material with change)


Response: Thanks for the comment. We think it is an excellent suggestion. We have used three national-scale land cover data, including CLUD2015, CLCD2015 for China, and NLCD2016 for America, as candidate maps for fusion to improve the mapping performance of GLC-2015. Since the classification system of CLUD2015 and NLCD2016 is different with our classification system, the land cover classes of CLUD2015 and NLCD2016 were reclassified according to the classification system we adopted. Then, we re-performed land cover mapping with national-scale products included.

To figure out whether the GLC-2015 performed better when three national-scale land cover products were used, we compared mapping results without national-scale data (previous GLC-2015) and with national-scale data (updated GLC-2015) at the national scale and global scale.

Assessed with point-based samples at national scale (Tables R3-4), the OA of the updated GLC-2015 over China and the United States achieved 88.8% and 91.0%, which had an improvement of 8.3% and 11.6% compared to the OA of previous GLC-2015 (80.5% for China and 79.4% for the USA). For each land cover class, the updated GLC-2015 performed better than the previous one in both nations. As for

shrubland and wetland, the previous GLC-2015 showed relatively low accuracy. Compared to the previous GLC-2015, the updated one greatly improved the mapping accuracy for these two land cover classes in two nations.

**Table R3. Comparison of mapping accuracy for the previous GLC-2015 and updated GLC-2015 in China.**

| | | Cropland | Forest | Grassland | Shrubland | Wetland | Water bodies | Impervious surfaces | Bare land | Permanent snow and ice | OA (Kappa coefficient) |
|---|---|---|---|---|---|---|---|---|---|---|---|
| **Previous** | PA | 0.795 | 0.949 | 0.802 | 0.263 | 0.334 | 0.844 | 0.818 | 0.873 | 0.810 | **0.805** |
| | UA | 0.862 | 0.811 | 0.738 | 0.657 | 0.682 | 0.730 | 0.918 | 0.856 | 0.870 | **(0.763)** |
| **Updated** | PA | 0.844 | 0.965 | 0.968 | 0.316 | 0.598 | 0.896 | 0.905 | 0.891 | 0.793 | **0.888** |
| | UA | 0.930 | 0.928 | 0.803 | 0.923 | 0.870 | 0.741 | 0.899 | 0.962 | 0.958 | **(0.864)** |

**Table R4. Comparison of mapping accuracy for the previous GLC-2015 and updated GLC-2015 in USA.**

| | | Cropland | Forest | Grassland | Shrubland | Wetland | Water bodies | Impervious surfaces | Bare land | Permanent snow and ice | OA (Kappa coefficient) |
|---|---|---|---|---|---|---|---|---|---|---|---|
| **Previous** | PA | 0.858 | 0.972 | 0.865 | 0.556 | 0.685 | 0.935 | 0.767 | 0.875 | 1.00 | **0.794** |
| | UA | 0.921 | 0.742 | 0.665 | 0.975 | 0.804 | 0.921 | 0.891 | 0.467 | 0.667 | **(0.754)** |
| **Updated** | PA | 0.890 | 0.958 | 0.917 | 0.869 | 0.903 | 0.935 | 0.867 | 0.911 | 1.00 | **0.910** |
| | UA | 0.944 | 0.932 | 0.815 | 0.972 | 0.878 | 0.977 | 0.903 | 0.689 | 1.00 | **(0.893)** |

When comparing the mapping performance at global scale (Table R5), it was found that **the OA of the updated GLC-2015 was 1.6% higher than the pervious one**. In addition, the PA and OA of the updated GLC-2015 had a slightly improvement for almost all the land cover classes.

**Table R5. Comparison of mapping accuracy for the previous GLC-2015 and updated GLC-2015 at global scale.**

| | | Cropland | Forest | Grassland | Shrubland | Wetland | Water bodies | Tundra | Impervious surfaces | Bare land | Permanent snow and ice | OA (Kappa coefficient) |
|---|---|---|---|---|---|---|---|---|---|---|---|---|
| **Previous** | PA | 0.755 | 0.925 | 0.713 | 0.412 | 0.395 | 0.874 | 0.669 | 0.857 | 0.881 | 0.891 | **0.780** |
| | UA | 0.864 | 0.797 | 0.504 | 0.815 | 0.708 | 0.852 | 0.833 | 0.795 | 0.776 | 0.928 | **(0.739)** |
| **Updated** | PA | 0.778 | 0.910 | 0.739 | 0.435 | 0.622 | 0.875 | 0.667 | 0.869 | 0.883 | 0.891 | **0.795** |
| | UA | 0.878 | 0.823 | 0.535 | 0.841 | 0.690 | 0.863 | 0.839 | 0.802 | 0.789 | 0.937 | **(0.757)** |

The quantitative comparison was also conducted in the areas of low inconsistency, moderate inconsistency, and high inconsistency, as listed in Table R6. The updated GLC-2015 performed better than the previous GLC-2015 in three areas, with an accuracy improvement of 0.3% in areas of low inconsistency, 0.7% in areas of moderate inconsistency, and 3.9% in areas of high inconsistency.

**Table R6. Comparison of mapping accuracy for the previous GLC-2015 and updated GLC-2015 in three areas.**

| | Previous | | Updated | |
|---|---|---|---|---|
| | OA | Kappa | OA | Kappa |
| Areas of low inconsistency | 0.948 | 0.931 | 0.951 | 0.938 |
| Areas of moderate inconsistency | 0.743 | 0.701 | 0.760 | 0.723 |

| | | | | |
|---|---|---|---|---|
| Areas of high inconsistency | 0.528 | 0.450 | 0.567 | 0.498 |

Correspondingly, we have added the description of three national-scale products in Section 2.1 as follows:

"Land cover products which focus on a national scale are more likely to possess higher accuracy because they were produced by experts who have good knowledge of land cover classes nationally. Thus, the National Land Cover Database 2016 (NLCD 2016) for the year 2016 (Yang et al., 2018), China's land-use/cover datasets (CLUDs) (Liu et al., 2014) for 2015, and the annual China land cover dataset (CLCD) (Yang and Huang, 2021) for 2015 were also included in the fusion. NLCD 2016 database, which provides continuous and accurate information about land cover and change from 2001 to 2016 at an interval of 2 or 3 years, was produced based on a pixel- and object-based approach and an effective post-classification process (Yang et al., 2018). The level-1 and level-2 overall accuracy of NLCD 2016 database for 2016 was 90.6% and 86.4%, respectively (Wickham et al., 2021). CLUDs, developed by the digital interpretation method using Landsat images, provide land cover information over China from 1980s to 2015. The overall accuracy of CLUDs reached 94.3% and 91.2% for level-1 and level-2 land cover classes, respectively (Liu et al., 2014). CLCD was generated with stable training samples derived from CLUDs and Landsat time series. Assessed with 5463 validation samples, CLCD obtained an overall accuracy of 79.31% (Yang and Huang, 2021).

**Table 1. Detailed information of GLC products and national-scale LC products used in this paper.**

| Product name | Satellite sensors | Year of reference | Access | Literature |
|---|---|---|---|---|
| Globeland30 | Landsat TM/ETM+ HJ-1 A/B | 2010 | http://www.globallandcover.com/ | (Chen et al., 2015) |
| FROM_GLC | Landsat TM/ETM+/OLI | 2015 | http://data.ess.tsinghua.edu.cn/ | (Gong et al., 2013) |
| GLC_FCS30 | Landsat OLI | 2015 | https://doi.org/10.5281/zenodo.3986872 | (Zhang et al., 2021) |
| GAUD | Landsat TM/ETM+/OLI | 2015 | https://doi.org/10.6084/m9.figshare.11513178.v1 | (Liu et al., 2020) |
| GFC | Landsat TM/ETM+ | 2015 | http://earthenginepartners.appspot.com/science-2013-global-forest | (Hansen et al., 2013) |
| JRC GSW | Landsat TM/ETM+/OLI | 2015 | http://global-surface-water.appspot.com/ | (Pekel et al., 2016) |
| GMW | ALOS PALSAR Landsat TM/ETM+ | 2015 | https://data.unep-wcmc.org/datasets/45 | (Bunting et al., 2018) |
| NLCD 2016 | Landsat TM /OLI | 2016 | https://www.mrlc.gov/data/nlcd-2016-land-cover-conus | (Yang et al., 2018) |
| CLUDs | Landsat TM HJ-1 CBERS-1 | 2015 | / | (Liu et al., 2014) |
| CLCD | Landsat TM/ETM+/OLI | 2015 | https://doi.org/10.5281/zenodo.4417810 | (Yang and Huang, 2021) |

" (Revised manuscript, Line 163-177)

The relationship between our classification system and the classification systems of three national-scale land cover products has been added in supplementary material:

"**Table S3. Relationship between our classification system and the classification systems of the three national-scale LC products.**

| Id | GLC-2015 | CLCD | CLUDs | NLCD 2016 |
|---|---|---|---|---|
| 10 | Cropland | Cropland | Rice paddy | Pasture |
| | | | Bare farmland | Cropland |
| | | | Orchard | |
| 20 | Forest | Forest | Wooden land | Deciduous forest |
| | | | | Evergreen forest |
| | | | | Mixed forest |
| 30 | Grassland | Grassland | Grassland, highly-covered | Grassland |
| | | | Grassland, medium-covered | |
| | | | Grassland, lowly-covered | |
| 40 | Shrubland | Shrub | Shrubland | Shrubland |
| 50 | Wetland | Wetland | Marshland | Woody wetlands |
| | | | Tidal flat | Herbaceous wetlands |
| | | | Salt marsh | |
| | | | Flooded flat | |
| 60 | Water bodies | Water | Rivers | Water |
| | | | Lakes | |
| | | | Reservoir and ponds | |
| 70 | Tundra | | | |
| 80 | Impervious surfaces | Impervious | Urban | Urban, open space |
| | | | Rural | Urban, low intensity |
| | | | Other construction sites | Urban, med. Intensity |
| | | | | Urban, high intensity |
| 90 | Bare land | Barren | Sandy land | Barren |
| | | | Gobi desert | |
| | | | Barren | |
| | | | Bare rocky land | |
| 100 | Permanent snow and ice | Snow/ice | Permanent snow and ice | Ice/snow |

" (Supplementary material with change)

Lastly, since we employed three national-scale products in the fusion process, all the related results about the GLC-2015 were updated in the revised manuscript. Meanwhile, we have re-uploaded the mapping results and changed the access to the GLC-2015 as follows:

"The improved global land cover map in 2015 with 30 m resolution is available at https://doi.org/10.6084/m9.figshare.22358143.v2 (Li et al., 2022)." (Revised manuscript, Line 847-848)


Response: Thanks for suggestion. The classification system used in our study was determined with the classification system of all the input maps taken into consideration. All single-class land cover products provided information for only one class with no subclass. For example, the GFC for 2015 provided the extent of forest but could not tell users where were broadleaf and needleleaf trees. For multi-class land cover products, most had a simple classification system. For example, the GlobeLand30 used a classification system that contained only 10 first-level classes. These input maps with a simple classification scheme had no contribution to the level-2 detailed land cover classes. So, the existing products used may be not enough to generate a new GLC product with a detailed classification system using the proposed fusion method. Therefore, we adopted a classification system that contains 10 major land cover classes. In future work, efforts will be made to improve the diversity of land cover classes in our GLC-2015 product. Since the data fusion method we proposed can be easily used to integrate a wide variety of land cover maps for different regions, an improved global land cover product with a fine classification system rather than a simple one-level classification system will be developed when land cover products with more diverse land cover classes are available. In addition, when there are enough source maps with detailed land cover classes, the large discrepancies in the definition and criteria to distinct level-2 land cover classes might still hinder the transformation into a uniform system. So, a feasible framework for the conversion of different level-2 classification systems into a uniform system should be created in the future. For example, the semantic similarity between each input map's scheme and the target classification system may facilitate the harmonization (Gao et al., 2020).

We have added the discussion about the classification system and the further work to improve the

diversity of land cover classes in our map. The detailed revision can be seen below.

"Lastly, most candidate LC products used a simple classification system without a level-2 classification system, so they made no contributions to a more detailed classification system when they served as source data for data fusion. Although some maps provided detailed LC classification results, such as the GLC_FCS30 and FROM_GLC for 2015, there might be several challenges in the standardization and uniformity of level-2 classification systems due to the large discrepancies in the definition and criteria. Therefore, the GLC-2015 adopted a simple classification system containing 10 major LC classes. In future work, measures will be taken to meet the expectation of a more detailed classification system for GLC mapping. An improved GLC product with a detailed classification system rather than a simple one-level classification system can be further developed based on the highly applicable and general DSET method whenever more products with diverse LC classes are available. Additionally, a feasible framework for the conversion of different level-2 classification systems into a uniform system should be developed." (Revised manuscript, Line 834-845)

Response: Thanks for suggestion. We have standardized the format of the reference.

**Reviewer #3:**

**General comment:**

**The other two reviewers have provided very professional suggestions for data and manuscript improvements. I fully agree with their opinions. Generally, I think this data has some value but I have two major concerns.**

Response: Thanks for the comment. These comments are very helpful for revising and improving our paper. The manuscript has been improved according to your and another reviewer's comments. The point-by-point responses are listed below in **blue**. The changes in our manuscript are marked with red.

**Comment #3-1. The first is about the classification system used. This study assimilated various information from other land use data products. However, the definitions of different land types greatly differ in each classification system. For example, in your data, 'cropland' is defined as 'Land areas used for food production and animal feed.', which means pasture was classified as cropland. However, pasture is classified as grassland in GLC_FCS30. When all these signals were combined, the approach used could cause problems depending on the LC definitions adopted. Moreover, during the validation process, how this definition differences were treated? I think, for example for the cropland, it does not indicate that GLC_FCS30 was wrong (as claimed in Line565), but because of the differed definitions. Besides, it is unclear how you deal with fallow land, which is often been mixed with grassland in classifications (need to be separated by examine the temporal information). Another example, in FROM_GLC, the forest was defined as tree cover >=10%, and GlobeLand30 defined forest as land with tree cover above 30% and also include sparse woodland with tree cover between 10%-30%. These are different from the fixed threshold of 30% adopted in this study. In other words, I do not agree that other datasets are 'wrong' (as claimed in Line565 and other places).**

Response: Thank you very much for the insightful comment on the classification system adopted in the study. We are sorry for the unclear explanation in the previous manuscript. We agree that LC definitions in each GLC product have great difference, which might cause uncertainties when integrating various products. We have made every effort to reduce the uncertainties from the discrepancy between various classification systems.

First, we employed a simple classification system containing 10 major LC classes for GLC-2015 (Table 2). This classification system is the same as that adopted by Globeland30. We sincerely apologize for not clearly describing the definitions of forest and shrubland in the classification system used in the study, which confused the reviewers and our readers. Correspondingly, we have supplemented the definitions of these two categories in Table 2. **In our classification system, the forest includes trees with a tree canopy cover over 30% and sparse trees with a tree canopy cover between 10% - 30%, which is the same as Globeland30.**

"**Table 2. Classification system adopted in this paper.**

| Id | LC class | Definition |
|---|---|---|
| 10 | Cropland | Land areas used for food production and animal feed. |
| 20 | Forest | Land areas dominated by trees with tree canopy cover over 30%, and sparse trees with tree canopy cover between 10%-30%. |
| 30 | Grassland | Land areas dominated by natural grass with a cover over 10%. |
| 40 | Shrubland | Land areas dominated by shrubs with a cover over 30%, including |

mountain shrubs, deciduous shrubs, evergreen shrubs and desert shrubs with a cover over 10%.

| | | | |
|---|---|---|---|
| 50 | Wetland | | Land areas dominated by wetland plants and water bodies. |
| 60 | Water bodies | | Land areas covered with accumulated liquid water. |
| 70 | Tundra | | Land areas dominated by lichen, moss, hardly perennial herb and shrubs in the polar regions. |
| 80 | Impervious surfaces | | Land areas covered with artificial structures. |
| 90 | Bare land | | Land areas with scarce vegetation with a cover lower than 10%. |
| 100 | Permanent snow and ice | | Land areas dominated by permanent snow, glacier and icecap. |

" (Revised manuscript, Line)

Second, according to the classification system adopted in the study, the original LC classes of FROM_GLC and GLC_FCS30 were converted into the 10 target land cover classes based on the similarity of LC definition (Table S2). It can be found that 10 level-1 classes of the FROM_GLC and 9 level-0 classes of the GLC_FCS30 are the same as the LC classes used in the target classification system despite that the definitions of some classes differ. For the FROM_GLC, all the level-2 classes, excluding pasture, were aggregated into their corresponding level-1 classes. Note that the cropland in our classification system was defined as land areas for food production and animal feed. **Therefore, "pasture" in level-2 classes of the FROM_GLC was converted into cropland rather than grassland.** For the GLC_FCS30, all fine LC classes excluding lichens/mosses were aggregated into their corresponding 9 level-0 classes. Although the level-0 classification system of the GLC_FCS30 lacks tundra, lichens/mosses in the level-2 detailed LC classes has little distinction with tundra. **Separately, we transformed Lichens/mosses into the tundra, one of the major classes in our classification system**.

**Table S2. Relationship between our classification system and the classification systems of the three GLC products.**

| Id | GLC-2015 | Globeland30 | FROM_GLC | GLC_FCS30 |
|---|---|---|---|---|
| 10 | Cropland | Cultivated land | Rice paddy | Rain-fed cropland |
| | | | Greenhouse | Herbaceous cover |
| | | | Other/orchard | Tree or shrub cover (orchard) |
| | | | Bare farmland | Irrigated cropland |
| | | | Pasture | |
| 20 | Forest | Forest | Broadleaf, leaf-on | Evergreen broadleaved forest |
| | | | Broadleaf, leaf-off | Deciduous broadleaved forest |
| | | | | Open/closed deciduous broadleaved forest |
| | | | Needleleaf, leaf-on | Evergreen needleleaved forest |
| | | | | Open/closed evergreen needleleaved forest |
| | | | Needleleaf, leaf-off | Deciduous needleleaved forest |

| | | | | |
|---|---|---|---|---|
| | | | | Open/closed deciduous needleleaved forest |
| | | | Mixed leaf, leaf-on | Mixed leaf forest |
| | | | Mixed leaf, leaf-off | |
| 30 | Grassland | Grassland | Natural grassland | Grassland |
| | | | Grassland, leaf-off | |
| 40 | Shrubland | Shrubland | | Shrubland |
| | | | Shrubland, leaf-on | Evergreen shrubland |
| | | | Shrubland, leaf-off | Deciduous shrubland |
| 50 | Wetland | Wetland | Marshland | Wetlands |
| | | | Mudflat | |
| | | | Marshland, leaf-off | |
| 60 | Water bodies | Water bodies | Water | Water body |
| 70 | Tundra | Tundra | Shrub and brush tundra | |
| | | | Herbaceous tundra | Lichens/ mosses |
| 80 | Impervious surfaces | Artificial surfaces | Impervious surfaces | Impervious surfaces |
| 90 | Bare land | Bare land | Bare land | Sparse vegetation |
| | | | | Sparse shrubland |
| | | | | Sparse herbaceous cover |
| | | | | Bare areas |
| | | | | Consolidated/unconsolidated bare areas |
| 100 | Permanent snow and ice | Permanent snow and ice | Snow | Permanent ice and snow |
| | | | Ice | |

By carefully considering the original LC definitions in each product and the similarity between various classification systems, we managed to transform these various classification systems into a uniform one with the principle of minimizing potential errors and inconsistencies caused by different classification systems. Even though, there are still uncertainties caused by the harmonization of classification systems. For example, in GLC_FCS30, the sparse herbaceous cover with a vegetation cover below 15% was directly transformed into bare land regardless that our classification system distinguished grassland using vegetation cover threshold of 10%. In this case, herbaceous cover between 10%-15% in GLC_FCS30 was inappropriately transformed into bare land rather than grassland in the study. Due to the different LC

definitions, these uncertainties in classification system conversion are inevitable (Zhang et al., 2017). However, we conducted a reliability evaluation of the candidate maps to reduce the effects of uncertainties in classification system conversion on the fusion using the DSET. **Note that all the point-based samples used for reliability evaluation were labeled referring to the LC definitions in our classification system.** When evaluating the reliability of candidate maps for BPA construction in the fusion, **all the maps were assessed under the criterion of the classification system we used**. For instance, herbaceous cover between 10%-15% in GLC_FCS30 was transformed into bare land, while point-based samples in areas with herbaceous cover between 10%-15% were labeled as grassland. In this case, bare land with the threshold between 10%-15% from GLC_FCS30 was confirmed to mismatch our classification system, and the GLC_FCS30 was assessed to have lower accuracy for areas where the mismatched information existed. When we integrated all the maps grid by grid, the mismatched information would contribute less to the output map. **Similarly, during the validation process, the mapping accuracy of Globeland30, FROM_GLC, and GLC_FCS30 was assessed under the criterion of the classification system we used.** In this case, the accuracy assessment results represented the consistency of each product with validation samples labeled referring to our classification system.

As for the fallow land you are concerned with, we treated it as cropland according to the definition in our classification system. So, the fallow land from any other products was converted into cropland. If the candidate maps showed confusion with fallow land and grassland, this misclassification might bias our mapping results since the GLC-2015 was developed based on the integration of the candidate maps.

Corresponding, we have added how we harmonized the different classification systems in the revised manuscript.

"According to the classification system adopted in the study, the original LC classes of FROM_GLC and GLC_FCS30, CLUD for 2015, and NLCD 2016 for 2016 were converted into the 10 target land cover classes based on the similarity of LC definition. Note that cropland in our classification system was defined as land areas for food production and animal feed. Therefore, pasture in level-2 classes of the FROM_GLC was converted into cropland rather than grassland. In addition, lichens/mosses in the level-2 detailed LC classes of GLC_FCS30 was converted into tundra." (Revised manuscript, Line 307-312)

In addition, we have added the discussion about the uncertainties brought by the LC definition differences:

" Third, due to the different LC definitions, uncertainties in classification system conversion are inevitable (Zhang et al., 2017), which might cause problems for the fusion based on the DSET method. However, we conducted a reliability evaluation of the candidate maps to reduce the influence of uncertainties in classification system conversion on the fusion. The point-based samples used for reliability evaluation were labeled referring to the LC definitions in our classification system so that all the maps were evaluated under the criterion of the classification system we used. By the reliability evaluation, the candidate maps were assessed to have lower accuracy for areas with mismatched information. When integrating all the maps grid by grid, the mismatched information would contribute less to the fusion." (Revised manuscript, Line 826-833)


Response: Thank you very much for pointing out this issue. We are really sorry for the inappropriate words against other GLC products. This was not our intention. We have tried our best to revise our manuscript to address this concern. **We carefully checked our words and replaced those impolite remarks with more modest ones in the revised manuscript**. For example, "worst performance", "wrongly", "overwhelming superiority", "great superiority" were replace with "lowest accuracy", "misclassified", "better mapping performance".

Meanwhile, we agree with you that the accuracy differences are partially from the discrepancy of LC definition in each classification system. In the LC products comparison, we chose a classification system containing 10 major LC classes as the basic system and reclassified the detailed LC classes of FROM_GLC and GLC_FCS30. During the classification system conversion, uncertainties are inevitable. However, the mapping accuracy of different GLC products was assessed with the criterion of the classification system we used. Thus, some LC classes in FROM_GLC and GLC_FCS30 regarded as accurate classification based on their original classification system disagreed with the validation samples and obtained relatively low accuracy.

In addition, the Globeland30, FROM_GLC, and GLC_FCS30 are excellent and indispensable GLC products that provide comprehensive and reliable information about the Earth's surface. These products have been widely used by researchers, policymakers, and other stakeholders worldwide. As GLC products at 30m resolution, the Globeland30, FROM_GLC, and GLC_FCS30 have provided the fundamental information for various applications, such as biodiversity conservation (Wu et al., 2020; Meng et al., 2023), climate change (Kim et al., 2016; Xue et al., 2021; Zheng et al., 2022), and land management (Shafizadeh-Moghadam et al., 2019), despite that they show unstable performance in certain LC classes and some specific areas (Sun et al., 2016; Kang et al., 2020). Thanks to these products, we developed the GLC-2015. Although data inter-comparison has shown that the GLC-2015 had some improvements, there are still some limitation, such as inaccurate mapping results for grassland, shrubland, and wetland. These issues should be the focus of future work. In any case, the GLC-2015 can complete the 30m-resolution GLC product pool and provide better data support for global change research and sustainable development in conjunction with the existing products.

Correspondingly, we have added the recognition of the Globeland30, FROM_GLC, and GLC_FCS30 in Introduction Section:

"The Globeland30, FROM_GLC, and GLC_FCS30 are excellent and indispensable GLC products which have contributed much to various researches, such as biodiversity conservation (Wu et al., 2020; Meng et al., 2023), climate change (Kim et al., 2016; Xue et al., 2021; Zheng et al., 2022), and land management (Shafizadeh-Moghadam et al., 2019)." (Revised manuscript, Line 70-74)

Response: Thanks for the comment. We used the local adaptive fusion model to combine the existing products for each grid. To avoid the inequacy in the size of local samples for rare land cover classes, we also used the global samples to evaluate products' reliability. Since the local samples play a more critical role in the local accuracy assessment, a higher weight should be assigned to the local samples in the construction of the BPA for each grid. To define the weights for the local and global samples, we randomly selected 8 geographical grids of 4°×4° to conduct a pre-test. The weight of the local samples was set from 60% to 90%, with 5% as an interval. The accuracies of the mapping results for each grid with different weights were calculated (FigS1). It was found that in some grinds, the performance of the fusion method was influenced by the weights. When the local samples counted for 75% of the whole sample set and the global samples counted for 25%, the fusion method exhibited robust performance and achieved relatively high accuracy.

[Figure]

**Figure S1. The relationship between the overall accuracy and the weight of local accuracy in BPA construction**

Correspondingly, we have explained why we used 75% and 25% as two thresholds in our manuscript.

"Given that the local accuracy for a 4°×4° grid was not able to adequately reflect the actual land cover landscape, especially for the rare LC classes, the global accuracy was incorporated into the construction of the BPA to avoid uncertainties from a local point of view. Since the assessment based on local samples plays a more critical role in BPA construction for a local grid, a higher weight should be assigned to local accuracy. To identify the best weight, we tested different weights of the local accuracy (see Figure S1). The result shows that using 75% performed robustly and obtained relatively higher overall accuracy. Therefore, we chose 75% as the weight for local accuracy and 25% for global accuracy." (Revised manuscript, Line 352-359)

**Comment #3-5. Line315, error matrix?**

Response: Thanks for comment. We are sorry for this slip of the pen. Instead, we have revised the wrong phrase as "confusion matrix" throughout the manuscript.

**Comment #3-6. Lines453 and 462, 'worst' should be replaced by, for example, lowest.**

Response: Thanks for suggestion. We have revised as suggested.

**Comment #3-7. Line537, 'overwhelming superiority' is too far. As I mentioned, it might because of the definition differences.**

Response: Thanks for comment. We agree that the definition difference might cause uncertainties in accuracy comparison between GLC products. We have replaced the original phrase with "outperformance".

**Comment #3-8. Line601, 'show great superiority over others', same as above.**

Response: Thanks for comment. The original words have been revised as "show better performance than others".

**Comment #3-9. Line585, 'capture most human activity'. I don't agree, 'capture the footprint of human activities' might be better.**

Response: Thanks for suggestion. The original expression has been replaced by "capture the footprint of human activities".

---

## Author Response (AR3)

Dear Editor and Referees:

We are particularly grateful for your careful reading, and for giving us many constructive comments on this work. According to the comments and suggestions, we have tried our best to improve the previous manuscript ESSD-2022-142 (An improved global land cover mapping in 2015 with 30 m resolution (GLC-2015) based on a multi-source product fusion approach). The modified words or sentences are marked as blue color in the revised manuscript. We are providing an item-by-item response to all questions and recommendations.

Thanks very much for your time.

Best regards,

Xiaoping Liu and all co-authors

**Reviewer #1:**

**General comment:**

**This study developed a 30m resolution GLC product by integrating multiple products using the DSET method. Accuracy assessment with two validation datasets demonstrates the high quality of the GLC-2015. The comparison between GLC-2015 and other GLC products (Globeland30, FROM_GLC, GLC_FCS30) is comprehensive and the analysis is reasonable. This data is valuable and can provide accurate information for many applications. Some improvements are needed before publication.**

Response: Thanks for the comment. These comments are very helpful for revising and improving our paper. The manuscript has been improved according to your and another reviewer's comments. The point-by-point responses are listed below in **blue**. The changes in our manuscript are marked with red.

**Comment #1-1. The authors used three national-scale products to improve the quality of the GLC-2015 in China and America. However, it is unknown that whether the GLC-2015 performed better than these national-scale products. It is advised that the authors quantitatively compared the GLC-2015 with NLCD, CLCD and CLUD. Also, area difference for various LC classes in GLC-2015 and national-scale data can be analyzed.**

Response: Thanks for the comment. Based on your suggestion, we have compared the accuracy of GLC-2015 with three national-scale products. Also, we quantified the area difference for each land cover class in our revised manuscript.

"**4.4 Inter-comparison with national-scale products**

Except for comparison with the existing GLC products, the GLC-2015 was also compared with three national-scale products (CLCD, CLUD, and NLCD 2016 over CONUS). We first compared the accuracy of the GLC-2015 with NLCD, CLCD, and CLUD using the point-based samples (Tables S5-S6). It can be found that the GLC-2015 obtained an overall accuracy of 88.8% in China, higher than CLCD (78.3%) and CLUD (70.2%). Specifically, the GLC-2015 achieved the highest PA and UA in all LC classes except wetland. In the CONUS, the GLC-2015 outperformed NCLD 2016 with an OA improvement of 13.2%. Additionally, the GLC-2015 exhibited better mapping performance in nearly all LC classes." (Revised manuscript, Line 684-692)

"**Table S5. Comparison of mapping accuracy for the GLC-2015, CLCD, and CLUD via point-based samples.**

| | | Cropland | Forest | Grassland | Shrubland | Wetland | Water bodies | Impervious surfaces | Bare land | Permanent snow and ice | OA (Kappa coefficient) |
|---|---|---|---|---|---|---|---|---|---|---|---|
| GLC-2015 | PA | 0.844 | 0.965 | 0.968 | 0.316 | 0.598 | 0.896 | 0.905 | 0.891 | 0.793 | 0.888 |
| | UA | 0.930 | 0.928 | 0.803 | 0.923 | 0.870 | 0.741 | 0.899 | 0.962 | 0.958 | (0.864) |
| CLCD | PA | 0.812 | 0.893 | 0.939 | 0.079 | 0.009 | 0.742 | 0.671 | 0.767 | 0.737 | 0.783 |
| | UA | 0.812 | 0.874 | 0.635 | 0.600 | 1.000 | 0.857 | 0.793 | 0.907 | 0.808 | (0.734) |
| CLUD | PA | 0.715 | 0.590 | 0.793 | 0.158 | 0.704 | 0.691 | 0.759 | 0.763 | 0.439 | 0.702 |
| | UA | 0.779 | 0.800 | 0.604 | 0.062 | 0.864 | 0.807 | 0.782 | 0.753 | 0.893 | (0.639) |

**Table S6. Comparison of mapping accuracy for the GLC-2015 and NLCD 2016 via point-based samples.**

| | | Cropland | Forest | Grassland | Shrubland | Wetland | Water bodies | Impervious surfaces | Bare land | Permanent snow and ice | OA (Kappa coefficient) |
|---|---|---|---|---|---|---|---|---|---|---|---|
| GLC-2015 | PA | 0.890 | 0.958 | 0.917 | 0.869 | 0.903 | 0.935 | 0.867 | 0.911 | 1.000 | 0.910 |
| | UA | 0.944 | 0.932 | 0.815 | 0.972 | 0.878 | 0.977 | 0.903 | 0.689 | 1.000 | (0.893) |
| NLCD 2016 | PA | 0.824 | 0.760 | 0.617 | 0.862 | 0.873 | 0.830 | 0.800 | 0.446 | 0.750 | 0.778 |
| | UA | 0.849 | 0.982 | 0.594 | 0.641 | 0.899 | 0.902 | 0.714 | 0.439 | 1.000 | (0.736) |

” (Supplementary material with change)

“An accuracy comparison between the GLC-2015 and three national-scale products was also performed using the patch-based samples (Tables S7-S8). Overall, the GLC-2015 achieved a better OA of 85.7% in China, with respect to CLCD (83.6%) and CLUD (75.4%). In terms of PA and UA, the GLC-2015 ranked first or second in most LC classes. In the CONUS, the GLC-2015 possessed an OA of 84.5% and a kappa coefficient of 0.787, outperforming NLCD 2016. Although the GLC-2015 had lower PAs in wetland and impervious surfaces, and lower UAs in cropland and forest compared to NLCD 2016, the GLC-2015 outperformed NLCD 2016 in most LC classes.” (Revised manuscript, Line 693-699)

**Table S7. Comparison of mapping accuracy for the GLC-2015, CLCD, and CLUD via patch-based samples.**

| | | Cropland | Forest | Grassland | Shrubland | Wetland | Water bodies | Impervious surfaces | Bare land | Permanent snow and ice | OA (Kappa coefficient) |
|---|---|---|---|---|---|---|---|---|---|---|---|
| GLC-2015 | PA | 0.915 | 0.914 | 0.512 | 0.002 | 0.000 | 0.915 | 0.837 | 0.397 | 0.841 | 0.857 |
| | UA | 0.929 | 0.922 | 0.075 | 0.005 | 0.000 | 0.770 | 0.805 | 0.953 | 0.700 | (0.789) |
| CLCD | PA | 0.916 | 0.914 | 0.497 | 0.000 | 0.000 | 0.846 | 0.742 | 0.280 | 0.856 | 0.836 |
| | UA | 0.900 | 0.925 | 0.065 | 0.000 | 0.000 | 0.873 | 0.757 | 0.930 | 0.633 | (0.755) |
| CLUD | PA | 0.831 | 0.782 | 0.478 | 0.002 | 0.385 | 0.823 | 0.703 | 0.280 | 0.875 | 0.754 |
| | UA | 0.892 | 0.906 | 0.041 | 0.000 | 0.023 | 0.733 | 0.686 | 0.900 | 0.652 | (0.647) |

**Table S8. Comparison of mapping accuracy for the GLC-2015 and NLCD 2016 via patch-based samples.**

| | | Cropland | Forest | Grassland | Shrubland | Wetland | Water bodies | Impervious surfaces | Bare land | OA (Kappa coefficient) |
|---|---|---|---|---|---|---|---|---|---|---|
| GLC-2015 | PA | 0.924 | 0.514 | 0.788 | 0.905 | 0.024 | 0.911 | 0.747 | 0.691 | 0.845 |
| | UA | 0.873 | 0.718 | 0.840 | 0.916 | 0.019 | 0.916 | 0.686 | 0.683 | (0.787) |
| NLCD 2016 | PA | 0.871 | 0.369 | 0.787 | 0.686 | 0.054 | 0.906 | 0.796 | 0.676 | 0.769 |
| | UA | 0.879 | 0.809 | 0.788 | 0.847 | 0.001 | 0.913 | 0.395 | 0.361 | (0.690) |

” (Supplementary material with change)

“We further performed an areal comparison for each LC class of GLC-2015 and three national-scale products (Figures S12-S13). Generally, the GLC-2015, CLCD, and CLUD exhibited similar areas in most classes. Notably, the areas of cropland, shrubland, and wetland in GLC-2015 were very close to CLCD but different from CLUD. In the CONUS, the areas of cropland, water bodies, and bare land in the GLC-2015 and NLCD 2016 were close. In contrast, the areas of the remaining LC classes in the GLC-2015 showed a large difference from NLCD 2016. The area differences in forest, grassland and shrubland between GLC-2015 and NLCD 2016 were mainly related to different LC definitions. For example, the minimum fraction of tree cover in the forest is 10% in GLC-2015, whereas NLCD 2016 used a minimum fraction of 20%. NCLD 2016 had higher area of impervious surfaces than the GLC-

2015 because open urban in NLCD 2016 includes too much vegetation." (Revised manuscript, Line 700-709)

"

[Figure]

**Figure S12. Areal comparison of various land cover classes among the GLC-2015, CLCD and CLUD. Class IDs 10, 20, 30, 40, 50, 60, 80, 90, and 100 denote cropland, forest, grassland, shrubland, wetland, water bodies, impervious surfaces, bare land, and permanent snow and sea ice, respectively.**

[Figure]

**Figure S13. Areal comparison of various land cover classes among the GLC-2015 and NLCD 2016. Class IDs 10, 20, 30, 40, 50, 60, 80, 90 denote cropland, forest, grassland, shrubland, wetland, water bodies, impervious surfaces, and bare land, respectively."** (Supplementary material with change)

**Comment #1-2. National-scale land cover products, such as CLUD, NLCD 2016, have a two-level classification system, how did you transform these classification systems into the adopted classification system? This should be explained in the study.**

Response: Thanks for the comment. The harmonization of different classification systems is the critical pre-process for the fusion. Based on the similarity of LC definition, a translation table (see Table S3) was made to converse the classification system of CLUD and NLCD 2016 to the target classification system.

**Table S3. Classification systems of three national-scale LC products and the translation table.**

| Id | GLC-2015 | CLCD | CLUDs | NLCD 2016 |
|---|---|---|---|---|
| 10 | Cropland | Cropland | Rice paddy | Pasture |
|  |  |  | Bare farmland | Cropland |
|  |  |  | Orchard |  |
| 20 | Forest | Forest | Wooden land | Deciduous forest |
|  |  |  |  | Evergreen forest |
|  |  |  |  | Mixed forest |
| 30 | Grassland | Grassland | Grassland, highly-covered | Grassland |
|  |  |  | Grassland, medium-covered |  |
|  |  |  | Grassland, lowly-covered |  |
| 40 | Shrubland | Shrub | Shrubland | Shrubland |
| 50 | Wetland | Wetland | Marshland | Woody wetlands |
|  |  |  | Tidal flat | Herbaceous wetlands |
|  |  |  | Salt marsh |  |
|  |  |  | Flooded flat |  |
| 60 | Water bodies | Water | Rivers | Water |
|  |  |  | Lakes |  |
|  |  |  | Reservoir and ponds |  |
| 70 | Tundra |  |  |  |
| 80 | Impervious surfaces | Impervious | Urban | Urban, open space |
|  |  |  | Rural | Urban, low intensity |
|  |  |  | Other construction sites | Urban, med. Intensity |
|  |  |  |  | Urban, high intensity |
| 90 | Bare land | Barren | Sandy land | Barren |
|  |  |  | Gobi desert |  |
|  |  |  | Barren |  |
|  |  |  | Bare rocky land |  |

| 100 | Permanent snow and ice | Snow/ice | Permanent snow and ice | Ice/snow |

Correspondingly, we have added how we translated level-2 classification systems of CLUD and NLCD into the target classification system in the revised manuscript.

"According to the LC translation tables (Tables S2-S3), the original LC classes of FROM_GLC and GLC_FCS30, CLUD for 2015, and NLCD 2016 for 2016 were converted into the 10 target land cover classes based on the similarity of LC definition." (Revised manuscript, Line 309-311)

**Comment #1-3. The national-scale land cover datasets, such as CLCD for 2015, were used in the fusion, why these products were not listed in the framework (Figure 4)?**

Response: Thanks for the comment. We have updated the framework in our revised manuscript.

"

[Figure]

**Figure 4. The framework for generating the GLC-2015 map using a multi-source product fusion approach based on DEST.**" (Revised manuscript, Line 292-294)

**Comment #1-4. Figures S8-S11 exhibit visual comparisons for various land cover classes at local scale. However, the detailed locations of examples were not clearly showed. It would be better to tell readers the specific locations with graticules or central point as well as area size.**

Response: Thanks for the comment. We have added the scale bar as well as the latitude and longitude of the center for each example in the revised manuscript.

[Figure]

**Figure S8. Comparing the crop extent from GLC-2015 and other widely used products in three agricultural regions of Egypt (30.365°N, 30.189°E), China (27.508°N, 110.976°E), and USA (41.449°N, 99.934°W).**

[Figure]

**Figure S9. Comparing the forest extent from GLC-2015 and other widely used products in three forest-dominated regions of Congo (4.044°S, 25.851°E), China (35.791°N, 109.594°E), and USA (38.626°N, 78.189°E).**

[Figure]

**Figure S10. Comparing the wetland extent from GLC-2015 and other widely used products in three wetland-dominated regions of Canada (49.549°N, 95.701°W), USA (30.647°N, 82.5201°W), and Sundarbans (22.044° N, 89.203°E).**

[Figure]

**Figure S11. Comparing the impervious extent from GLC-2015 and other widely used products in three megacities: Tokyo (35.925°N, 139.716°E), Shanghai (31.148°N, 121.451°E), and New York (40.907°N, 73.936° W).**" (Supplementary material with change)

**Comment #1-5. In abstract, "LC" should be fully spelled.**

Response: Thanks for the comment. We have fully spelled "LC" as "land cover" in the revised manuscript.

**Comment #1-6. Line 116: do you mean "the final fused results based on the DSET method"?**

Response: Thanks for the comment. We are sorry for this mistake. We have corrected it.

**Comment #1-7. Line 128: "are" should be "were". And line 211: "resulted" should be "result".**

Response: Thanks for the comment. We have revised the manuscript according to the suggestions. In addition to that, we have carefully checked the expression of the article sentence by sentence.

**Comment #1-8. Some relevant papers may be reviewed in the references:**
**Mapping 10 m global impervious surface area (GISA-10m) using multi-source geospatial data. Earth System Science Data, 2022, 14: 3649–3672.**
**30-m global impervious surface area dynamics and urban expansion pattern observed by Landsat satellites: from 1972 to 2019. SCIENCE CHINA Earth Sciences, 2021, 64(11): 1922-1933.**

Response: Thanks for the comment. We think these two papers are very useful. Correspondingly, we have cited them in the Introduction.

[revised manuscript text omitted]